# DEBIASING MINI-BATCH QUADRATICS FOR APPLICATIONS IN DEEP LEARNING

**Lukas Tatzel, Bálint Mucsányi, Osane Hackel & Philipp Hennig**
Tübingen AI Center
University of Tübingen
Tübingen, Germany
`{lukas.tatzel,balint.mucsanyi}@uni-tuebingen.de`

## ABSTRACT

Quadratic approximations form a fundamental building block of machine learning methods. E.g., second-order optimizers try to find the Newton step into the minimum of a local quadratic proxy to the objective function; and the second-order approximation of a network's loss function can be used to quantify the uncertainty of its outputs via the Laplace approximation. When computations on the entire training set are intractable—typical for deep learning—the relevant quantities are computed on mini-batches. This, however, distorts and biases the shape of the associated *stochastic* quadratic approximations in an intricate way with detrimental effects on applications. In this paper, we (i) show that this bias introduces a systematic error, (ii) provide a theoretical explanation for it, (iii) explain its relevance for second-order optimization and uncertainty quantification via the Laplace approximation in deep learning, and (iv) develop and evaluate debiasing strategies.

## 1 INTRODUCTION

Quadratic approximations of the loss landscape are increasingly used by algorithms in deep learning, from pruning methods (Dong et al., 2017; Zeng & Urtasun, 2019) and influence functions (Koh & Liang, 2017) to second-order optimizers (Amari, 1998; Martens, 2010; Martens & Grosse, 2015; Grosse & Martens, 2016; Botev et al., 2017; Zhang et al., 2017; George et al., 2018; Martens et al., 2018; Osawa et al., 2019) and uncertainty quantification via the Laplace approximation (Ritter et al., 2018b;a; Kristiadi et al., 2020; Daxberger et al., 2021; Immer et al., 2021). When the computations are intractable on the entire training set—typical for deep learning—the quantities of interest are computed on mini-batches subsampled from the training data. The goal of this work is to highlight that mini-batching systematically biases the shape of a quadratic approximation.

**A systematic bias?** Figure 1 illustrates the phenomenon. It shows five *mini-batch* quadratics in their top-curvature 2D subspace for the fully trained ALL-CNN-C model on CIFAR-100 data. For comparison, the full-batch quadratic, where all quantities are evaluated on the *entire* training set, is projected into the *same* 2D subspace. Within that subspace, the two quadratics are quite different: The mini-batch quadratic is much "narrower" (exhibits larger curvature) than the full-batch version. Given that the full-batch quadratic is the "right" object to serve as the basis for, e.g., a Newton step or a Laplace approximation,[1] the mini-batch version is *not* a meaningful surrogate: Its Newton step is overly small and a Laplace approximation yields an overconfident uncertainty estimate.

**Contributions.** To enable stable and efficient stochastic second-order optimizers, as well as reliable techniques for uncertainty quantification, we analyze this phenomenon and develop strategies to mitigate it. More specifically, our contributions are as follows: (i) We study mini-batch quadratics empirically and show that their geometry is systematically biased; (ii) we provide an explanation for this phenomenon, explaining the bias as an instance of the classic regression to the mean (directions of extreme steepness/curvature for one particular mini-batch are less extreme for other mini-batches),

---

[1]We are *not* concerned with the approximation error arising from the quadratic approximation of the non-quadratic function $\mathcal{L}_{\mathrm{reg}}(\,\cdot\,;\mathcal{D}) \approx q(\,\cdot\,;\mathcal{D})$ (see Equation (2)), but only with the consequences of replacing the full-batch quantities by their mini-batch counterparts $q(\,\cdot\,;\mathcal{D}) \approx q(\,\cdot\,;\mathcal{B})$.

Top two eigenvectors computed on mini-batch

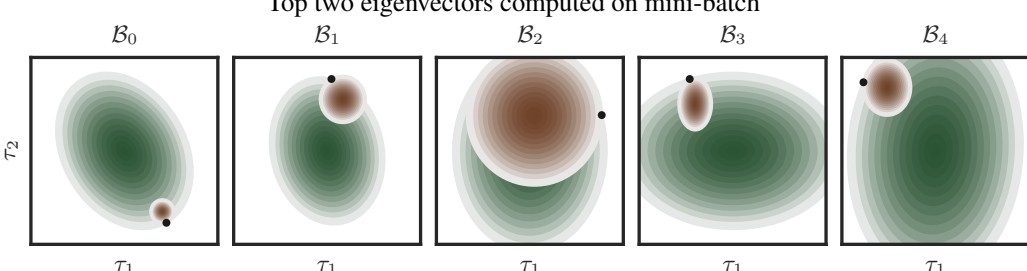

**Figure 1: A systematic bias?** We compute five mini-batch quadratics $q(\,\cdot\,; \mathcal{B}_m)$ with batch size $|\mathcal{B}_m| = 512$ for the loss landscape of the fully trained ALL-CNN-C model on CIFAR-100 data around $\boldsymbol{\theta}_0 \leftarrow \boldsymbol{\theta}_\star$ (shown as ●). Each mini-batch quadratic defines a 2D subspace spanned by the top two eigenvectors $\boldsymbol{u}_1, \boldsymbol{u}_2$ of $\boldsymbol{H}_{\mathcal{B}_m}$, in which we evaluate (i) the quadratic $q(\boldsymbol{\theta}_\star + \tau_1 \boldsymbol{u}_1 + \tau_2 \boldsymbol{u}_2; \mathcal{B}_m)$ itself (shown in �) and (ii) the full-batch quadratic $q(\boldsymbol{\theta}_\star + \tau_1 \boldsymbol{u}_1 + \tau_2 \boldsymbol{u}_2; \mathcal{D})$ (shown in ▮). In that subspace, the mini-batch quadratic is much "narrower" than the full-batch version which leads to overly small Newton steps and overconfident uncertainty estimates via the Laplace approximation.

(iii) we explain the relevance of this bias for second-order optimization and uncertainty quantification via the Laplace approximation, and (iv) develop and evaluate debiasing strategies.

## 2 NOTATION & BACKGROUND

**The regularized loss.** We consider a general supervised learning problem, where we try to find the optimal parameters $\boldsymbol{\theta}_\star = \arg\min_{\boldsymbol{\theta} \in \mathbb{R}^P} \mathcal{L}_{\mathrm{reg}}(\boldsymbol{\theta}, \mathcal{D})$ for the parameterized function $f_{\boldsymbol{\theta}} : \mathbb{R}^D \to \mathbb{R}^C$ by minimizing the regularized loss $\mathcal{L}_{\mathrm{reg}}$ on $N$ training examples $\mathbb{D} := \{(\boldsymbol{x}_n, \boldsymbol{y}_n) \in \mathbb{R}^D \times \mathbb{R}^C\}_{n \in \mathcal{D}}$,

$$\mathcal{L}_{\mathrm{reg}}(\boldsymbol{\theta}; \mathcal{D}) := \mathcal{L}(\boldsymbol{\theta}; \mathcal{D}) + r(\boldsymbol{\theta}) \quad \text{with} \quad \mathcal{L}(\boldsymbol{\theta}; \mathcal{D}) := \frac{1}{|\mathcal{D}|} \sum_{n \in \mathcal{D}} \ell(f_{\boldsymbol{\theta}}(\boldsymbol{x}_n), \boldsymbol{y}_n), \quad \mathcal{D} = \{1, \ldots, N\}. \tag{1}$$

The empirical risk $\mathcal{L}$ measures the dissimilarity between the model's predictions $f_{\boldsymbol{\theta}}(\boldsymbol{x}_n)$ and the true outputs $\boldsymbol{y}_n$ via a loss function $\ell : \mathbb{R}^C \times \mathbb{R}^C \to \mathbb{R}$. The regularizer $r : \mathbb{R}^P \to \mathbb{R}$, $r(\boldsymbol{\theta}) := \beta/2 \|\boldsymbol{\theta}\|_2^2$ with $\beta \in \mathbb{R}_{\geq 0}$, penalizes the "complexity" of the model.

### 2.1 LOCAL FULL-BATCH & MINI-BATCH QUADRATIC APPROXIMATIONS

**Full-batch quadratic.** A local quadratic approximation of the regularized loss around $\boldsymbol{\theta}_0 \in \mathbb{R}^P$ is given by the second-order Taylor expansion

$$\mathcal{L}_{\mathrm{reg}}(\boldsymbol{\theta}; \mathcal{D}) \approx q(\boldsymbol{\theta}; \mathcal{D}) := \frac{1}{2}(\boldsymbol{\theta} - \boldsymbol{\theta}_0)^\top \boldsymbol{H}_{\mathcal{D}}(\boldsymbol{\theta} - \boldsymbol{\theta}_0) + (\boldsymbol{\theta} - \boldsymbol{\theta}_0)^\top \boldsymbol{g}_{\mathcal{D}} + c_{\mathcal{D}}, \tag{2}$$

where $c_{\mathcal{D}} := \mathcal{L}_{\mathrm{reg}}(\boldsymbol{\theta}_0; \mathcal{D})$, $\boldsymbol{g}_{\mathcal{D}} := \nabla \mathcal{L}_{\mathrm{reg}}(\boldsymbol{\theta}_0; \mathcal{D})$ and $\boldsymbol{H}_{\mathcal{D}} := \nabla^2 \mathcal{L}_{\mathrm{reg}}(\boldsymbol{\theta}_0; \mathcal{D}) = \nabla^2 \mathcal{L}(\boldsymbol{\theta}_0; \mathcal{D}) + \beta \boldsymbol{I}$ is (some approximation of) the Hessian at $\boldsymbol{\theta}_0$ (all derivatives are with respect to the parameters $\boldsymbol{\theta}$ unless stated otherwise). As all quantities are evaluated on the *entire* training set $\mathcal{D}$, we refer to this as the *full-batch* quadratic. It holds that $\nabla q(\boldsymbol{\theta}; \mathcal{D}) = \boldsymbol{H}_{\mathcal{D}}(\boldsymbol{\theta} - \boldsymbol{\theta}_0) + \boldsymbol{g}_{\mathcal{D}}$ and $\nabla^2 q(\boldsymbol{\theta}; \mathcal{D}) \equiv \boldsymbol{H}_{\mathcal{D}}$.

**Mini-batch quadratic.** When the computations are intractable on the entire training set, the quantities in Equation (2) are typically computed on a *mini-batch*—a small randomly drawn subset—of the training data $\mathcal{B} \subset \mathcal{D}, |\mathcal{B}| \ll N$, resulting in a *stochastic* quadratic approximation $q(\,\cdot\,; \mathcal{B}) \approx q(\,\cdot\,; \mathcal{D})$. As $c_{\mathcal{B}}$, $\boldsymbol{g}_{\mathcal{B}}$ and $\boldsymbol{H}_{\mathcal{B}}$ are unbiased estimates of $c_{\mathcal{D}}$, $\boldsymbol{g}_{\mathcal{D}}$ and $\boldsymbol{H}_{\mathcal{D}}$, this substitution may seem innocent, but, as we will see in Section 3, it affects the geometry of the quadratic approximation *substantially*.

**Directional slope and curvature.** Consider a cut $r$ through the quadratic $q(\,\cdot\,; \mathcal{B})$ from $\boldsymbol{\theta}_\bullet \in \mathbb{R}^P$ along the normalized direction $\boldsymbol{d}, \|\boldsymbol{d}\| = 1$. It holds that (derivation in Appendix A.1) $r(\tau) := q(\boldsymbol{\theta}_\bullet + \tau \boldsymbol{d}; \mathcal{B}) = 1/2\tau^2 \boldsymbol{d}^\top \nabla^2 q(\boldsymbol{\theta}_\bullet; \mathcal{B}) \boldsymbol{d} + \tau \boldsymbol{d}^\top \nabla q(\boldsymbol{\theta}_\bullet; \mathcal{B}) + \text{const}$. So, as a function of $\tau$, $r : \mathbb{R} \to \mathbb{R}$ is a 1D parabola with derivatives $r'(\tau) = \tau \boldsymbol{d}^\top \nabla^2 q(\boldsymbol{\theta}_\bullet; \mathcal{B}) \boldsymbol{d} + \boldsymbol{d}^\top \nabla q(\boldsymbol{\theta}_\bullet; \mathcal{B})$ and $r''(\tau) \equiv \boldsymbol{d}^\top \nabla^2 q(\boldsymbol{\theta}_\bullet; \mathcal{B}) \boldsymbol{d}$. We denote the *directional* slope and curvature of the quadratic $q(\,\cdot, \mathcal{B})$ at $\boldsymbol{\theta}_\bullet$ in direction $\boldsymbol{d}$ by

$$\partial_{\boldsymbol{d}} q(\boldsymbol{\theta}_\bullet; \mathcal{B}) := r'(0) = \boldsymbol{d}^\top \nabla q(\boldsymbol{\theta}_\bullet; \mathcal{B}) \quad \text{and} \quad \partial_{\boldsymbol{d}}^2 q(\boldsymbol{\theta}_\bullet; \mathcal{B}) := r''(0) = \boldsymbol{d}^\top \nabla^2 q(\boldsymbol{\theta}_\bullet; \mathcal{B}) \boldsymbol{d}.$$

The directional slope and curvature at $\boldsymbol{\theta}_{\bullet}$ are thus simply the projections of the quadratic's gradient and Hessian at that location onto the direction.

**Eigenvalues as directional curvatures.** The directional curvature of the quadratic $q(\,\cdot\,;\mathcal{B})$ along one of $\boldsymbol{H}_{\mathcal{B}}$'s normalized eigenvectors $\boldsymbol{u}$ coincides with the corresponding eigenvalue $\lambda$ since $\partial_{\boldsymbol{u}}^2 q(\boldsymbol{\theta}_{\bullet};\mathcal{B}) = \boldsymbol{u}^\top \nabla^2 q(\boldsymbol{\theta}_{\bullet};\mathcal{B})\,\boldsymbol{u} = \boldsymbol{u}^\top \boldsymbol{H}_{\mathcal{B}}\,\boldsymbol{u} = \lambda\|\boldsymbol{u}\|^2 = \lambda$. Thus, in the context of a quadratic, an eigenvalue of the Hessian $\boldsymbol{H}_{\mathcal{B}}$ has a geometric interpretation as the directional curvature along the respective eigenvector.

**GGN & FIM.** Next, we discuss second-order optimization methods and the Laplace approximation (LA) for neural networks. Both techniques rely on local quadratic approximations of the regularized loss function and require a *positive definite* curvature matrix $\boldsymbol{H}_{\mathcal{B}}$. The empirical risk's Hessian $\nabla^2 \mathcal{L}(\boldsymbol{\theta}_0;\mathcal{B})$ can be *indefinite* and is therefore typically replaced by (an approximation of) the positive semi-definite Generalized Gauss-Newton matrix (GGN) $\boldsymbol{G}_{\mathcal{B}}$ or Fisher information matrix (FIM) $\boldsymbol{F}_{\mathcal{B}}$. In fact, GGN and FIM are often identical (Martens, 2020, Sec. 9.2). The resulting $\boldsymbol{H}_{\mathcal{B}}$ is positive definite if $\beta > 0$, or when a damping term $\delta\boldsymbol{I}$, $\delta \in \mathbb{R}_{>0}$ is added (e.g., in trust-region methods).

## 2.2 SECOND-ORDER METHODS & CONJUGATE GRADIENTS

**The Newton step.** Due to the simple polynomial form of the quadratic $q(\,\cdot\,;\mathcal{B})$, its minimum can be derived in closed form and is given by the Newton step $\arg\min_{\boldsymbol{\theta}} q(\boldsymbol{\theta};\mathcal{B}) = \boldsymbol{\theta}_0 - \boldsymbol{H}_{\mathcal{B}}^{-1}\boldsymbol{g}_{\mathcal{B}}$. This serves as the basis for second-order optimizers like L-BFGS (Liu & Nocedal, 1989; Nocedal, 1980), the Hessian-free approach (Martens, 2010) or K-FAC (Kronecker-factored approximate curvature) (Martens & Grosse, 2015; Grosse & Martens, 2016; Martens et al., 2018).

**Conjugate gradients.** We focus on the method of conjugate gradients (CG) (Hestenes & Stiefel, 1952), as it is a powerful method specifically designed to minimize quadratics with positive definite Hessians effectively (details in Appendix A.2). It is particularly useful in the context of large-scale optimization because it only requires access to matrix-vector products with the curvature matrix $\boldsymbol{v} \mapsto \boldsymbol{H}_{\mathcal{B}}\,\boldsymbol{v}$ that can be computed in a matrix-free manner (Pearlmutter, 1994; Schraudolph, 2002), i.e. without ever materializing the full matrix in memory—and it has been used successfully for training neural networks (Martens, 2010). Starting at $\boldsymbol{\theta}_0$, CG creates a sequence of iterates $(\boldsymbol{\theta}_0, \ldots, \boldsymbol{\theta}_P)$. In each iteration $p$, two main steps are performed: (i) Given the current position $\boldsymbol{\theta}_p$ and a normalized search direction $\boldsymbol{d}_p$, the algorithm finds the minimum of the quadratic along $\boldsymbol{\theta}_p + \tau \boldsymbol{d}_p$, i.e.

$$\boldsymbol{\theta}_{p+1} = \boldsymbol{\theta}_p + \tau_p \boldsymbol{d}_p \quad \text{with} \quad \tau_p := \underset{\tau \in \mathbb{R}}{\arg\min}\, q(\boldsymbol{\theta}_p + \tau \boldsymbol{d}_p;\mathcal{B}) = -\frac{\partial_{\boldsymbol{d}_p} q(\boldsymbol{\theta}_p;\mathcal{B})}{\partial_{\boldsymbol{d}_p}^2 q(\boldsymbol{\theta}_p;\mathcal{B})}. \tag{3}$$

In the second step (ii), CG constructs the next search direction $\boldsymbol{d}_{p+1}$ such that it is *conjugate* to all previous search directions, i.e. $\boldsymbol{d}_{p+1}^\top \boldsymbol{H}_{\mathcal{B}}\,\boldsymbol{d}_i = 0$ for all $i \in \{1, \ldots, p\}$.

## 2.3 LAPLACE APPROXIMATION FOR NEURAL NETWORKS

**Laplace approximation.** The Laplace approximation (LA) turns a trained standard neural network into a Bayesian neural network in a post-hoc manner (MacKay, 1991; Ritter et al., 2018b; Kristiadi et al., 2020; Daxberger et al., 2021) (details in Appendix A.3). The idea is to reinterpret the regularized loss $\mathcal{L}_{\mathrm{reg}}$ as the negative unnormalized log posterior $\log p(\boldsymbol{\theta} \mid \mathbb{D})$ of a specific Bayesian model. This interpretation identifies the optimal parameters $\boldsymbol{\theta}_\star = \arg\min_{\boldsymbol{\theta}} \mathcal{L}_{\mathrm{reg}}(\boldsymbol{\theta};\mathcal{D}) = \arg\max_{\boldsymbol{\theta}} p(\boldsymbol{\theta} \mid \mathbb{D})$ as the mode of the posterior, i.e. as the maximum a posteriori (MAP) estimate. A second-order approximation of the regularized loss around $\boldsymbol{\theta}_0 \leftarrow \boldsymbol{\theta}_\star$ then translates into a Gaussian approximation of the posterior—the so-called Laplace approximation (MacKay, 1992), i.e.

$$\mathcal{L}_{\mathrm{reg}}(\boldsymbol{\theta};\mathcal{D}) \approx q(\boldsymbol{\theta};\mathcal{D}) = \frac{1}{2}(\boldsymbol{\theta} - \boldsymbol{\theta}_\star)^\top \boldsymbol{H}_{\mathcal{D}}(\boldsymbol{\theta} - \boldsymbol{\theta}_\star) + \mathrm{const.} \quad \rightsquigarrow \quad p(\boldsymbol{\theta} \mid \mathbb{D}) \approx \mathcal{N}(\boldsymbol{\theta};\boldsymbol{\theta}_\star, \boldsymbol{\Sigma}_{\mathcal{D}}), \tag{4}$$

with $\boldsymbol{\Sigma}_{\mathcal{D}} := N^{-1}\boldsymbol{H}_{\mathcal{D}}^{-1}$ (or an approximation thereof). We obtain the predictive uncertainty $p(\boldsymbol{y}_\diamond \mid \boldsymbol{x}_\diamond, \mathbb{D})$ for some test input $\boldsymbol{x}_\diamond \in \mathbb{R}^D$ by propagating the parameter uncertainty $\mathcal{N}(\boldsymbol{\theta}_\star, \boldsymbol{\Sigma}_{\mathcal{D}})$ to the model's outputs via the linearized network (Immer et al., 2021; Roy et al., 2024).

**Full-batch vs. mini-batch LA.** It is common practice to compute the LA on the *entire* training set. Depending on the curvature approximation, this comes at considerable computational costs. For

example, a full-batch LA is prohibitive even for moderately sized models/datasets when a low-rank approximation of the Hessian is computed via repeated Hessian-vector products (each of which requires a full pass over the training data); and it is essentially infeasible for model classes like LLMs, which are trained on massive datasets. However, what adds most to the costs is that it is standard to *tune* the prior precision (Daxberger et al., 2021). This adds another outer loop that requires the same procedure to be performed multiple times. Being able to emulate the behavior of the full-batch quadratic on a mini-batch would thus be useful. We therefore study the mini-batch setting, i.e. we replace $q(\,\cdot\,;\mathcal{D})$ by $q(\,\cdot\,;\mathcal{B})$ in Equation (4). We will see in Section 6.2 that, when mitigating the associated biases, a mini-batch LA can be a good proxy for the full-batch LA.

## 3 THE SHAPE OF A MINI-BATCH QUADRATIC

This section studies the "shape" of a mini-batch quadratic $q(\,\cdot\,;\mathcal{B})$ and how it differs from $q(\,\cdot\,;\mathcal{D})$.

### 3.1 EMPIRICAL STUDY OF THE DIRECTIONAL SLOPES AND CURVATURES

**The high-curvature subspace is relevant for all common use cases.** In our empirical study, we focus on the *top*-curvature subspace. This subspace is particularly relevant for several reasons:

1. By the Eckart-Young-Mirsky Theorem, a truncated SVD is Frobenius norm-optimal, i.e. a low-rank approximation using the top eigenvectors is ideal from a theoretical perspective. As the spectrum of the Hessian typically decays quickly (Ghorbani et al., 2019), the bulk of the curvature information is contained in the top-curvature subspace.

2. In the context of optimization, it has been observed that the gradient mainly lives in the high-curvature space (Gur-Ari et al., 2018; Dangel et al., 2022). Thus, it makes sense for a second-order method to operate mainly in that space, as steps outside of it can not be expected to reduce the objective function significantly.

3. In the context of the LA, directions of large curvature correspond to directions in the weight space with *low* variance, i.e. these directions capture what we *know* about the model's parameters. Consequently, Daxberger et al. (2021, p. 19) describe a low-rank approximation of the Hessian based on its top eigenvectors.

**Experimental procedure.** We use the same setting as in Figure 1: The fully trained ALL-CNN-C model on the CIFAR-100 dataset. To isolate the effect of data subsampling, we eliminate *all other* sources of noise. Thus, we remove the dropout layers from the model. We use the cross-entropy loss, an $\ell^2$-regularizer, and train the model with SGD for 350 epochs, see Appendix B.1 for details.

We then pick a mini-batch $\mathcal{B}_m$ of size 512 and compute the 100 eigenvectors $\boldsymbol{u}_1, \ldots, \boldsymbol{u}_{100}$ to the 100 *largest* eigenvalues of $\boldsymbol{H}_{\mathcal{B}_m} \leftarrow \boldsymbol{G}_{\mathcal{B}_m} + \beta\boldsymbol{I}$. That is the Hessian of the $\ell^2$-regularized mini-batch loss, where we replaced $\nabla^2 \mathcal{L}(\boldsymbol{\theta}_\star; \mathcal{B}_m)$ with the GGN approximation. Next, we compute the directional slopes and curvatures for *all* mini-batch quadratics $q(\,\cdot\,;\mathcal{B}_{m'})$, $m' \in \{1, \ldots, M\}$ along those 100 eigenvectors. For a fixed eigenvector $\boldsymbol{u}_p$, the *average* of those directional slopes/curvatures over *all* mini-batches coincides with the directional slope/curvature of the full-batch quadratic, i.e.

$$\frac{1}{M}\sum_{m'=1}^{M}\underbrace{\partial_{\boldsymbol{u}_p} q(\boldsymbol{\theta}_\star; \mathcal{B}_{m'})}_{\text{one} \bullet \text{ and many} \bullet} = \underbrace{\partial_{\boldsymbol{u}_p} q(\boldsymbol{\theta}_\star; \mathcal{D})}_{+} \quad \text{and} \quad \frac{1}{M}\sum_{m'=1}^{M}\underbrace{\partial^2_{\boldsymbol{u}_p} q(\boldsymbol{\theta}_\star; \mathcal{B}_{m'})}_{\text{one} \bullet \text{ and many} \bullet} = \underbrace{\partial^2_{\boldsymbol{u}_p} q(\boldsymbol{\theta}_\star; \mathcal{D})}_{+}, \quad (5)$$

derivation in in Appendix A.1. The colored markers in Equation (5) refer to Figure 2. We repeat this procedure for three mini-batches $\mathcal{B}_m$, $m \in \{0, 1, 2\}$.

**Directional slopes and curvatures are biased.** Figure 2 reveals a systematic bias in the directional slopes and, more pronounced, in the directional curvatures: When eigenvectors and directional derivatives are computed on the *same* mini-batch, curvature is overestimated by roughly one order of magnitude compared to the curvature of the full-batch quadratic $q(\,\cdot\,;\mathcal{D})$. Within the space that actually carries curvature information—the top-eigenspace of the quadratic's Hessian—the curvature magnitude is thus not at all representative of the true underlying loss landscape. For *other* mini-batches, the directional slopes and curvatures are similar to the full-batch quadratic. This is because the averages in Equation (5) are dominated by these unbiased samples.

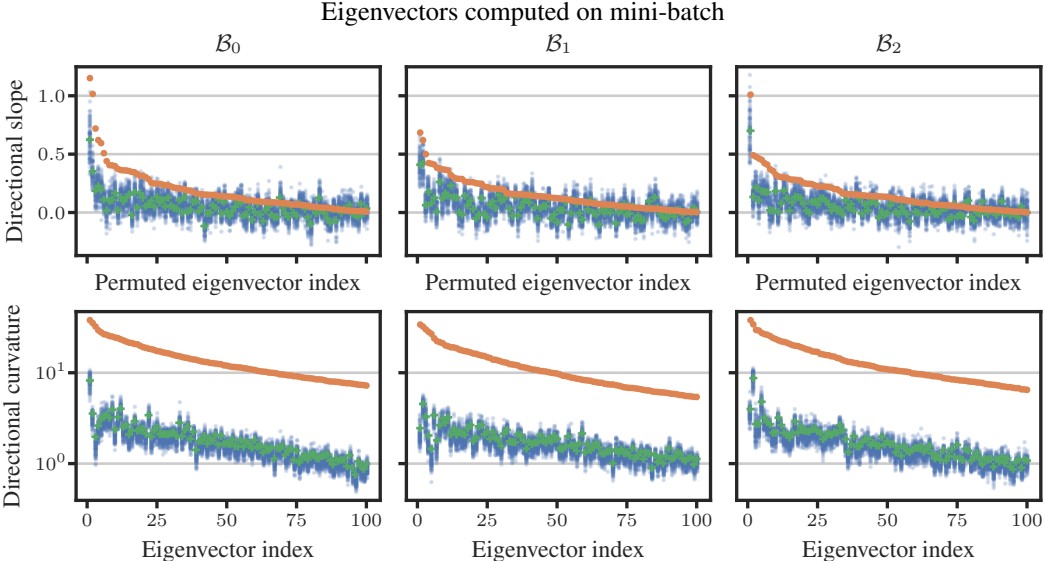

**Figure 2: Directional slopes and curvatures are biased.** We use the CIFAR-100 dataset with the fully trained ALL-CNN-C model and draw three mini-batches $\mathcal{B}_m$ of size $|\mathcal{B}_m| = 512$ to compute the top 100 eigenvectors $\boldsymbol{u}_1, \ldots, \boldsymbol{u}_{100}$. For each mini-batch/column, we show the directional slopes *(Top)* and curvatures *(Bottom)* evaluated on (i) $q(\boldsymbol{\theta}_\star; \mathcal{B}_m)$ (i.e. on the *same* mini-batch of data) as ●, (ii) $q(\boldsymbol{\theta}_\star; \mathcal{B}_{m'})$ for $m' \neq m$ (i.e. for all *other* mini-batches) as ● and (iii) the full-batch quadratic (the average of the orange and all blue dots, see Equation (5)) as ✚. For the top panel, we switch the order and sign of the eigenvectors such that the orange dots are all above zero and in descending order. There is a strong, systematic bias, particularly in the curvature: Computing the eigenvectors and directional curvatures on the same data results in over-estimation by roughly one order of magnitude.

We present additional results for batch size 2048 (instead of 512), the Hessian $\nabla^2 \mathcal{L}(\boldsymbol{\theta}_\star; \mathcal{B}_m)$ (instead of its GGN approximation) and CG search directions (instead of eigenvectors) in Appendix B.4. The bias is present in *all* settings, but less pronounced for larger batch sizes. For the CG search directions, the bias in directional slope is much more distinct than for the eigenvectors: The biased slope is always negative, while the full-batch quadratic's slope is in fact positive for most CG directions!

## 3.2 WHERE DO THE BIASES COME FROM?

In the following, we provide an explanation for the biases in the directional slopes and curvatures (i.e. the gap between the orange and a blue dot in Figure 2) from a theoretical perspective.

### 3.2.1 BIAS IN THE DIRECTIONAL SLOPE

Here, we focus on the CG search directions since the bias in the directional slope is even more pronounced for those directions than for the eigenvectors. This is not accidental! The CG directions are constructed from gradients—and gradients are directions that *maximize* the steepness ● for one particular mini-batch quadratic. For *other* mini-batch quadratics, the steepness along those directions (i.e. the directional slope ●) is therefore *less* extreme. We formalize this intuition in Appendix A.4.

### 3.2.2 BIAS IN THE DIRECTIONAL CURVATURE

Next, we consider the bias in the directional curvature along the eigenvectors of the curvature matrix.

**Directional curvature along $H_{\mathcal{B}}$'s eigenvectors.** Let $\boldsymbol{u}_1, \ldots, \boldsymbol{u}_P$ and $\tilde{\boldsymbol{u}}_1, \ldots, \tilde{\boldsymbol{u}}_P$ denote the eigenvectors of two mini-batch Hessians $H_{\mathcal{B}}$ and $H_{\tilde{\mathcal{B}}}$, respectively. Assume that the corresponding eigenvalues are in descending order, i.e. $\lambda_1 \geqslant \ldots \geqslant \lambda_P$ and $\tilde{\lambda}_1 \geqslant \ldots \geqslant \tilde{\lambda}_P$. We can write the

directional curvatures on $\mathcal{B}$ and $\tilde{\mathcal{B}}$ along an eigenvector $\boldsymbol{u}_i$ as (details in Appendix A.5)

$$\underbrace{\partial^2_{\boldsymbol{u}_i} q(\boldsymbol{\theta}_\bullet; \tilde{\mathcal{B}})}_{\bullet} = \boldsymbol{u}_i^\top \boldsymbol{H}_{\tilde{\mathcal{B}}} \boldsymbol{u}_i = \sum_{p=1}^P \tilde{\lambda}_p \, \boldsymbol{\Omega}_{i,p}, \quad \text{and} \quad \underbrace{\partial^2_{\boldsymbol{u}_i} q(\boldsymbol{\theta}_\bullet; \mathcal{B})}_{\bullet} = \boldsymbol{u}_i^\top \boldsymbol{H}_\mathcal{B} \boldsymbol{u}_i = \lambda_i. \tag{6}$$

The weights $\{\boldsymbol{\Omega}_{i,p} := (\boldsymbol{u}_i^\top \tilde{\boldsymbol{u}}_p)^2\}_{p=1}^P$ are non-negative and sum to one, i.e. $\sum_{p=1}^P \boldsymbol{\Omega}_{i,p} = 1$. The bias in the directional curvature thus originates from misalignment of the eigenspaces of the two curvature matrices—captured by the weights $\boldsymbol{\Omega}_{i,p}$—and/or a systematic difference in their spectra.

**Curvature bias is not due to differing spectra...** Assume that the eigenspaces are perfectly aligned, i.e. $\boldsymbol{\Omega}_{i,i} = 1$ and $\boldsymbol{\Omega}_{i,p} = 0 \; \forall p \neq i$. In this case, we obtain $\partial^2_{\boldsymbol{u}_i} q(\boldsymbol{\theta}_\bullet; \tilde{\mathcal{B}}) = \tilde{\lambda}_i$ from Equation (6), so the bias originates *exclusively* from the differences in the spectra. Figure 2 shows the eigenvalues (as directional curvatures) for three different CIFAR-100 mini-batches. As they are very similar, this cannot serve as the main explanation for the curvature bias.

**...but due to misaligned eigenspaces.** Now, assume that the spectra of $\boldsymbol{H}_\mathcal{B}$ and $\boldsymbol{H}_{\tilde{\mathcal{B}}}$ are identical, i.e. $\lambda_p = \tilde{\lambda}_p \; \forall p \in \{1, \dots, P\}$ which simplifies the unbiased estimate in Equation (6) to $\partial^2_{\boldsymbol{u}_i} q(\boldsymbol{\theta}_\bullet; \tilde{\mathcal{B}}) = \sum_{p=1}^P \lambda_p \, \boldsymbol{\Omega}_{i,p}$. Consider the curvature along $\boldsymbol{u}_1$ as an example. If $\boldsymbol{u}_1 = \tilde{\boldsymbol{u}}_1$, there is only one non-zero weight $\boldsymbol{\Omega}_{1,1} = 1$ and the two directional curvatures are identical. However, if there is significant overlap with any other eigenvectors (i.e. some weight is distributed also on the *lower*-curvature directions), the resulting curvature $\partial^2_{\boldsymbol{u}_1} q(\boldsymbol{\theta}_\bullet; \tilde{\mathcal{B}})$ is *smaller* than $\partial^2_{\boldsymbol{u}_1} q(\boldsymbol{\theta}_\bullet; \mathcal{B})$. Analogously, for $\boldsymbol{u}_P$, the directional curvature on $\tilde{\mathcal{B}}$ is *larger* than on $\mathcal{B}$ if there is significant overlap between $\boldsymbol{u}_P$ and some of $\boldsymbol{H}_{\tilde{\mathcal{B}}}$'s *higher*-curvature eigenvectors. This can be formalized by the following inequalities (derivation in Appendix A.5):

$$\underbrace{\partial^2_{\boldsymbol{u}_1} q(\boldsymbol{\theta}_\bullet; \tilde{\mathcal{B}})}_{\bullet} \leqslant \underbrace{\partial^2_{\boldsymbol{u}_1} q(\boldsymbol{\theta}_\bullet; \mathcal{B})}_{\bullet} \quad \text{and} \quad \underbrace{\partial^2_{\boldsymbol{u}_P} q(\boldsymbol{\theta}_\bullet; \tilde{\mathcal{B}})}_{\bullet} \geqslant \underbrace{\partial^2_{\boldsymbol{u}_P} q(\boldsymbol{\theta}_\bullet; \mathcal{B})}_{\bullet}. \tag{7}$$

In general, if the eigenspaces are not perfectly aligned such that weight is distributed on several eigenvectors, this leads to *less extreme* directional curvatures on mini-batch $\tilde{\mathcal{B}}$ on *both* ends of the spectrum. Figure 3 shows the weights $\boldsymbol{\Omega}_{i,p}$ as pixels. The overlap between the eigenspaces is far from perfect, i.e. one eigenvector from $\mathcal{B}$ overlaps with several eigenvectors from $\tilde{\mathcal{B}}$ to some extent. This explains the curvature overestimation in the top curvature subspace we observe in Figure 2 and identifies the misalignment of the eigenspaces as the dominating factor for the curvature bias.

### 3.2.3 SUMMARY OF THE THEORETICAL FINDINGS

The CG search directions and the eigenvectors of the curvature matrix are both *designed* to be "extreme" in some sense: The CG directions are based on gradients that maximize steepness of the quadratic $q(\,\cdot\,; \mathcal{B})$, while the top/bottom eigenvectors subsume directions of largest/smallest curvature. However, these directions are extreme only for one particular mini-batch $\mathcal{B}$. Another mini-batch $\tilde{\mathcal{B}}$ has its own extreme directions that typically differ from those of $\mathcal{B}$. Thus, **the extreme directions for $\mathcal{B}$ are less extreme for $\tilde{\mathcal{B}}$**. Projecting both quadratics onto $\mathcal{B}$'s directions consequently leads to extreme steepness and curvatures for $\mathcal{B}$, but less extreme values for $\tilde{\mathcal{B}}$. This result is an instance of the classic **regression to the mean** (Galton, 1886): Using an algorithm to find the directions of most extreme steepness/curvature in one particular mini-batch, we must expect the steepness/curvature to be *less* extreme on most *other* batches (and thus also on the entire dataset). We can expect this phenomenon to occur for other datasets, models, and curvature proxies as well, since the underlying mechanism is the stochasticity of the geometric information.

## 4 DEBIASING MINI-BATCH QUADRATICS FOR APPLICATIONS

We here argue that the biases affect second-order applications, and develop debiasing strategies.

### 4.1 IMPLICATIONS FOR SECOND-ORDER OPTIMIZATION AND THE LAPLACE APPROXIMATION

**Detrimental updates in second-order optimizers.** In the context of second-order optimization, both the biases in the directional slopes and curvatures are relevant. From Equation (3), the CG

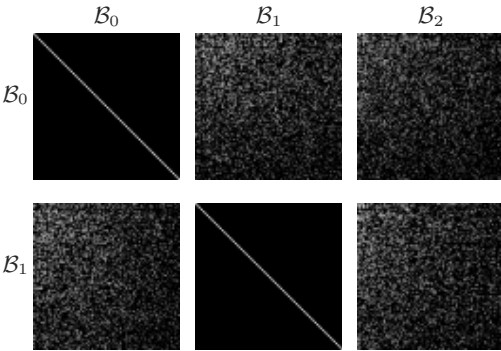

**Figure 3: In practice, eigenspaces are misaligned.** We reuse the setting of Figure 2 and compute the top 100 eigenvectors $\boldsymbol{U}_m \in \mathbb{R}^{P \times 100}$ for $\{\mathcal{B}_m\}_{m \in \{0,1,2\}}$. The weights $\boldsymbol{\Omega}_{i,j}$ are shown as a $100 \times 100$ greyscale image (color ranges from black for $\boldsymbol{\Omega}_{i,j} \leqslant 10^{-8}$ to white for $\boldsymbol{\Omega}_{i,j} = 1$) for $m \in \{0,1\}$, $m' \in \{0,1,2\}$. Clearly, the eigenspaces for different mini-batches are not perfectly aligned as eigenvectors from $\mathcal{B}_m$ overlap with several eigenvectors from $\mathcal{B}_{m'}$.

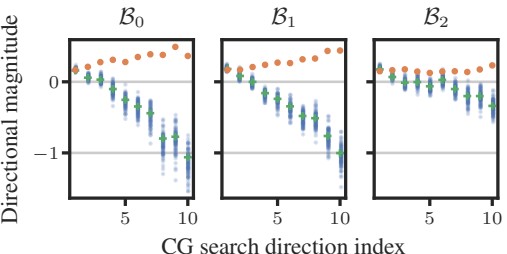

**Figure 4: CG update magnitudes are biased.** Same setting as Figure 9 *(Bottom)*. We run CG on $\{\mathcal{B}_m\}_{m \in \{0,1,2\}}$ and show the directional update magnitudes $\tau_1, \ldots, \tau_{10}$ for the first 10 CG steps using (i) the same mini-batch $\mathcal{B}_m$ (as ●), (ii) all other mini-batches (as ●) and (iii) the entire training set (as ➕). The magnitudes are given by the negative ratio of the directional slope and curvature (see Equation (3)) and thus inherit the attached biases. Note that most of the update magnitudes that are based on a single mini-batch of data (●) have the wrong sign resulting in detrimental updates in the wrong direction.

update magnitude $\tau_p$ to reach the minimum of $q(\boldsymbol{\theta}_p + \tau \boldsymbol{d}_p; \mathcal{B}_m)$ is given by the negative ratio of the directional slope and curvature at the current iterate $\boldsymbol{\theta}_p$ in the direction $\boldsymbol{d}_p$—that is the 1D Newton step along that cut.[2] As both these quantities are biased, so is $\tau_p$ as shown in Figure 4: While the update magnitudes for $\mathcal{B}_m$ are *always positive* (a property of CG), minimizing the other—*equally valid*—quadratics would require *negative* update magnitudes for most of the CG directions. In this sense, naive CG on a single mini-batch of data makes updates in the wrong direction. This can be attributed to the bias in the directional slope since the slope determines the sign of $\tau_p$. Overestimation of the directional curvature is "accidentally beneficial" in this case as it leads to smaller steps.

**Unreliable uncertainty quantification with the Laplace approximation.** For the LA, only the bias in the directional curvature is relevant. After Section 2.3, the approximate posterior covariance over the network's parameters is given by $\boldsymbol{\Sigma}_\mathcal{B} = N^{-1} \boldsymbol{H}_\mathcal{B}^{-1}$. Via the eigendecomposition $\boldsymbol{H}_\mathcal{B} = \sum_{p=1}^{P} \lambda_p \boldsymbol{u}_p \boldsymbol{u}_p^\top$, we obtain $\boldsymbol{\Sigma}_\mathcal{B} = N^{-1} \sum_{p=1}^{P} \lambda_p^{-1} \boldsymbol{u}_p \boldsymbol{u}_p^\top$. Due to the biases we describe in Section 3, the directional curvatures $\lambda_p$ are not representative of the true underlying curvature, resulting in a *deformed* posterior covariance $\boldsymbol{\Sigma}_\mathcal{B}$. Specifically, the overestimation of the curvature in the top-curvature subspace translates into an *underestimation* of the uncertainty in the posterior covariance (due to the inversion) with potentially severe consequences in safety-critical applications.

## 4.2 DEBIASING STRATEGIES

We now turn to strategies that mitigate the biases in second-order optimization and the LA. An empirical evaluation of these methods follows in Sections 6.1 and 6.2.

**Decoupling directions from magnitudes with a two-batch strategy.** The approaches we propose are simple yet effective. The idea is to commit to the imperfect directions from one mini-batch (as before) but use an *additional* mini-batch to estimate the directional derivatives. With this, we *decouple* the mechanism that determines the parameter subspace in which the method operates from the slope and curvature measurements within the space. Effectively, we use the blue dots from Figures 2 and 4 instead of the orange dots and thus obtain much more realistic estimates of the actual loss function's geometry (within the subspace defined by the first mini-batch). Next, we describe in more detail how this strategy can be applied to debias CG and the LA.

---

[2] The exact same argument can also be made for the eigenvector directions as the Newton step can be decomposed into 1D Newton steps along the eigenvectors.

### 4.2.1 DEBIASED CONJUGATE GRADIENTS

**Debiased approach.** For the debiased CG method, we need to implement two processes: (i) The first process applies $K \leqslant P$ CG iterations to the mini-batch quadratic $q(\,\cdot\,;\mathcal{B})$ and collects the search directions $(\boldsymbol{d}_1, \ldots, \boldsymbol{d}_K)$. As before, this defines the subspace in which CG operates. (ii) The second process recomputes the trajectory $(\tilde{\boldsymbol{\theta}}_1, \ldots, \tilde{\boldsymbol{\theta}}_K)$ within that subspace using debiased update magnitudes that are computed on a different mini-batch $\tilde{\mathcal{B}}$, i.e. we use $\tilde{\boldsymbol{\theta}}_0 := \boldsymbol{\theta}_0$ and

$$\tilde{\boldsymbol{\theta}}_{p+1} := \tilde{\boldsymbol{\theta}}_p + \tilde{\tau}_p \boldsymbol{d}_p \quad \text{with} \quad \underbrace{\tilde{\tau}_p := -\frac{\partial_{\boldsymbol{d}_p} q(\tilde{\boldsymbol{\theta}}_p; \tilde{\mathcal{B}})}{\partial^2_{\boldsymbol{d}_p} q(\tilde{\boldsymbol{\theta}}_p; \tilde{\mathcal{B}})}}_{\textcolor{blue}{\bullet}} \quad \text{instead of} \quad \underbrace{\tau_p = -\frac{\partial_{\boldsymbol{d}_p} q(\tilde{\boldsymbol{\theta}}_p; \mathcal{B})}{\partial^2_{\boldsymbol{d}_p} q(\tilde{\boldsymbol{\theta}}_p; \mathcal{B})}}_{\textcolor{orange}{\bullet}}. \quad (8)$$

For $\tilde{\mathcal{B}} = \mathcal{B}$, $(\tilde{\boldsymbol{\theta}}_1, \ldots, \tilde{\boldsymbol{\theta}}_K)$ is congruent with the original trajectory $(\boldsymbol{\theta}_1, \ldots, \boldsymbol{\theta}_K)$ from the single-batch CG approach. If the two mini-batches are different, the debiased trajectory will use updates into the directional minima of $q(\,\cdot\,;\tilde{\mathcal{B}})$ instead of $q(\,\cdot\,;\mathcal{B})$. Processes (i) and (ii) can either be run side by side in an alternating fashion (offering more flexibility regarding e.g. the termination criterion at the cost of a small memory overhead, details in Appendix A.2) or one after the other.

### 4.2.2 DEBIASED LAPLACE APPROXIMATION

**Laplace approximation with K-FAC.** We briefly explore another popular curvature proxy: Kronecker-Factored Approximate Curvature (K-FAC), which is commonly used both in optimization (Martens & Grosse, 2015; Martens et al., 2018; Eschenhagen et al., 2023) and uncertainty quantification (Ritter et al., 2018b). It is a block-diagonal approximation to the FIM $\boldsymbol{F}_{\mathcal{B}} \approx \boldsymbol{K}_{\mathcal{B}} := \text{blockdiag}_{l=1,\ldots,L}(\boldsymbol{K}_{\mathcal{B}}^{(l)})$, where each block $\boldsymbol{K}_{\mathcal{B}}^{(l)} := \boldsymbol{A}^{(l)} \otimes \boldsymbol{B}^{(l)}$ is approximated by the Kronecker product of two smaller matrices $\boldsymbol{A}^{(l)}$ and $\boldsymbol{B}^{(l)}$. Sampling from the respective LA with covariance $\boldsymbol{\Sigma}_{\mathcal{B}} = N^{-1}(\boldsymbol{K}_{\mathcal{B}} + \beta \boldsymbol{I})^{-1}$ can be performed efficiently due to the specific structure of the K-FAC approximation. For details, see Appendix A.3.2.

**Debiased approach.** For the debiased LA, we use two K-FAC approximations $\boldsymbol{K}_{\mathcal{B}}$ and $\boldsymbol{K}_{\tilde{\mathcal{B}}}$ computed on different mini-batches. Via the eigendecomposition $\boldsymbol{K}_{\mathcal{B}} = \boldsymbol{U}\boldsymbol{\Lambda}\boldsymbol{U}^\top$ with $\boldsymbol{U} = (\boldsymbol{u}_1, \ldots, \boldsymbol{u}_P)$ and $\boldsymbol{\Lambda} = \text{diag}(\lambda_1, \ldots, \lambda_P)$, we can re-write the covariance matrix as $\boldsymbol{\Sigma}_{\mathcal{B}} = N^{-1}(\boldsymbol{K}_{\mathcal{B}} + \beta \boldsymbol{I})^{-1} = N^{-1}\boldsymbol{U}(\boldsymbol{\Lambda} + \beta \boldsymbol{I})^{-1}\boldsymbol{U}$. For the debiased approach, we keep the eigenspace defined by $\boldsymbol{K}_{\mathcal{B}}$ but recompute the directional curvatures based on $\boldsymbol{K}_{\tilde{\mathcal{B}}}$, i.e. we use

$$\tilde{\boldsymbol{\Sigma}}_{\mathcal{B}} = \frac{1}{N}\boldsymbol{U}\big(\text{diag}(\tilde{\lambda}_1, \ldots, \tilde{\lambda}_P) + \beta \boldsymbol{I}\big)^{-1}\boldsymbol{U}^\top \quad \text{with} \quad \underbrace{\tilde{\lambda}_p = \boldsymbol{u}_p^\top \boldsymbol{K}_{\tilde{\mathcal{B}}}\boldsymbol{u}_p}_{\textcolor{blue}{\bullet}} \quad \text{instead of} \quad \underbrace{\lambda_p = \boldsymbol{u}_p^\top \boldsymbol{K}_{\mathcal{B}}\boldsymbol{u}_p}_{\textcolor{orange}{\bullet}}. \quad (9)$$

### 4.2.3 COMPUTATIONAL COST OF DEBIASING

Both debiasing techniques roughly double runtime compared to their single-batch counterparts: Debiased CG performs one extra matrix-vector product with $\boldsymbol{H}_{\tilde{\mathcal{B}}}$ per CG iteration to compute the debiased update magnitude (details in Appendix A.2). Debiased LA requires an additional K-FAC approximation $\boldsymbol{K}_{\tilde{\mathcal{B}}}$; all subsequent operations to compute the debiased K-FAC can be carried out efficiently at Kronecker factor level (details in Appendix A.3.3). In Section 6, we thus use the debiased versions at *half* the batch size, for a fair comparison. We will see that, at the resulting *similar* computational cost, the debiased versions clearly outperform the single-batch alternatives.

## 5 RELATED WORK

We here briefly list other, related forms of bias correction that have been suggested elsewhere.

**A different two-batch approach.** Other works have proposed to use different (not necessarily disjoint) mini-batches for the gradient and the Hessian (Martens, 2010; Byrd et al., 2011). Benzing (2022, Sec. I.5) mentions the idea of using *independent* mini-batches for the gradient and the Hessian to obtain an, in some sense, unbiased estimate $-\boldsymbol{H}_{\mathcal{B}}^{-1}\boldsymbol{g}_{\tilde{\mathcal{B}}}$ of the exact Newton step. This does, however, not resolve the biases described in this paper. Via the eigendecomposition of the Hessian $\boldsymbol{H}_{\mathcal{B}} = \sum_{p=1}^P \lambda_p \boldsymbol{u}_p \boldsymbol{u}_p^\top$, we obtain $-\boldsymbol{H}_{\mathcal{B}}^{-1}\boldsymbol{g}_{\tilde{\mathcal{B}}} = -\sum_{p=1}^P \partial_{\boldsymbol{u}_p} q(\boldsymbol{\theta}_\star; \tilde{\mathcal{B}})(\partial^2_{\boldsymbol{u}_p} q(\boldsymbol{\theta}_\star; \mathcal{B}))^{-1}\boldsymbol{u}_p$. While the numerator yields an unbiased estimate of the directional slope (similar to a *blue* dot in the upper

panel of Figure 2), the bias in the denominator remains since eigenvectors and directional curvatures are based on the same mini-batch (as for the *orange* dots in the bottom panel of Figure 2).

**Another debiasing approach.** EKFAC (Eigenvalue-corrected Kronecker Factorization) (George et al., 2018) corrects the eigenvalues of the K-FAC approximation (similar to Equation (9)) which, provably, yields a more accurate approximation of the FIM than K-FAC (in Frobenius norm). This correction, however, is designed to resolve a different kind of bias that is specific to the K-FAC approximation and does not address the biases described in this work.

**Running averages.** Other popular deep learning optimizers *aggregate* curvature information over multiple steps via exponential moving averages (see e.g. (Martens & Grosse, 2015)). This emulates larger mini-batch sizes and thus reduces the curvature biases. However, when curvature evolves rapidly, obsolete curvature estimates from past steps might slow down training. Aggregating more robust *debiased* curvature estimates instead might allow for shorter moving average windows and accelerate the optimization progress. We leave it to future work to explore these interactions.

## 6 EXPERIMENTS

In this section, we evaluate the effectiveness of the debiasing strategies from Section 4.2. In Appendix B, we provide the experimental details as well as additional empirical analyses. For instance, we show how the curvature biases with K-FAC depend on the mini-batch size (see Appendix B.8), how the biases evolve over the course of training (see Appendix B.9), and that the curvature biases become more pronounced for deeper/wider models (see Appendix B.10).

### 6.1 DEBIASED CONJUGATE GRADIENTS

We compare the standard single-batch CG method to the debiased version (see Section 4.2.1) on the fully trained ALL-CNN-C model without curvature damping. For a fair comparison, the single-batch approach uses one mini-batch of size $1024$ while the debiased approach uses two mini-batches of size $512$, such that a similar amount of data and runtime budget is used. Details in Appendix B.5.

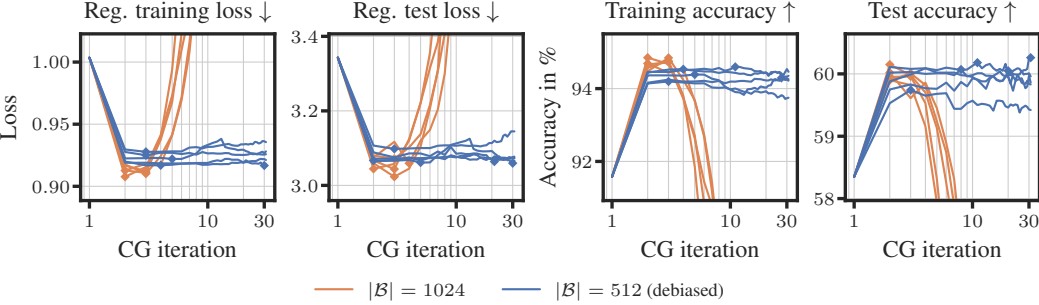

**Figure 5: Debiased CG is much more stable than the single-batch approach.** We compare CG runs without curvature damping ($\delta = 0$) with $K = 30$ iterations for the fully trained ALL-CNN-C model on the CIFAR-100 dataset in terms of training/test loss/accuracy at similar computational cost: The single-batch approach (shown as ▬) uses one mini-batch of size $1024$ while the debiased approach (shown as ▬) uses two mini-batches of size $512$ each. Both approaches use the GGN curvature proxy and are run $5$ times on different mini-batches. The markers ◆ and ◆ are placed at peak performance. While the single-batch runs diverge quickly, the debiased CG runs are stable.

**Results & discussion.** All CG runs in Figure 5 achieve a significant improvement in all four performance metrics. The most striking difference between the two approaches is their stability: The single-batch runs quickly reach peak performance and then diverge. In contrast, although the steps tend to be *larger* (see Figure 4), the debiased CG runs are much more stable (without using any curvature damping). This suggests that the reason for the divergence in the single-batch approach is *not* the missing damping but the misinformed update magnitudes. The peak performance is slightly better for the single-batch runs which can be attributed to its more informative search directions (they were computed using *double* the data). The peak performance of the debiased runs could likely be improved (at the same computational cost) by using a larger batch size for the directions and a smaller one for the update magnitudes (as the former seem harder to estimate, see Section 3.2).

## 6.2 Debiased Laplace approximation

Here, we use a fully trained ALL-CNN-C model on CIFAR-10 data and compare (i) the vanilla model without LA, (ii) the single-batch K-FAC LA approach, (iii) the debiased version (see Section 4.2.2), and (iv) the full-batch approach (where we compute K-FAC on the entire training set) in terms of accuracy, NLL and ECE. Again, we apply the debiased approach at half the batch size of the single-batch approach for a fair comparison. We use prior precisions between $10^{-4}$ and 10. Appendix B.6 contains the experimental details and additional results on the training and OOD data.

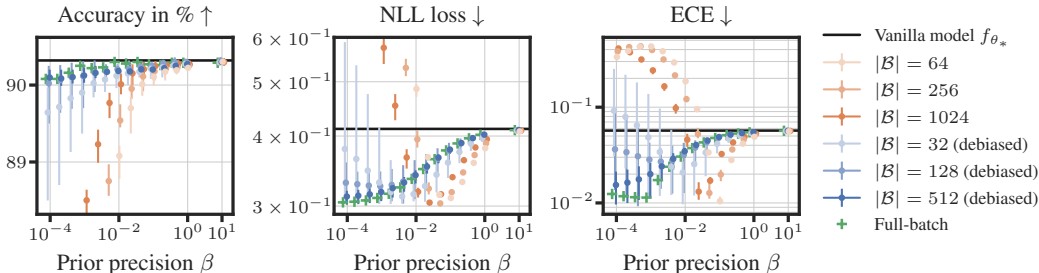

Figure 6: **Debiased LA mimics the full-batch LA very well.** We compare LAs for the fully trained ALL-CNN-C model on CIFAR-10 in terms of accuracy, negative log likelihood loss (NLL) and expected calibration error (ECE) on the CIFAR-10 test set. For each mini-batch size (lighter color indicating smaller batch size), we draw 5 mini-batches and report the mean performance as dot and min/max as vertical line. We also report the performance of the vanilla model without LA (shown as —) and the full-batch approach based on $K_\mathcal{D}$ (shown as +). In contrast to the single-batch approach, the debiased version mimics the behavior of the full-batch approach very well over the entire range of prior precisions.

**Results & discussion.** Figure 6 shows the results. If the prior precision is low (i.e. the LA relies mainly on the curvature information without the regularizing diagonal term), the single-batch version acts erratically due to the deformed curvature model—its performance drops drastically. The single-batch approach is also more sensitive to the choice of prior precision (in fact, it suggests a much larger prior precision than the full-batch approach). In contrast, the debiased approach achieves good performance over a wider range of prior precisions and mimics the full-batch approach ($N = 40,000$) very well despite using only a tiny fraction of the data.

In order to showcase our debiasing strategy's scalability, we provide additional results on RESNET-50 and VIT LITTLE on the IMAGENET dataset in Appendix B.6.

**Summary of the experimental results.** The use of mini-batch quadratics is a simple way to keep the costs of second-order optimization and uncertainty quantification manageable. The resulting biases, however, severely degrade their value, requiring large mini-batches or algorithmic add-ons. Our experiments suggest that even simple debiasing strategies can largely mitigate this issue.

## 7 Conclusion

Our main takeaway is a general principle: **Quadratic approximations to the training loss computed on *mini-batches* of the training data provide a *distorted* representation of the true underlying loss landscape.** In particular, along the directions of large curvature, the mini-batch quadratics tend to strongly overestimate the curvature of the true loss. Our theoretical analysis shows that these biases can be traced back to the misalignment of the curvature matrices' eigenspaces. These insights are highly relevant for applications: As we demonstrated empirically, the biases in the directional slope and curvature lead to severely misinformed updates in stochastic second-order optimizers, and cause unreliable uncertainty estimates with Laplace approximations. We also proposed simple two-batch strategies to mitigate these biases. Our experiments demonstrate their superiority over the single-batch approaches in terms of stability and quality at similar computational costs. Our findings reveal a design prerequisite for building better stochastic curvature-based methods, which should be generally considered, and further developed, for all methods using such curvature evaluations.

ACKNOWLEDGMENTS

The authors gratefully acknowledge co-funding by the European Union (ERC, ANUBIS, 101123955). Views and opinions expressed are however those of the author(s) only and do not necessarily reflect those of the European Union or the European Research Council. Neither the European Union nor the granting authority can be held responsible for them. Philipp Hennig is a member of the Machine Learning Cluster of Excellence, funded by the Deutsche Forschungsgemeinschaft (DFG, German Research Foundation) under Germany's Excellence Strategy - EXC number 2064/1 - Project number 390727645; The authors further gratefully acknowledge financial support by the DFG through Project HE 7114/5-1 in SPP2298/1; the German Federal Ministry of Education and Research (BMBF) through the Tübingen AI Center (FKZ:01IS18039A); and funds from the Ministry of Science, Research and Arts of the State of Baden-Württemberg. Lukas Tatzel and Bálint Mucsányi are grateful to the International Max Planck Research School for Intelligent Systems (IMPRS-IS) for support. Further, the authors thank Felix Dangel and Runa Eschenhagen for providing feedback to the manuscript.

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

SUPPLEMENTARY MATERIAL

Below, we provide additional details on the mathematical derivations, describe the experimental setup and present additional results.

# A MATHEMATICAL DETAILS

## A.1 DIRECTIONAL DERIVATIVES OF A QUADRATIC

We claim in Section 2.1 that a cut $r$ through the quadratic $q(\,\cdot\,; \mathcal{B})$ from $\boldsymbol{\theta}_\bullet \in \mathbb{R}^P$ along the normalized direction $\boldsymbol{d}$ can be written as

$$r(\tau) := q(\boldsymbol{\theta}_\bullet + \tau\boldsymbol{d}; \mathcal{B}) = \frac{1}{2}\tau^2 \boldsymbol{d}^\top \nabla^2 q(\boldsymbol{\theta}_\bullet; \mathcal{B})\,\boldsymbol{d} + \tau \boldsymbol{d}^\top \nabla q(\boldsymbol{\theta}_\bullet; \mathcal{B}) + \text{const.} \tag{10}$$

**Proof for Equation (10).** Here, we provide the derivation for Equation (10). Let $\boldsymbol{\theta}_\bullet \in \mathbb{R}^P$ be a point in parameter space and $\boldsymbol{d} \in \mathbb{R}^P$ a normalized direction, i.e. $\|\boldsymbol{d}\| = 1$. We consider the quadratic approximation $q(\boldsymbol{\theta}; \mathcal{B})$ around $\boldsymbol{\theta}_0$ and evaluate it along the cut $\boldsymbol{\theta}_\bullet + \tau\boldsymbol{d}$ for $\tau \in \mathbb{R}$. We assume that the Hessian (or its approximation) $\boldsymbol{H}_\mathcal{B}$ is symmetric and obtain

$$
\begin{aligned}
r(\tau) &:= q(\boldsymbol{\theta}_\bullet + \tau\boldsymbol{d}; \mathcal{B}) \\
&= \frac{1}{2}(\boldsymbol{\theta}_\bullet + \tau\boldsymbol{d} - \boldsymbol{\theta}_0)^\top \boldsymbol{H}_\mathcal{B}(\boldsymbol{\theta}_\bullet + \tau\boldsymbol{d} - \boldsymbol{\theta}_0) + (\boldsymbol{\theta}_\bullet + \tau\boldsymbol{d} - \boldsymbol{\theta}_0)^\top \boldsymbol{g}_\mathcal{B} + c_\mathcal{B} \\
&= \frac{1}{2}\tau^2 \boldsymbol{d}^\top \boldsymbol{H}_\mathcal{B}\boldsymbol{d} + \tau \boldsymbol{d}^\top \boldsymbol{H}_\mathcal{B}(\boldsymbol{\theta}_\bullet - \boldsymbol{\theta}_0) + \frac{1}{2}(\boldsymbol{\theta}_\bullet - \boldsymbol{\theta}_0)^\top \boldsymbol{H}_\mathcal{B}(\boldsymbol{\theta}_\bullet - \boldsymbol{\theta}_0) \\
&\quad + \tau \boldsymbol{d}^\top \boldsymbol{g}_\mathcal{B} + (\boldsymbol{\theta}_\bullet - \boldsymbol{\theta}_0)^\top \boldsymbol{g}_\mathcal{B} + c_\mathcal{B} \\
&= \frac{1}{2}\tau^2 \boldsymbol{d}^\top \boldsymbol{H}_\mathcal{B}\boldsymbol{d} + \tau \boldsymbol{d}^\top (\boldsymbol{H}_\mathcal{B}(\boldsymbol{\theta}_\bullet - \boldsymbol{\theta}_0) + \boldsymbol{g}_\mathcal{B}) + \text{const.}
\end{aligned}
$$

Since $\nabla q(\boldsymbol{\theta}_\bullet; \mathcal{B}) = \boldsymbol{H}_\mathcal{B}(\boldsymbol{\theta}_\bullet - \boldsymbol{\theta}_0) + \boldsymbol{g}_\mathcal{B}$ and $\nabla^2 q(\boldsymbol{\theta}_\bullet; \mathcal{B}) \equiv \boldsymbol{H}_\mathcal{B}$, we arrive at Equation (10). $\square$

**Proof for Equation (5).** Next, we show that the average directional slope/curvature over all mini-batches in the training set coincides with the directional slope/curvature of the full-batch quadratic. Let's assume that all $M$ mini-batches $\{\mathcal{B}_{m'}\}_{m'=1}^M$ are disjoint, have the same size $|\mathcal{B}_1|$ and that their union is the training set, i.e.

$$|\mathcal{B}_{m'}| = |\mathcal{B}_1| \;\; \forall m' \in \{1, \ldots, M\} \quad \text{and} \quad \bigcup_{m'=1}^M \mathcal{B}_{m'} = \mathcal{D}. \tag{11}$$

This implies $M|\mathcal{B}_1| = |\mathcal{D}|$. It holds

$$
\begin{aligned}
\frac{1}{M}\sum_{m'=1}^M \underbrace{\partial_{\boldsymbol{u}_p} q(\boldsymbol{\theta}_\star; \mathcal{B}_{m'})}_{\text{one} \; \bullet \; \text{and many} \; \bullet} &= \frac{1}{M}\sum_{m'=1}^M \boldsymbol{u}_p^\top \nabla q(\boldsymbol{\theta}_\star; \mathcal{B}_{m'}) \quad \text{with } \nabla q(\boldsymbol{\theta}_\star; \mathcal{B}_{m'}) = \boldsymbol{g}_{\mathcal{B}_{m'}} \text{ since } \boldsymbol{\theta}_0 \leftarrow \boldsymbol{\theta}_\star \\
&= \boldsymbol{u}_p^\top \frac{1}{M}\sum_{m'=1}^M \nabla \mathcal{L}_{\text{reg}}(\boldsymbol{\theta}_\star; \mathcal{B}_{m'}) \\
&= \boldsymbol{u}_p^\top \left( \frac{1}{M}M\nabla r(\boldsymbol{\theta}_\star) + \frac{1}{M}\sum_{m'=1}^M \nabla \mathcal{L}(\boldsymbol{\theta}_\star; \mathcal{B}_{m'}) \right) \\
&= \boldsymbol{u}_p^\top \left( \nabla r(\boldsymbol{\theta}_\star) + \frac{1}{M}\sum_{m'=1}^M \frac{1}{|\mathcal{B}_{m'}|}\sum_{n\in\mathcal{B}_{m'}} \nabla \ell(f_{\boldsymbol{\theta}_\star}(\boldsymbol{x}_n), \boldsymbol{y}_n) \right) \\
&\overset{(11)}{=} \boldsymbol{u}_p^\top \left( \nabla r(\boldsymbol{\theta}_\star) + \frac{1}{M|\mathcal{B}_1|}\sum_{m'=1}^M \sum_{n\in\mathcal{B}_{m'}} \nabla \ell(f_{\boldsymbol{\theta}_\star}(\boldsymbol{x}_n), \boldsymbol{y}_n) \right) \\
&= \boldsymbol{u}_p^\top \nabla \mathcal{L}_{\text{reg}}(\boldsymbol{\theta}_\star; \mathcal{D}) \\
&= \underbrace{\partial_{\boldsymbol{u}_p} q(\boldsymbol{\theta}_\star; \mathcal{D})}_{\text{\textcolor{green}{+}}}.
\end{aligned}
$$

As the mini-batch curvature $\nabla^2 \mathcal{L}(\boldsymbol{\theta}_\star; \mathcal{B}_{m'})$ is also an *average* over the samples in the mini-batch (this applies to both the actual Hessian and its GGN approximation), the same argument holds for the associated directional curvatures. For K-FAC, the derivation above does *not* hold as $(1/2)(\boldsymbol{K}_\mathcal{B} + \boldsymbol{K}_{\tilde{\mathcal{B}}}) \neq \boldsymbol{K}_{\mathcal{B}\cup\tilde{\mathcal{B}}}$. $\square$

## A.2 THE METHOD OF CONJUGATE GRADIENTS (CG)

**Using CG for minimizing a quadratic.** The argmin of a quadratic $q(\,\cdot\,; \mathcal{B})$ is given by $\boldsymbol{\theta}_0 - \boldsymbol{H}_\mathcal{B}^{-1}\boldsymbol{g}_\mathcal{B}$ (assuming that $\boldsymbol{H}_\mathcal{B}$ is symmetric and positive definite), see Section 2.2. We can use the method of conjugate gradients (see Algorithm 1) for computing the Newton step (or an approximation thereof) by setting $\boldsymbol{A} \leftarrow \boldsymbol{H}_\mathcal{B}$ and $\boldsymbol{b} \leftarrow -\boldsymbol{g}_\mathcal{B}$.

---

**Algorithm 1:** Method of conjugate gradients (CG), based on (Nocedal & Wright, 2006, Alg. 5.2)

**Input.** Access to matrix-vector products $\boldsymbol{v} \mapsto \boldsymbol{Av}$, where $\boldsymbol{A} \in \mathbb{R}^{P \times P}$ is symmetric and positive definite, right-hand side $\boldsymbol{b} \in \mathbb{R}^P$, convergence tolerance $\epsilon \in \mathbb{R}_{>0}$, maximum number of iterations $P_{\max} \in \mathbb{N}$, $P_{\max} \leqslant P$

**Output.** Approximate solution $\boldsymbol{x}_p$ to the linear system $\boldsymbol{Ax} = \boldsymbol{b}$

1   Initialize $\boldsymbol{x}_0 = \boldsymbol{0}$, $\boldsymbol{r}_0 \leftarrow -\boldsymbol{b}$ and $\boldsymbol{s}_0 \leftarrow -\boldsymbol{r}_0$
2   **for** $p = 0, 1, \ldots, P$ **do**
3      **if** $p = P_{\max}$ **or** $\|\boldsymbol{r}_p\|_2 \leqslant \epsilon$ **then**              Termination criteria
4          **return** (approximate) solution $\boldsymbol{x}_p$

5      $\boldsymbol{t}_p \leftarrow \boldsymbol{As}_p$              Compute the matrix-vector product only once
6      $\alpha_p \leftarrow \dfrac{\boldsymbol{r}_p^\top \boldsymbol{r}_p}{\boldsymbol{s}_p^\top \boldsymbol{t}_p}$              Update magnitude $\alpha_p = \dfrac{\boldsymbol{r}_p^\top \boldsymbol{r}_p}{\boldsymbol{s}_p^\top \boldsymbol{As}_p}$
7      $\boldsymbol{x}_{p+1} \leftarrow \boldsymbol{x}_p + \alpha_p \boldsymbol{s}_p$              Update along search direction $\boldsymbol{s}_p$
8      $\boldsymbol{r}_{p+1} \leftarrow \boldsymbol{r}_p + \alpha_p \boldsymbol{t}_p$              Residual $\boldsymbol{r}_p = \boldsymbol{Ax}_p - \boldsymbol{b}$ computed via recursion
9      $\beta_{p+1} \leftarrow \dfrac{\boldsymbol{r}_{p+1}^\top \boldsymbol{r}_{p+1}}{\boldsymbol{r}_p^\top \boldsymbol{r}_p}$
10     $\boldsymbol{s}_{p+1} \leftarrow -\boldsymbol{r}_{p+1} + \beta_{p+1}\boldsymbol{s}_p$              Construction of new (conjugate) search direction

---

**Properties of CG & geometric interpretations.** In the following, we provide some important properties and a geometric interpretation of the quantities involved in the CG method.

- **Residual $\boldsymbol{r}_p$.** Let $\boldsymbol{x}_p := \boldsymbol{\theta}_p - \boldsymbol{\theta}_0$. The residual $\boldsymbol{r}_p$ can be written as $\boldsymbol{r}_p = \boldsymbol{Ax}_p - \boldsymbol{b} = \boldsymbol{H}_\mathcal{B}(\boldsymbol{\theta}_p - \boldsymbol{\theta}_0) + \boldsymbol{g}_\mathcal{B} = \nabla q(\boldsymbol{\theta}_p; \mathcal{B})$, i.e. it coincides with the gradient of the quadratic at $\boldsymbol{\theta}_p$.

- **Update magnitude $\alpha_p$.** Consider a cut through the quadratic from $\boldsymbol{\theta}_p$ into the direction $\boldsymbol{s}_p$, i.e. $r(\tau) = q(\boldsymbol{\theta}_p + \tau\boldsymbol{s}_p; \mathcal{B})$. Minimizing this 1D quadratic requires its first derivative to vanish. It holds (see Equation (10))

$$r'(\tau) = 0 \quad \Leftrightarrow \quad \tau \boldsymbol{s}_p^\top \nabla^2 q(\boldsymbol{\theta}_p; \mathcal{B})\, \boldsymbol{s}_p + \boldsymbol{s}_p^\top \nabla q(\boldsymbol{\theta}_p; \mathcal{B}) = 0$$

$$\Leftrightarrow \quad \tau = -\frac{\boldsymbol{s}_p^\top \nabla q(\boldsymbol{\theta}_p; \mathcal{B})}{\boldsymbol{s}_p^\top \nabla^2 q(\boldsymbol{\theta}_p; \mathcal{B})\, \boldsymbol{d}_p} = -\frac{\boldsymbol{s}_p^\top \boldsymbol{r}_p}{\boldsymbol{s}_p^\top \boldsymbol{A}\, \boldsymbol{s}_p} = \frac{\boldsymbol{r}_p^\top \boldsymbol{r}_p}{\boldsymbol{s}_p^\top \boldsymbol{As}_p} = \alpha_p,$$

where the equality $-\boldsymbol{s}_p^\top \boldsymbol{r}_p = \boldsymbol{r}_p^\top \boldsymbol{r}_p$ is due to Equation (5.14a) and Theorem 5.2 in (Nocedal & Wright, 2006). So, the update magnitude $\alpha_p$ is chosen such that it minimizes the quadratic along the direction $\boldsymbol{s}_p$. Note that the update magnitude can also be written as a ratio of the negative directional slope and directional curvature at $\boldsymbol{\theta}_p$, i.e. a 1D directional Newton step. Let $\boldsymbol{d}_p := \boldsymbol{s}_p / \|\boldsymbol{s}_p\|$ denote the *normalized* search direction. It holds

$$\boldsymbol{x}_{p+1} = \boldsymbol{x}_p + \alpha_p \boldsymbol{s}_p = \boldsymbol{x}_p - \frac{\boldsymbol{s}_p^\top \boldsymbol{r}_p}{\boldsymbol{s}_p^\top \boldsymbol{A}\, \boldsymbol{s}_p}\boldsymbol{s}_p = \boldsymbol{x}_p - \frac{\boldsymbol{d}_p^\top \boldsymbol{r}_p}{\boldsymbol{d}_p^\top \boldsymbol{A}\, \boldsymbol{d}_p}\boldsymbol{d}_p = \boldsymbol{x}_p - \underbrace{\frac{\partial_{\boldsymbol{d}_p} q(\boldsymbol{\theta}_d; \mathcal{B})}{\partial_{\boldsymbol{d}_p}^2 q(\boldsymbol{\theta}_d; \mathcal{B})}}_{=: \tau_p} \boldsymbol{d}_p$$

i.e. $\tau_p = \alpha_p \|\boldsymbol{s}_p\|$. Via the shift $\boldsymbol{\theta}_p = \boldsymbol{\theta}_0 + \boldsymbol{x}_p$, we arrive at Equation (3).

- **Search direction $\boldsymbol{s}_p$.** CG constructs the search directions to be conjugate, i.e. $\boldsymbol{s}_{p+1}^\top \boldsymbol{A}\, \boldsymbol{s}_i = 0$ for all $i \leqslant p$. Note that this property also applies to the normalized search directions.

**Efficient implementation of the debiased CG approach.** For the debiased CG version, we need to re-evaluate the update magnitudes $\tilde{\tau}_p$ for a given set of search directions $\{d_1, \ldots d_P\}$ on a second mini-batch $\tilde{\mathcal{B}}$ (see Equation (8)), i.e.

$$\tilde{\tau}_p = -\frac{\partial_{d_p} q(\tilde{\theta}_p; \tilde{\mathcal{B}})}{\partial_{d_p}^2 q(\tilde{\theta}_p; \tilde{\mathcal{B}})} = -\frac{d_p^\top \nabla q(\tilde{\theta}_p; \tilde{\mathcal{B}})}{d_p^\top \nabla^2 q(\tilde{\theta}_p; \tilde{\mathcal{B}}) \, d_p} = -\frac{d_p^\top (H_{\tilde{\mathcal{B}}}(\tilde{\theta}_p - \theta_0) + g_{\tilde{\mathcal{B}}})}{d_p^\top H_{\tilde{\mathcal{B}}} \, d_p}$$

Implemented naively, this requires *two* matrix-vector products—one for the numerator and one for the denominator. However, we can use a recursive formula for the numerator:

$$
\begin{aligned}
\nabla q(\tilde{\theta}_p; \tilde{\mathcal{B}}) &= H_{\tilde{\mathcal{B}}}(\tilde{\theta}_p - \theta_0) + g_{\tilde{\mathcal{B}}} \\
&= H_{\tilde{\mathcal{B}}}(\tilde{\theta}_{p-1} + \tilde{\tau}_{p-1} d_{p-1} - \theta_0) + g_{\tilde{\mathcal{B}}} \\
&= H_{\tilde{\mathcal{B}}}(\tilde{\theta}_{p-1} - \theta_0) + g_{\tilde{\mathcal{B}}} + \tilde{\tau}_{p-1} H_{\tilde{\mathcal{B}}} d_{p-1} \\
&= \nabla q(\tilde{\theta}_{p-1}; \tilde{\mathcal{B}}) + \tilde{\tau}_{p-1} H_{\tilde{\mathcal{B}}} d_{p-1}.
\end{aligned}
\tag{12}
$$

So, if we store the gradient from the previous iteration $\nabla q(\tilde{\theta}_{p-1}; \tilde{\mathcal{B}})$ and the Hessian vector product with the previous direction $H_{\tilde{\mathcal{B}}} d_{p-1}$, the current gradient can be computed *without* an additional matrix-vector product. At iteration $p$, we thus (i) compute the gradient $\nabla q(\tilde{\theta}_p; \tilde{\mathcal{B}})$ recursively from the cached vectors $\nabla q(\tilde{\theta}_{p-1}; \tilde{\mathcal{B}})$ and $H_{\tilde{\mathcal{B}}} d_{p-1}$ via Equation (12) (for $p = 0$, we have $\nabla q(\tilde{\theta}_0; \tilde{\mathcal{B}}) = g_{\tilde{\mathcal{B}}}$), (ii) compute the Hessian-vector product with the current direction $H_{\tilde{\mathcal{B}}} d_p$, (iii) store both these vectors and (iv) compute the update magnitude $\tilde{\tau}_p$ (both the numerator and the denominator only require a simple dot product of two pre-computed vectors).

### A.3 LAPLACE APPROXIMATION FOR NEURAL NETWORKS

#### A.3.1 DERIVATION OF THE LAPLACE APPROXIMATION FOR NEURAL NETWORKS

**Preliminaries.** The softmax function $\text{softmax} : \mathbb{R}^C \to \mathbb{R}^C$ is defined as

$$\text{softmax}(z) := \left( \frac{\exp(z_1)}{\sum_{c'=1}^C \exp(z_{c'})}, \ldots, \frac{\exp(z_C)}{\sum_{c'=1}^C \exp(z_{c'})} \right).$$

It maps an arbitrary vector $z \in \mathbb{R}^C$ to a vector whose entries are non-negative and sum up to one. So, the output of the softmax function can be interpreted as a probability distribution over $C$ classes. With this, the cross-entropy loss for a single datum $(x, y)$ is given by

$$\ell(f_\theta(x), y) := -\sum_{c=1}^C y_c \cdot \log(\text{softmax}(f_\theta(x))_c). \tag{13}$$

Here, we assume that $y \in \{0, 1\}^C$ is a one-hot encoded vector representing the true class label. So, if $c_\star \in \{1, \ldots, C\}$ is the correct class (i.e. $y_{c_\star} = 1$), the cross-entropy loss is given by the negative logarithm of the probability the network assigns to this class $\ell(f_\theta(x), y) = -\log(\text{softmax}(f_\theta(x))_{c_\star})$. Finally, let

$$\text{Cat}(y; p) := \prod_{c=1}^C p_c^{y_c} \tag{14}$$

denote the probability mass function of the categorical distribution, where $p \in \mathbb{R}^C$ is a vector of probabilities (i.e. $p_c \geqslant 0$ and $\sum_c p_c = 1$), and $y \in \{0, 1\}^C$ is a one-hot encoded vector representing a class label.

**Probabilistic interpretation of the regularized loss.** Recall from Equation (1) that the regularized loss function is given by

$$\mathcal{L}_{\text{reg}}(\theta; \mathcal{D}) = \mathcal{L}(\theta; \mathcal{D}) + r(\theta) \quad \text{with} \quad \mathcal{L}(\theta; \mathcal{D}) = \frac{1}{N} \sum_{n \in \mathcal{D}} \ell(f_\theta(x_n), y_n).$$

We assume a classification problem that uses the cross-entropy loss (see Equation (13)) and an $\ell^2$ regularizer $r(\theta) = \beta/2 \|\theta\|_2^2$ with parameter $\beta \in \mathbb{R}_{>0}$. We use one-hot encoded labels $y_n \in \{0, 1\}^C$ (with $y_{nc}$, we denote the $c$-th entry of $y_n$) and obtain

$$N \cdot \mathcal{L}_{\text{reg}}(\theta; \mathcal{D}) = \sum_{n \in \mathcal{D}} \ell(f_\theta(x_n), y_n) + \frac{N\beta}{2} \|\theta\|_2^2$$

$$\overset{(13)}{=} -\sum_{n \in \mathcal{D}} \sum_{c=1}^{C} \boldsymbol{y}_{nc} \cdot \log(\mathrm{softmax}(f_{\boldsymbol{\theta}}(\boldsymbol{x}_n))_c) + \frac{N\beta}{2} \|\boldsymbol{\theta}\|_2^2$$

$$\overset{(14)}{=} -\sum_{n \in \mathcal{D}} \log\left(\mathrm{Cat}(\boldsymbol{y}_n; \mathrm{softmax}(f_{\boldsymbol{\theta}}(\boldsymbol{x}_n)))\right) - \left(-\frac{1}{2}\boldsymbol{\theta}^\top (N\beta \cdot \boldsymbol{I})\,\boldsymbol{\theta}\right)$$

$$= -\log\left(\underbrace{\prod_{n \in \mathcal{D}} \mathrm{Cat}(\boldsymbol{y}_n; \mathrm{softmax}(f_{\boldsymbol{\theta}}(\boldsymbol{x}_n)))}_{\text{likelihood } p(\mathbb{D} \mid \boldsymbol{\theta})}\right) - \log\left(\underbrace{\mathcal{N}\left(\theta; \boldsymbol{0}, \frac{1}{N\beta}\boldsymbol{I}\right)}_{\text{prior } p(\boldsymbol{\theta})}\right) - Z,$$

where $Z := {}^{P}\!/\!_{2}\log(2\pi/N\beta)$ absorbs the normalization constant of the Gaussian prior. The cross entropy loss $\mathcal{L}$ for the training set is thus connected to the negative log categorical likelihood and the regularizer can be seen as a negative log Gaussian prior over the parameters. Note that a similar derivation is possible for other loss functions as well, e.g. the MSE loss (which is equivalent to a negative log Gaussian likelihood).

The derivation above shows that the (rescaled) regularized loss function can be interpreted as the negative unnormalized log posterior of a Bayesian model

$$N \cdot \mathcal{L}_{\mathrm{reg}}(\boldsymbol{\theta}; \mathcal{D}) \overset{\mathrm{c}}{=} -\log p(\mathbb{D} \mid \boldsymbol{\theta}) - \log p(\boldsymbol{\theta}) \overset{\mathrm{c}}{=} -\log p(\boldsymbol{\theta} \mid \mathbb{D}) \tag{15}$$

with Gaussian prior and categorical likelihood. With $\overset{\mathrm{c}}{=}$, we denote equality up to an additive constant.

**Training as MAP estimation.** Equation (15) allows re-interpreting the training procedure of a neural network as a maximum a posteriori (MAP) estimation problem since minimizing the regularized loss

$$\underbrace{\underset{\boldsymbol{\theta} \in \mathbb{R}^P}{\arg\min} \mathcal{L}_{\mathrm{reg}}(\boldsymbol{\theta}, \mathcal{D})}_{= \boldsymbol{\theta}_\star} = \underset{\boldsymbol{\theta} \in \mathbb{R}^P}{\arg\min} N \cdot \mathcal{L}_{\mathrm{reg}}(\boldsymbol{\theta}, \mathcal{D}) \overset{(15)}{=} \underset{\boldsymbol{\theta} \in \mathbb{R}^P}{\arg\max} \log p(\boldsymbol{\theta} \mid \mathbb{D}) = \underbrace{\underset{\boldsymbol{\theta} \in \mathbb{R}^P}{\arg\max} p(\boldsymbol{\theta} \mid \mathbb{D})}_{=: \boldsymbol{\theta}_\star}$$

is equivalent to maximizing the posterior $p(\boldsymbol{\theta} \mid \mathbb{D})$.

**Laplace approximation.** The idea of the Laplace approximation is to approximate the posterior distribution $p(\boldsymbol{\theta} \mid \mathbb{D}) \approx \mathcal{N}(\boldsymbol{\theta}; \boldsymbol{\theta}_\star, \boldsymbol{\Sigma}_{\mathcal{D}})$ with a Gaussian at the mode $\boldsymbol{\theta}_\star$ of the posterior, i.e. after training the network. For this, we approximate the log posterior by a second-order Taylor expansion around the mode $\boldsymbol{\theta}_0 \leftarrow \boldsymbol{\theta}_\star$, i.e.

$$\log p(\boldsymbol{\theta} \mid \mathbb{D}) \overset{\mathrm{c}}{=} -N \cdot \mathcal{L}_{\mathrm{reg}}(\boldsymbol{\theta}; \mathcal{D}) \overset{(2)}{\approx} -N \cdot q(\boldsymbol{\theta}; \mathcal{D}) \overset{\mathrm{c}}{=} -\frac{1}{2}(\boldsymbol{\theta} - \boldsymbol{\theta}_\star)^\top (N \cdot \boldsymbol{H}_{\mathcal{D}})(\boldsymbol{\theta} - \boldsymbol{\theta}_\star), \tag{16}$$

where we assumed that $\boldsymbol{\theta}_\star$ is a (local) minimum of the regularized loss function (i.e. $\boldsymbol{g}_{\mathcal{D}} = \nabla\mathcal{L}_{\mathrm{reg}}(\boldsymbol{\theta}_\star; \mathcal{D}) = 0$) such that the linear term $(\boldsymbol{\theta} - \boldsymbol{\theta}_0)^\top \boldsymbol{g}_{\mathcal{D}}$ in the quadratic approximation vanishes. The additive constants we did not state explicitly in Equation (16) turn into some multiplicative factor (denoted by $Z^{-1}$) when taking the exponential

$$p(\boldsymbol{\theta} \mid \mathbb{D}) \approx \frac{1}{Z}\exp\left(-\frac{1}{2}(\boldsymbol{\theta} - \boldsymbol{\theta}_\star)^\top (N \cdot \boldsymbol{H}_{\mathcal{D}})(\boldsymbol{\theta} - \boldsymbol{\theta}_\star)\right).$$

This immediately identifies $Z$ as the normalization constant of the Gaussian and we obtain the Laplace approximation

$$p(\boldsymbol{\theta} \mid \mathbb{D}) \approx \mathcal{N}(\boldsymbol{\theta}; \boldsymbol{\theta}_\star, \boldsymbol{\Sigma}_{\mathcal{D}}) \quad \text{with} \quad \boldsymbol{\Sigma}_{\mathcal{D}} := (N \cdot \boldsymbol{H}_{\mathcal{D}})^{-1} = \frac{1}{N}\boldsymbol{H}_{\mathcal{D}}^{-1}. \tag{17}$$

**Mini-batch version of the Laplace approximation.** In order to reduce the computational cost of the Laplace approximation, we can replace the full-batch quadratic $q(\boldsymbol{\theta}; \mathcal{D})$ in Equation (16) by a mini-batch quadratic $q(\boldsymbol{\theta}; \mathcal{B})$, i.e. we obtain $\boldsymbol{\Sigma}_{\mathcal{B}} = N^{-1}\boldsymbol{H}_{\mathcal{B}}^{-1}$.

**Predictive uncertainty via the Laplace approximation.** Ultimately, we want to use the Laplace approximation to equip the prediction for an unknown test input $\boldsymbol{x}_\diamond$ with uncertainty. Ideally, we would compute the expectation of the model likelihood under the approximate posterior, i.e.

$$p(\boldsymbol{y}_\diamond \mid \boldsymbol{x}_\diamond, \mathbb{D}) = \int p(\boldsymbol{y}_\diamond \mid \boldsymbol{x}_\diamond, \boldsymbol{\theta})\, p(\boldsymbol{\theta} \mid \mathbb{D})\, d\boldsymbol{\theta}$$

$$\overset{(17)}{\approx} \int \mathrm{Cat}(\boldsymbol{y}_\diamond; \mathrm{softmax}(f_{\boldsymbol{\theta}}(\boldsymbol{x}_\diamond)))\, \mathcal{N}(\boldsymbol{\theta}; \boldsymbol{\theta}_\star, \boldsymbol{\Sigma}_\mathcal{B})\, d\boldsymbol{\theta}.$$

One way to approximate this integral is via Monte Carlo sampling, i.e.

$$p(\boldsymbol{y}_\diamond \mid \boldsymbol{x}_\diamond, \mathbb{D}) \approx \frac{1}{S} \sum_{s=1}^{S} \mathrm{Cat}\big(\boldsymbol{y}_\diamond; \mathrm{softmax}(f_{\boldsymbol{\theta}^{(s)}}^{\mathrm{lin}}(\boldsymbol{x}_\diamond))\big) \quad \text{where} \quad \boldsymbol{\theta}^{(s)} \sim \mathcal{N}(\boldsymbol{\theta}_\star, \boldsymbol{\Sigma}_\mathcal{B}). \tag{18}$$

As suggested by Immer et al. (2021); Roy et al. (2024), we use the linearized network $f_{\boldsymbol{\theta}}^{\mathrm{lin}}(\boldsymbol{x}) :=$ $f_{\boldsymbol{\theta}_\star}(\boldsymbol{x}) + \nabla f_{\boldsymbol{\theta}_\star}(\boldsymbol{x})(\boldsymbol{\theta} - \boldsymbol{\theta}_\star)$, where $\nabla f_{\boldsymbol{\theta}_\star}(\boldsymbol{x}) \in \mathbb{R}^{C \times P}$ is the network's Jacobian at $\boldsymbol{\theta}_\star$.

### A.3.2 SAMPLING FROM THE K-FAC LAPLACE APPROXIMATION

The Monte Carlo (MC) approach from Equation (18) requires samples from the Gaussian $\mathcal{N}(\boldsymbol{\theta}_\star, \boldsymbol{\Sigma}_\mathcal{B})$ (the following derivations work exactly the same when the full-batch LA based on the entire training set is used). However, $\boldsymbol{\Sigma}_\mathcal{B} \in \mathbb{R}^{P \times P}$ is often too large to be built explicitly in memory and it requires inverting the $P \times P$ Hessian $\boldsymbol{H}_\mathcal{B}$ of the regularized loss function. One approach for that issue is to use the K-FAC curvature approximation $\boldsymbol{H}_\mathcal{B} = \nabla^2 \mathcal{L}(\boldsymbol{\theta}; \mathcal{B}) + \beta \boldsymbol{I} \approx \boldsymbol{K}_\mathcal{B} + \beta \boldsymbol{I}$, i.e. $\boldsymbol{\Sigma}_\mathcal{B} \approx N^{-1}(\boldsymbol{K}_\mathcal{B} + \beta \boldsymbol{I})^{-1}$. As we will see in the following, the K-FAC approximation enables us to sample efficiently from the corresponding LA.

**Leveraging K-FAC's block-diagonal structure.** K-FAC is a block-diagonal curvature approximation, i.e. $\boldsymbol{K}_\mathcal{B} = \mathrm{blockdiag}_{l=1,\dots,L}(\boldsymbol{K}_\mathcal{B}^{(l)})$ and $\boldsymbol{K}_\mathcal{B} + \beta \boldsymbol{I}$ inherits this structure. The inverse of a block-diagonal matrix is also block-diagonal with the block inverses on the diagonal, i.e.

$$\begin{aligned}
\boldsymbol{\Sigma}_\mathcal{B} &\approx \frac{1}{N}(\boldsymbol{K}_\mathcal{B} + \beta \boldsymbol{I})^{-1} \\
&= \frac{1}{N}\big(\mathrm{blockdiag}_{l=1,\dots,L}(\boldsymbol{K}_\mathcal{B}^{(l)} + \beta \boldsymbol{I})\big)^{-1} \\
&= \mathrm{blockdiag}_{l=1,\dots,L}\big(\underbrace{\frac{1}{N}(\boldsymbol{K}_\mathcal{B}^{(l)} + \beta \boldsymbol{I})^{-1}}_{=:\,\boldsymbol{\Sigma}^{(l)}}\big) \\
&= \mathrm{blockdiag}_{l=1,\dots,L}(\boldsymbol{\Sigma}^{(l)}).
\end{aligned}$$

To draw a sample $\boldsymbol{v} \in \mathbb{R}^P$ from $\mathcal{N}(\boldsymbol{\theta}_\star, \boldsymbol{\Sigma}_\mathcal{B})$, we can thus simply sample from the block covariances $\boldsymbol{v}^{(l)} \sim \mathcal{N}(\boldsymbol{0}, \boldsymbol{\Sigma}^{(l)})$, stack these samples and add the mean, i.e. $\boldsymbol{v} = \boldsymbol{\theta}_\star + (\boldsymbol{v}^{(1)}, \dots, \boldsymbol{v}^{(L)})$.

**Leveraging the blocks' Kronecker structure.** To sample from $\mathcal{N}(\boldsymbol{0}, \boldsymbol{\Sigma}^{(l)})$, we can exploit the Kronecker structure of the blocks $\boldsymbol{K}_\mathcal{B}^{(l)} = \boldsymbol{A} \otimes \boldsymbol{B}$ (we omit the layer index $l$ for $\boldsymbol{A}$ and $\boldsymbol{B}$ for brevity). First, we compute the eigendecompositions of the Kronecker factors, i.e. $\boldsymbol{A} = \boldsymbol{U}_A \boldsymbol{S}_A \boldsymbol{U}_A^\top$ and $\boldsymbol{B} = \boldsymbol{U}_B \boldsymbol{S}_B \boldsymbol{U}_B^\top$. It follows

$$\begin{aligned}
\boldsymbol{K}_\mathcal{B}^{(l)} + \beta \boldsymbol{I} &= \boldsymbol{A} \otimes \boldsymbol{B} + \beta \boldsymbol{I} \\
&= \boldsymbol{U}_A \boldsymbol{S}_A \boldsymbol{U}_A^\top \otimes \boldsymbol{U}_B \boldsymbol{S}_B \boldsymbol{U}_B^\top + \beta \boldsymbol{I} \\
&= \underbrace{(\boldsymbol{U}_A \otimes \boldsymbol{U}_B)}_{=:\,\boldsymbol{U}} \underbrace{(\boldsymbol{S}_A \otimes \boldsymbol{S}_B)}_{=:\,\boldsymbol{S}} (\boldsymbol{U}_A \otimes \boldsymbol{U}_B)^\top + \beta \boldsymbol{I} \\
&= \boldsymbol{U} \boldsymbol{S} \boldsymbol{U}^\top + \beta \boldsymbol{I} \\
&= \boldsymbol{U}(\boldsymbol{S} + \beta \boldsymbol{I})\boldsymbol{U}^\top.
\end{aligned}$$

$\boldsymbol{U} = \boldsymbol{U}_A \otimes \boldsymbol{U}_B$ forms an orthogonal eigenbasis of the block and the diagonal matrix $\boldsymbol{S} + \beta \boldsymbol{I} = \boldsymbol{S}_A \otimes \boldsymbol{S}_B + \beta \boldsymbol{I}$ contains the corresponding eigenvalues (see e.g. (George et al., 2018)). It follows

$$\begin{aligned}
\boldsymbol{\Sigma}^{(l)} &= \frac{1}{N}(\boldsymbol{K}_\mathcal{B}^{(l)} + \beta \boldsymbol{I})^{-1} \\
&= \frac{1}{N} \boldsymbol{U}(\boldsymbol{S} + \beta \boldsymbol{I})^{-1} \boldsymbol{U}^\top \\
&= \underbrace{\frac{1}{\sqrt{N}} \boldsymbol{U}(\boldsymbol{S} + \beta \boldsymbol{I})^{-1/2}}_{=:\,\boldsymbol{V}} \frac{1}{\sqrt{N}}(\boldsymbol{S} + \beta \boldsymbol{I})^{-1/2} \boldsymbol{U}^\top
\end{aligned}$$

$$= VV^\top.$$

So, in order to draw a sample $v^{(l)} \sim \mathcal{N}(\mathbf{0}, \Sigma^{(l)})$, we first draw from a standard Gaussian $w \sim \mathcal{N}(\mathbf{0}, I)$ and then transform the sample via $v^{(l)} = Vw$. The resulting vector $v^{(l)}$ has mean $\mathbf{0}$ and covariance $VIV^\top = \Sigma^{(l)}$. As Gaussians are closed under affine linear transformations, $v^{(l)}$ is indeed distributed according to $\mathcal{N}(\mathbf{0}, \Sigma^{(l)})$.

The matrix-vector product $w \mapsto Vw$ can be computed efficiently without actually forming $V$ in memory. The first step is to multiply with $(S + \beta I)^{-1/2}$. Since this is a diagonal matrix, we can simply multiply the vector $w$ element-wise with the inverse square root of the diagonal entries of $S + \beta I$. The second step is to multiply with $U = U_A \otimes U_B$, which can be implemented efficiently by using the property $(U_A \otimes U_B) \operatorname{vec}(W) = \operatorname{vec}(U_B W U_A^\top)$. Finally, we scale by $N^{-1/2}$.

**Summary.** Drawing a sample from the K-FAC LA $\mathcal{N}(\theta_\star, N^{-1}(K_\mathcal{B} + \beta I)^{-1})$ can be done without ever forming the blocks of the covariance matrix explicitly. Using properties of the Kronecker product, we can efficiently transform a sample from the standard Gaussian to a sample $v^{(l)} \sim \mathcal{N}(\mathbf{0}, \Sigma^{(l)})$. Stacking the samples from all blocks and adding the mean $\theta_\star$ yields a sample from the LA due to the block-diagonal structure of the covariance matrix.

### A.3.3 DEBIASED K-FAC LAPLACE APPROXIMATION

The idea of the debiased K-FAC Laplace approximation is to construct one Kronecker-factored curvature matrix from two mini-batches $\mathcal{B}$ and $\tilde{\mathcal{B}}$.

**Eigenbasis of a K-FAC block.** First, recall that the eigendecomposition of a block $K_\mathcal{B}^{(l)}$ from a K-FAC matrix can be constructed from the eigendecompositions of the Kronecker factors (see e.g. (George et al., 2018)). Consider the $l$-th block from both K-FAC approximations $K_\mathcal{B}^{(l)} = A \otimes B$ and $K_{\tilde{\mathcal{B}}}^{(l)} = C \otimes D$ (we omit the layer index $l$ for $A$, $B$, $C$, and $D$ for brevity). Again, let $A = U_A S_A U_A^\top$ and $B = U_B S_B U_B^\top$ denote the eigendecompositions of the Kronecker factors. It holds

$$
\begin{aligned}
K_\mathcal{B}^{(l)} &= A \otimes B \\
&= U_A S_A U_A^\top \otimes U_B S_B U_B^\top \\
&= \underbrace{(U_A \otimes U_B)}_{=: U} \underbrace{(S_A \otimes S_B)}_{=: S} (U_A \otimes U_B)^\top \\
&= U S U^\top
\end{aligned}
$$

$U = U_A \otimes U_B$ forms an orthogonal eigenbasis of the block and the diagonal matrix $S = S_A \otimes S_B$ contains the eigenvalues.

**Re-evaluation of the directional curvatures.** For the debiased approach, we keep the block's eigenbasis $U$, but instead of using the directional curvatures $S$, we re-evaluate these measurements on the second mini-batch $\tilde{\mathcal{B}}$. First, consider the projection of the block $K_{\tilde{\mathcal{B}}}^{(l)}$ onto the eigenvectors $U$, i.e.

$$U^\top K_{\tilde{\mathcal{B}}}^{(l)} U = (U_A \otimes U_B)^\top (C \otimes D)(U_A \otimes U_B) = U_A^\top C U_A \otimes U_B^\top D U_B.$$

The debiased directional curvatures are on the diagonal of $U^\top K_{\tilde{\mathcal{B}}}^{(l)} U$. For a square matrix $X \in \mathbb{R}^{n \times n}$, let $\operatorname{Diag}(X)$ denote the operator that maps the matrix onto its diagonal, i.e. $\operatorname{Diag} : \mathbb{R}^{n \times n} \to \mathbb{R}^n$ with $\operatorname{Diag}(X)_i := X_{ii}$ for $i \in \{1, \dots, n\}$. It holds:

$$
\begin{aligned}
\operatorname{Diag}(U^\top K_{\tilde{\mathcal{B}}}^{(l)} U) &= \operatorname{Diag}(U_A^\top C U_A \otimes U_B^\top D U_B) \\
&= \underbrace{\operatorname{Diag}(U_A^\top C U_A)}_{=: \tilde{s}_A} \otimes \underbrace{\operatorname{Diag}(U_B^\top D U_B)}_{=: \tilde{s}_B} \\
&= \tilde{s}_A \otimes \tilde{s}_B \\
&=: \tilde{s}.
\end{aligned}
$$

**Construction of a debiased block.** Now that we have the eigenbasis $U$ and the debiased directional curvatures $\tilde{s}$, we construct the debiased block. Let $\tilde{S}_A := \mathrm{diag}(\tilde{s}_A)$, $\tilde{S}_B := \mathrm{diag}(\tilde{s}_B)$ and

$$\tilde{S} := \mathrm{diag}(\tilde{s}) = \mathrm{diag}(\tilde{s}_A \otimes \tilde{s}_B) = \mathrm{diag}(\tilde{s}_A) \otimes \mathrm{diag}(\tilde{s}_B) = \tilde{S}_A \otimes \tilde{S}_B.$$

The debiased block is given by $U\tilde{S}U^\top$. It can be written as the Kronecker product of two matrices $\tilde{A}$ and $\tilde{B}$ since

$$U\tilde{S}U^\top = (U_A \otimes U_B)(\tilde{S}_A \otimes \tilde{S}_B)(U_A \otimes U_B)^\top = \underbrace{(U_A \tilde{S}_A U_A^\top)}_{=: \tilde{A}} \otimes \underbrace{(U_B \tilde{S}_B U_B^\top)}_{=: \tilde{B}}.$$

This is important as it allows for efficient sampling as described in [Appendix A.3.2](#).

**Computational cost.** Since we need the eigendecompositions of the Kronecker factors $A = U_A S_A U_A^\top$ and $B = U_B S_B U_B^\top$ for sampling in any case (see [Appendix A.3.2](#)), the computational overhead for the debiased K-FAC approximation $\tilde{A} \otimes \tilde{B}$ consists of computing a K-FAC approximation $C \otimes D$ on another mini-batch, re-evaluating the directional curvatures $\tilde{s}_A = \mathrm{Diag}(U_A^\top C U_A)$ and $\tilde{s}_B = \mathrm{Diag}(U_B^\top D U_B)$ and finally computing $\tilde{A} = U_A \mathrm{diag}(\tilde{s}_A) U_A^\top$ and $\tilde{B} = U_B \mathrm{diag}(\tilde{s}_B) U_B^\top$.

**From block-level to full matrix.** So far, we have only considered the debiasing of a single block. However, correcting the blocks's eigenvalues is sufficient because they coincide with the eigenvalues of the full matrix (due to the block-diagonal structure). Let $u^{(l)}$ denote an eigenvector of the $l$-th block $X^{(l)}$ of some block-diagonal matrix $X = \mathrm{blockdiag}_{l=1,\dots,L}(X^{(l)})$ corresponding to the eigenvalue $\lambda$. Then, $u^\top := (0^\top, \dots, u^{(l)\top}, \dots, 0^\top)$ is an eigenvector of $X$ corresponding to the same eigenvalue $\lambda$, because

$$X \cdot u = \mathrm{blockdiag}_{l=1,\dots,L}(X^{(l)}) \cdot u = \begin{pmatrix} X^{(1)} \cdot 0 \\ \vdots \\ X^{(l)} \cdot u^{(l)} \\ \vdots \\ X^{(L)} \cdot 0 \end{pmatrix} = \begin{pmatrix} 0 \\ \vdots \\ \lambda\, u^{(l)} \\ \vdots \\ 0 \end{pmatrix} = \lambda u.$$

The eigenvalues of $X$ thus coincide with the eigenvalues of its blocks; and $X$'s eigenvectors can be constructed from the the eigenvectors of the blocks by filling them up with zeros.

**Connection to [Equation (9)](#):** The equivalence between [Equation (9)](#) and the approach we describe above may not be obvious. We thus show here that the directional curvature of the debiased matrix $\hat{K}$ along an eigenvector $u$ of $K_\mathcal{B}$ indeed coincides with the directional curvature $u^\top K_{\tilde{\mathcal{B}}} u$ on $\tilde{\mathcal{B}}$.

More concretely, let $\hat{K} = \mathrm{blockdiag}_{l=1,\dots,L}(\hat{K}^{(l)})$ denote the debiased K-FAC approximation constructed from $K_\mathcal{B}$ and $K_{\tilde{\mathcal{B}}}$ as described above. Also, let $u^\top := (0^\top, \dots, u^{(l)\top}, \dots, 0^\top)$ denote an eigenvector of $K_\mathcal{B}$, where $u^{(l)}$ is the $i$-th eigenvector of $K_\mathcal{B}^{(l)}$ (i.e. the $i$-th column of $U$). It holds

$$u^\top \hat{K} u = u^{(l)\top} \hat{K}^{(l)} u^{(l)} = u^{(l)\top} U\tilde{S}U^\top u^{(l)} = e_i^\top \tilde{S} e_i = \tilde{s}_i = u^{(l)\top} K_{\tilde{\mathcal{B}}}^{(l)} u^{(l)} = u^\top K_{\tilde{\mathcal{B}}} u,$$

where $e_i$ denotes the $i$-th eigenvector.

## A.4 Bias in the directional slope

**Biased directional slopes along negative gradient directions.** Assume that we are given a quadratic $q(\,\cdot\,;\mathcal{B})$ around the current parameters $\theta_0$. We consider the negative normalized gradient direction $d = -\nabla q(\theta_\bullet;\mathcal{B}) \cdot \|\nabla q(\theta_\bullet;\mathcal{B})\|^{-1}$ at some location $\theta_\bullet$ evaluated on mini-batch $\mathcal{B}$. We have

$$\underbrace{\partial_d q(\theta_\bullet;\tilde{\mathcal{B}})}_{\bullet} = \underbrace{\partial_d q(\theta_\bullet;\mathcal{B})}_{\bullet} + \|\nabla q(\theta_\bullet;\mathcal{B})\|(1 - \cos(\alpha)) \geqslant \underbrace{\partial_d q(\theta_\bullet;\mathcal{B})}_{\bullet}, \tag{19}$$

where $\alpha := \angle(\nabla q(\theta_\bullet;\mathcal{B}), \nabla q(\theta_\bullet;\tilde{\mathcal{B}}))$ and we assumed $\|\nabla q(\theta_\bullet;\tilde{\mathcal{B}})\| = \|\nabla q(\theta_\bullet;\mathcal{B})\|$ which is true at least in expectation. Projecting a gradient onto its negative direction—the right-hand side of the

inequality—will *always* result in a directional slope that is $\leqslant 0$ since

$$\partial_{\boldsymbol{d}} \, q(\boldsymbol{\theta}_{\bullet}; \mathcal{B}) = \boldsymbol{d}^{\top} \nabla q(\boldsymbol{\theta}_{\bullet}; \mathcal{B}) = -\frac{\nabla q(\boldsymbol{\theta}_{\bullet}; \mathcal{B})^{\top}}{\|\nabla q(\boldsymbol{\theta}_{\bullet}; \mathcal{B})\|} \nabla q(\boldsymbol{\theta}_{\bullet}; \mathcal{B}) = -\|\nabla q(\boldsymbol{\theta}_{\bullet}; \mathcal{B})\| \leqslant 0$$

and $= 0$ only if $\nabla q(\boldsymbol{\theta}_{\bullet}; \mathcal{B}) = \boldsymbol{0}$. Projecting a *different* gradient (of equal length) onto that direction—that is the left-hand side of the inequality—will result in a larger (possibly even positive) directional slope.

**Proof of Equation (19).** Equation (19) quantifies the bias in the directional slope along $\boldsymbol{d}$ as a function of the alignment of the two mini-batch gradients. It holds

$$\underbrace{\partial_{\boldsymbol{d}} \, q(\boldsymbol{\theta}_{\bullet}; \mathcal{B})}_{\bullet} - \underbrace{\partial_{\boldsymbol{d}} \, q(\boldsymbol{\theta}_{\bullet}; \tilde{\mathcal{B}})}_{\bullet} = \boldsymbol{d}^{\top} \nabla q(\boldsymbol{\theta}_{\bullet}; \mathcal{B}) - \boldsymbol{d}^{\top} \nabla q(\boldsymbol{\theta}_{\bullet}; \tilde{\mathcal{B}})$$
$$= \underbrace{\|\boldsymbol{d}\|}_{=1} \|\nabla q(\boldsymbol{\theta}_{\bullet}; \mathcal{B})\| \underbrace{\cos(\pi)}_{=-1} - \underbrace{\|\boldsymbol{d}\|}_{=1} \|\nabla q(\boldsymbol{\theta}_{\bullet}; \tilde{\mathcal{B}})\| \cos(\gamma)$$
$$= \|\nabla q(\boldsymbol{\theta}_{\bullet}; \mathcal{B})\| (-1 - \cos(\gamma)),$$

where $\gamma = \angle(\boldsymbol{d}, \nabla q(\boldsymbol{\theta}_{\bullet}; \tilde{\mathcal{B}}))$ and we assumed that $\|\nabla q(\boldsymbol{\theta}_{\bullet}; \mathcal{B})\| = \|\nabla q(\boldsymbol{\theta}_{\bullet}; \tilde{\mathcal{B}})\|$ in the last step (which is true, at least, in expectation). Next, we re-write $\gamma$ as

$$\gamma = \angle(\boldsymbol{d}, \nabla q(\boldsymbol{\theta}_{\bullet}; \tilde{\mathcal{B}})) = \angle(-\nabla q(\boldsymbol{\theta}_{\bullet}; \mathcal{B}), \nabla q(\boldsymbol{\theta}_{\bullet}; \tilde{\mathcal{B}})) = \pi - \underbrace{\angle(\nabla q(\boldsymbol{\theta}_{\bullet}; \mathcal{B}), \nabla q(\boldsymbol{\theta}_{\bullet}; \tilde{\mathcal{B}}))}_{=:\alpha}.$$

It follows $-1 - \cos(\gamma) = -1 - \cos(\pi - \alpha) = \cos(\alpha) - 1$. Substituting this into the expression for the bias, we arrive at

$$\underbrace{\partial_{\boldsymbol{d}} \, q(\boldsymbol{\theta}_{\bullet}; \mathcal{B})}_{\bullet} - \underbrace{\partial_{\boldsymbol{d}} \, q(\boldsymbol{\theta}_{\bullet}; \tilde{\mathcal{B}})}_{\bullet} = \|\nabla q(\boldsymbol{\theta}_{\bullet}; \mathcal{B})\| (-1 - \cos(\gamma)) = \|\nabla q(\boldsymbol{\theta}_{\bullet}; \mathcal{B})\| (\cos(\alpha) - 1),$$

from which Equation (19) follows. $\square$

**The first CG search direction.** Assume that the current parameters are $\boldsymbol{\theta}_0$ and we apply CG to the local quadratic approximation $q(\,\cdot\,; \mathcal{B})$. The very first search direction is the quadratic's normalized negative gradient $\boldsymbol{d}_0 = -\nabla q(\boldsymbol{\theta}_0; \mathcal{B}) \cdot \|\nabla q(\boldsymbol{\theta}_0; \mathcal{B})\|^{-1}$ at $\boldsymbol{\theta}_0$. This is exactly the situation we describe above, where $\boldsymbol{\theta}_{\bullet}$ is set to $\boldsymbol{\theta}_0$. Equation (19) thus explains the bias for the first CG direction.

**The subsequent CG search directions.** For the subsequent CG search directions ($\boldsymbol{d}_p$ with $p \geqslant 1$), the situation is more complex: Each direction $\boldsymbol{d}_p$ is a linear combination of the current normalized negative residual $-\boldsymbol{r}_p$ (note that $\boldsymbol{r}_p$ coincides with the gradient $\nabla q(\boldsymbol{\theta}_p; \mathcal{B})$, see Appendix A.2) and the previous search direction $\boldsymbol{d}_{p-1}$ (see Appendix A.2) and the previous search direction $\boldsymbol{d}_{p-1}$ (see Algorithm 1), i.e.

$$\boldsymbol{d}_p = \frac{\boldsymbol{s}_p}{\|\boldsymbol{s}_p\|} = \frac{1}{\|\boldsymbol{s}_p\|}(-\boldsymbol{r}_p) + \frac{\beta_p}{\|\boldsymbol{s}_p\|} \boldsymbol{s}_{p-1} = \underbrace{\frac{\|\boldsymbol{r}_p\|}{\|\boldsymbol{s}_p\|}}_{=:\eta_1 \geqslant 0} \frac{-\boldsymbol{r}_p}{\|\boldsymbol{r}_p\|} + \underbrace{\frac{\beta_p \|\boldsymbol{s}_{p-1}\|}{\|\boldsymbol{s}_p\|}}_{=:\eta_2 \geqslant 0} \boldsymbol{d}_{p-1} = \eta_1 \frac{-\boldsymbol{r}_p}{\|\boldsymbol{r}_p\|} + \eta_2 \, \boldsymbol{d}_{p-1}.$$

The additional correction term ensures conjugacy. The directional slope along $\boldsymbol{d}_p$ thus also splits into two corresponding terms: The slope along the negative gradient direction and the slope along the previous search direction. It holds

$$\partial_{\boldsymbol{d}_p} q(\boldsymbol{\theta}_p; \mathcal{B}) = \boldsymbol{d}_p^{\top} \nabla q(\boldsymbol{\theta}_p; \mathcal{B})$$
$$= \eta_1 \frac{-\boldsymbol{r}_p}{\|\boldsymbol{r}_p\|}^{\top} \nabla q(\boldsymbol{\theta}_p; \mathcal{B}) + \eta_2 \, \boldsymbol{d}_{p-1}^{\top} \nabla q(\boldsymbol{\theta}_p; \mathcal{B})$$
$$= \eta_1 \partial_{-\boldsymbol{r}_p / \|\boldsymbol{r}_p\|} q(\boldsymbol{\theta}_p; \mathcal{B}) + \eta_2 \, \partial_{\boldsymbol{d}_{p-1}} q(\boldsymbol{\theta}_p; \mathcal{B}).$$

The bias in the first term—as explained by Equation (19)—also introduces a bias along the search direction $\boldsymbol{d}_p$.

A.5 BIAS IN THE DIRECTIONAL CURVATURE

**Derivation for Equation (6).** Let $\boldsymbol{H}_{\mathcal{B}} = \boldsymbol{U} \boldsymbol{\Lambda} \boldsymbol{U}^{\top}$ and $\boldsymbol{H}_{\tilde{\mathcal{B}}} = \tilde{\boldsymbol{U}} \tilde{\boldsymbol{\Lambda}} \tilde{\boldsymbol{U}}^{\top}$ denote the eigendecompositions of the mini-batch Hessians, where $\boldsymbol{U} = (\boldsymbol{u}_1, \dots, \boldsymbol{u}_P), \tilde{\boldsymbol{U}} = (\tilde{\boldsymbol{u}}_1, \dots, \tilde{\boldsymbol{u}}_P) \in \mathbb{R}^{P \times P}$ contain

the orthonormal eigenvectors and $\mathbf{\Lambda} = \mathrm{diag}(\lambda_1, \ldots, \lambda_P)$, $\tilde{\mathbf{\Lambda}} = \mathrm{diag}(\tilde{\lambda}_1, \ldots, \tilde{\lambda}_P)$ the respective eigenvalues in descending order, i.e. $\lambda_1 \geqslant \ldots \geqslant \lambda_P$ and $\tilde{\lambda}_1 \geqslant \ldots \geqslant \tilde{\lambda}_P$. The directional curvature along one of $\boldsymbol{H}_{\mathcal{B}}$'s eigenvectors $\boldsymbol{u}_i$ on mini-batch $\mathcal{B}$ is given by the corresponding eigenvalue since

$$\underbrace{\partial^2_{\boldsymbol{u}_i} q(\boldsymbol{\theta}_\bullet; \mathcal{B})}_{\bullet} = \boldsymbol{u}_i^\top \boldsymbol{H}_{\mathcal{B}} \boldsymbol{u}_i = \boldsymbol{u}_i^\top \lambda_i \boldsymbol{u}_i = \lambda_i \|\boldsymbol{u}_i\|_2^2 = \lambda_i.$$

For $\tilde{\mathcal{B}}$, we obtain

$$\underbrace{\partial^2_{\boldsymbol{u}_i} q(\boldsymbol{\theta}_\bullet; \tilde{\mathcal{B}})}_{\bullet} = \boldsymbol{u}_i^\top \boldsymbol{H}_{\tilde{\mathcal{B}}} \boldsymbol{u}_i = \boldsymbol{u}_i^\top \left( \sum_{p=1}^{P} \tilde{\lambda}_p \tilde{\boldsymbol{u}}_p \tilde{\boldsymbol{u}}_p^\top \right) \boldsymbol{u}_i = \sum_{p=1}^{P} \tilde{\lambda}_p \underbrace{(\tilde{\boldsymbol{u}}_p^\top \boldsymbol{u}_i)^2}_{=:\boldsymbol{\Omega}_{i,p}} = \sum_{p=1}^{P} \tilde{\lambda}_p \, \boldsymbol{\Omega}_{i,p}.$$

**Weights sum up to one.** The weights $\boldsymbol{\Omega}_{i,p}$ are non-negative and sum up to one, i.e. $\sum_{p=1}^{P} \boldsymbol{\Omega}_{i,p} = 1$. This is because

$$\sum_{p=1}^{P} \boldsymbol{\Omega}_{i,p} = \sum_{p=1}^{P} (\tilde{\boldsymbol{u}}_p^\top \boldsymbol{u}_i)^2 = \|\tilde{\boldsymbol{U}}^\top \boldsymbol{u}_i\|_2^2 = \boldsymbol{u}_i^\top \underbrace{\tilde{\boldsymbol{U}} \tilde{\boldsymbol{U}}^\top}_{=\boldsymbol{I}} \boldsymbol{u}_i = \|\boldsymbol{u}_i\|_2^2 = 1 \qquad (20)$$

where we used that $\tilde{\boldsymbol{U}}$ is an orthogonal matrix and that $\boldsymbol{u}_i$ is normalized.

**Proof of Equation (7).** Here, we assume the simplified case where the spectra of $\boldsymbol{H}_{\mathcal{B}}$ and $\boldsymbol{H}_{\tilde{\mathcal{B}}}$ are identical, i.e. $\lambda_p = \tilde{\lambda}_p \, \forall p \in \{1, \ldots, P\}$.

First, consider $\boldsymbol{u}_1$ and assume that $\boldsymbol{u}_1$ and $\tilde{\boldsymbol{u}}_1$ are *not* perfectly aligned, i.e. $\boldsymbol{\Omega}_{1,1} = (\boldsymbol{u}_1^\top \tilde{\boldsymbol{u}}_1)^2 < 1$. It holds

$$\underbrace{\partial^2_{\boldsymbol{u}_1} q(\boldsymbol{\theta}_0, \tilde{\mathcal{B}})}_{\bullet} \stackrel{(6)}{=} \sum_{p=1}^{P} \lambda_p \boldsymbol{\Omega}_{1,p} = \lambda_1 \boldsymbol{\Omega}_{1,1} + \sum_{p=2}^{P} \lambda_p \boldsymbol{\Omega}_{1,p}.$$

The sum can be bounded from above by putting all remaining weight $1 - \boldsymbol{\Omega}_{1,1}$ (see Equation (20)) on the second-largest eigenvalue $\lambda_2$, i.e.

$$\underbrace{\partial^2_{\boldsymbol{u}_1} q(\boldsymbol{\theta}_0, \tilde{\mathcal{B}})}_{\bullet} \leqslant \lambda_1 \boldsymbol{\Omega}_{1,1} + \lambda_2 (1 - \boldsymbol{\Omega}_{1,1}) \stackrel{(*)}{\leqslant} \lambda_1 \boldsymbol{\Omega}_{1,1} + \lambda_1 (1 - \boldsymbol{\Omega}_{1,1}) = \lambda_1 = \underbrace{\partial^2_{\boldsymbol{u}_1} q(\boldsymbol{\theta}_0, \mathcal{B})}_{\bullet}.$$

The inequality $(*)$ turns into $<$ if the top two eigenvalues are separated, i.e. $\lambda_1 > \lambda_2$. The proof for $\boldsymbol{u}_P$ is similar. We assume that $\boldsymbol{u}_P$ and $\tilde{\boldsymbol{u}}_P$ are *not* perfectly aligned, i.e. $\boldsymbol{\Omega}_{P,P} = (\boldsymbol{u}_P^\top \tilde{\boldsymbol{u}}_P)^2 < 1$. It follows

$$\underbrace{\partial^2_{\boldsymbol{u}_P} q(\boldsymbol{\theta}_0, \tilde{\mathcal{B}})}_{\bullet} \stackrel{(6)}{=} \sum_{p=1}^{P} \lambda_p \boldsymbol{\Omega}_{P,p} = \sum_{p=1}^{P-1} \lambda_p \boldsymbol{\Omega}_{P,p} + \lambda_P \boldsymbol{\Omega}_{P,P}.$$

This time, we bound the sum from below by putting all remaining weight $1 - \boldsymbol{\Omega}_{P,P}$ (see Equation (20)) on the second-smallest eigenvalue $\lambda_{P-1}$, i.e.

$$\underbrace{\partial^2_{\boldsymbol{u}_P} q(\boldsymbol{\theta}_0, \tilde{\mathcal{B}})}_{\bullet} \geqslant \lambda_{P-1}(1 - \boldsymbol{\Omega}_{P,P}) + \lambda_P \boldsymbol{\Omega}_{P,P} \stackrel{(*)}{\geqslant} \lambda_P(1 - \boldsymbol{\Omega}_{P,P}) + \lambda_P \boldsymbol{\Omega}_{P,P} = \lambda_P = \underbrace{\partial^2_{\boldsymbol{u}_P} q(\boldsymbol{\theta}_0, \mathcal{B})}_{\bullet}.$$

Again, the inequality $(*)$ turns into $>$ if the bottom two eigenvalues are separated, i.e. $\lambda_{P-1} > \lambda_P$. This concludes the proof of Equation (7). $\square$

## B EXPERIMENTAL DETAILS

In the following, we provide information on the experimental setup and give detailed instructions on how to replicate all empirical results.

B.1 TEST PROBLEMS AND TRAINING PROCEDURES

Throughout the paper, we use a series of test problems with different models, datasets, and training procedures which we describe in more detail in the following. We use DEEPOBS (Schneider et al., 2019) on top of PYTORCH (Paszke et al., 2019) as our general benchmarking framework as it provides easy access to a variety of datasets and model architectures.

**Data.** We use the datasets CIFAR-10 (with $C = 10$ classes) and CIFAR-100 (with $C = 100$ classes) (Krizhevsky, 2009). Each dataset contains 60,000 data points that are split into 40,000 training samples, 10,000 validation samples and 10,000 test samples. For the experiments on out-of-distribution (OOD) data, we create the datasets CIFAR-10-C and CIFAR-100-C, each containing 10,000 images, as described in (Hendrycks & Dietterich, 2019). In these datasets, each image is corrupted using one out of 15 different corruptions (chosen uniformly at random) at a specific severity level (a number between 1 and 5). We also use the IMAGENET dataset (Deng et al., 2009) which contains images from $C = 1000$ different classes.

**Test problems.** All of the following test problems use the cross-entropy loss function.

(A) **ALL-CNN-C on CIFAR-100.** This test problem uses the ALL-CNN-C model architecture (Springenberg et al., 2015) (where we removed the dropout layers as explained in Section 3.1) and CIFAR-100 data. The training hyperparameters are taken from an existing benchmark (Schmidt et al., 2021): We train the model with SGD (learning rate 0.171234) with batch size 256 for 350 epochs. Weight decay $\beta = 0.0005$ is used on the weights but not the biases of the model.

(B) **ALL-CNN-C on CIFAR-10.** This test problem is similar to (A) but uses the CIFAR-10 dataset. The model is trained with SGD (learning rate 0.025, momentum 0.9) with batch size 256 for 350 epochs. The learning rate is reduced by a factor of 10 at epochs 200, 250 and 300, as suggested in (Springenberg et al., 2015). Weight decay $\beta = 0.001$ is used on the weights but not the biases of the model.

(C) **WIDERESNET 40-4 on CIFAR-100.** This is a test problem from DEEPOBS. For details on the architecture, see (Zagoruyko & Komodakis, 2017). We train this model with SGD (learning rate 0.1, momentum 0.9) with batch size 128 for 160 epochs. The learning rate is reduced by a factor of 5 after 60 and 120 epochs. Weight decay $\beta = 0.0005$ is used on the non-bias weights of the model.

(D) **CONVNET on CIFAR-10.** This test problem uses a simple convolutional neural network with variable depth $d$ and width $w$ (denoted by "model $d$-$w$" in Figures 7 and 22). The *first* block of the model consists of a convolutional layer (kernel size 5, padding 2) with 3 input and $w$ output channels, a ReLU activation function, and a max-pooling layer (kernel size 2). The *last* block consists of a max-pooling layer (kernel size 2), a flatten layer, and a dense linear layer. In between those blocks, there are $d$ *hidden* blocks each consisting of a convolutional layer (kernel size 5, padding 2) with $w$ input and $w$ output channels followed by a ReLU activation function. So, the depth $d$ determines the number of hidden blocks, whereas $w$ controls the number of parameters in the layers and is thus referred to as the width. We use $d \in \{1, 4, 7\}$ and $w \in \{32, 64, 128\}$ (resulting in 9 models) and train each model for 100 epochs on the CIFAR-10 dataset using ADAM with standard hyperparameters (learning rate 0.001, $\beta_1 = 0.9$, $\beta_2 = 0.999$, $\epsilon = 10^{-8}$) with batch size 256. No weight decay is used for this test problem.

(E) **RESNET-50 on IMAGENET.** This test problem uses the RESNET-50 model architecture (He et al., 2016) and the IMAGENET dataset. We use the IMAGENET1K_V1 weights pretrained on IMAGENET-1K using SGD (initial learning rate 0.1, momentum 0.9) with batch size 256 for 90 epochs. For the pre-training, the learning rate was set by a multistep scheduler which multiplied the learning rate by a factor of 0.1 after every 30 epochs. Additionally, a weight decay of $\beta = 0.0001$ was used on all weights apart from biases and the learned batch normalization weights.

(F) **VIT LITTLE on IMAGENET.** This test problem uses the VIT LITTLE architecture of the VISION TRANSFORMER model family (Dosovitskiy et al., 2021) on the IMAGENET dataset. We use the VIT_LITTLE_PATCH16_REG4_GAP_256.SBB_IN1K weights pre-trained on IMAGENET-1K using NADAMW. For the exact hyperparameters used for the weights, refer to the HuggingFace Model Card.

During training, we store the model's parameters at 10 checkpoints spaced log-equidistantly between the first and last epoch. The training metrics for all test problems are shown in Figure 7.

## B.2 MATRIX-VECTOR PRODUCTS WITH THE CURVATURE MATRIX

**Access to curvature matrices via BACKPACK.** In order to compute an eigendecomposition of the curvature matrix or apply the CG method to a quadratic, we need access to matrix-vector products with the curvature matrix $v \mapsto H_\mathcal{B} \cdot v$. When products with the full-batch curvature matrix are required, we accumulate the mini-batch quantities over manageable chunks of the training set. Our implementation uses BACKPACK (Dangel et al., 2020) that provides access to products with the Hessian $v \mapsto \nabla^2 \mathcal{L}(\theta; \mathcal{B}) \cdot v$ and GGN $v \mapsto G_\mathcal{B} \cdot v$ of the empirical risk $\mathcal{L}$ as well as the K-FAC curvature approximation $K_\mathcal{B}$.

**Eigenvectors of the curvature matrix.** Given access to matrix-vector products with the curvature matrix, we can construct an instance of a `scipy.sparse.linalg.LinearOperator` that can be used in `scipy.sparse.linalg.eigsh`.

## B.3 SECTION 1 (INTRODUCTION): FIGURE 1

For the visual abstract in Figure 1, we use the fully trained ALL-CNN-C model from test problem (A) (see Appendix B.1). The experimental procedure below is repeated for 5 different mini-batches $\mathcal{B}_m$, $m \in \{0, 1, 2, 3, 4\}$ of size $|\mathcal{B}_m| = 512$ (we omit the index $m$ in the following).

**Experimental procedure.** We use the GGN curvature proxy $H_\mathcal{B} \leftarrow G_\mathcal{B} + \beta I$ and compute its top two eigenvectors $u_1$ and $u_2$ (see Appendix B.2). In order to evaluate the quadratic $q(\theta_\star; \mathcal{B})$ in the 2D space spanned by those eigenvectors efficiently, we use the following equation:

$$q(\theta_\star + \tau_1 u_1 + \tau_2 u_2; \mathcal{B}) = \frac{1}{2}\tau_1^2[u_1^\top H_\mathcal{B} u_1] + \tau_1\tau_2[u_1^\top H_\mathcal{B} u_2] + \frac{1}{2}\tau_2^2[u_2^\top H_\mathcal{B} u_2]$$
$$+ \tau_1[u_1^\top g_\mathcal{B}] + \tau_2[u_2^\top g_\mathcal{B}] + [c_\mathcal{B}].$$

We can compute all the terms on the right-hand side in brackets *once*, store them, and then evaluate the quadratic for arbitrary values of $\tau_1$ and $\tau_2$ at basically no cost. The derivation for the full-batch quadratic $q(\theta_\star + \tau_1 u_1 + \tau_2 u_2; \mathcal{D})$ is analogous.

## B.4 SECTION 3 (THE SHAPE OF A MINI-BATCH QUADRATIC): FIGURES 2 TO 4

For the evaluation of the directional derivatives, we use the fully trained ALL-CNN-C model from test problem (A) (see Appendix B.1).

**Three settings.** Throughout this section, we consider three different settings: (i) Quadratics that use the Hessian of the empirical risk, i.e. $H_\mathcal{B} \leftarrow \nabla^2 \mathcal{L}(\theta_\star; \mathcal{B}) + \beta I$ at batch size 512, (ii) quadratics that use the Hessian's GGN approximation, i.e. $H_\mathcal{B} \leftarrow G_\mathcal{B} + \beta I$ at batch size 512 and (iii) at batch size 2048. For each setting, the experimental procedure below is repeated for 3 mini-batches $\mathcal{B}_m$, $m \in \{0, 1, 2\}$. The prior precision $\beta$ is set to 0.0005 (the same $\beta$ was used for training, see Appendix B.1) and only acts on the weights of the model but not its biases.

**Experimental procedure (biases).** The experimental procedure consists of two steps: The computations of the directions on mini-batch $\mathcal{B}_m$ and the evaluation of the directional derivatives on *all* mini-batches (of the same size) in the training dataset.

1. **Directions based on $\mathcal{B}_m$.** First, we use the quadratic $q(\theta_\star; \mathcal{B}_m)$ with the given curvature proxy and batch size to compute a set of 100 directions. This is either the top 100 eigenvectors of $H_\mathcal{B}$ computed via `scipy.sparse.linalg.eigsh` (see Appendix B.2) or the first 100 CG search directions (see Appendix B.5). In the case of CG, we also store the trajectory of the iterates $\theta_\star = \theta_0, \theta_1, \ldots, \theta_{100}$.

2. **Directional derivatives on all mini-batches.** Finally, we evaluate the directional slope and curvature for all directions from step 1 on *all* mini-batches $\mathcal{B}_{m'}$, $m' \in \{0, \ldots, M-1\}$ in the training dataset (where $M$ denotes the number of mini-batches in the training data).

   - **Directional derivatives along eigenvectors.** For an eigenvector $u$ the directional slope and curvature are given by $\partial_u q(\theta_\star; \mathcal{B}_{m'})$ and $\partial_u^2 q(\theta_\star; \mathcal{B}_{m'})$, respectively.

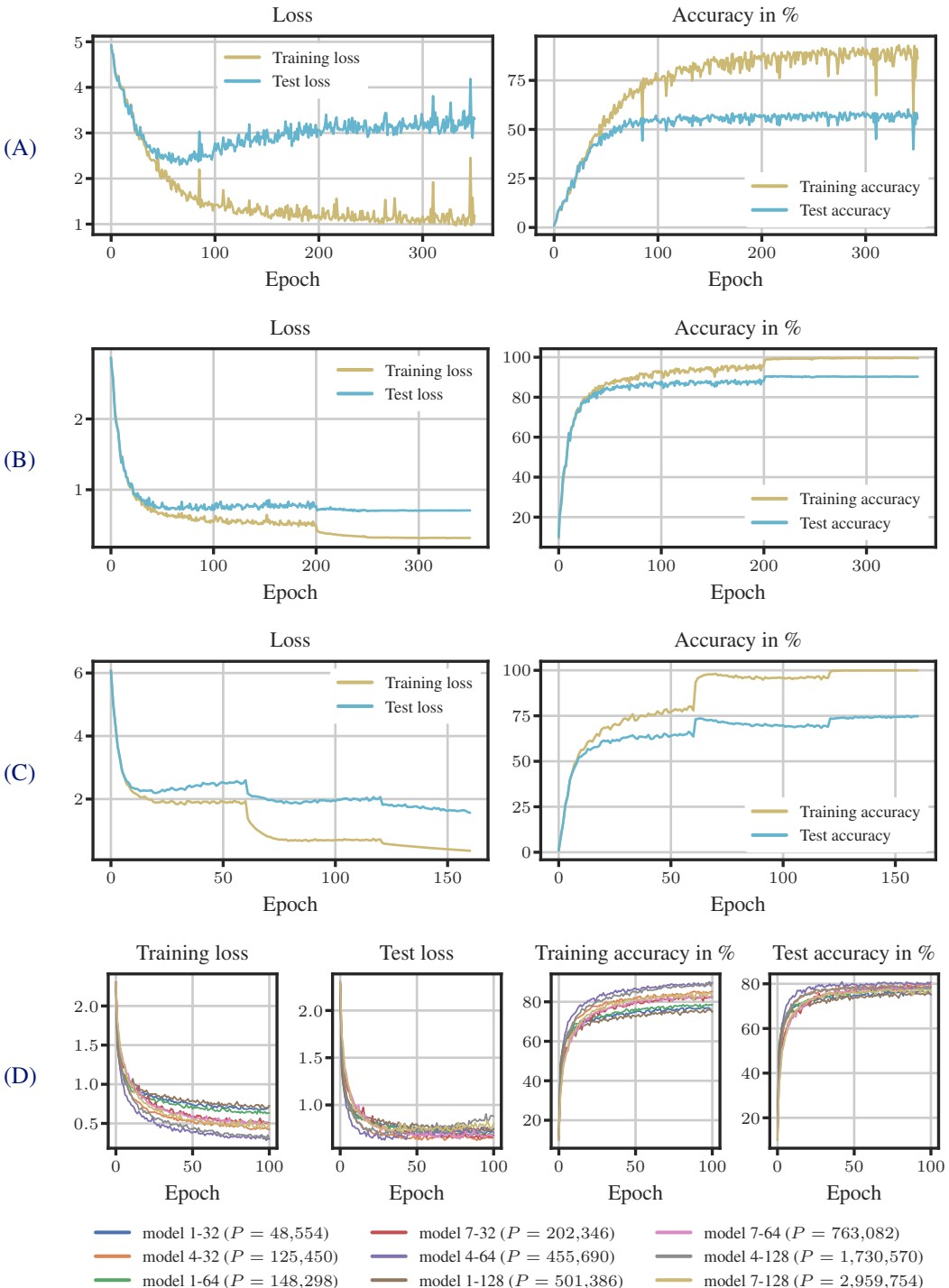

**Figure 7: Training metrics for all test problems.** The panels show the training/test loss/accuracy during training for all test problems (A) (ALL-CNN-C on CIFAR-100), (B) (ALL-CNN-C on CIFAR-10), (C) (WIDERESNET 40-4 on CIFAR-100) and (D) (CONVNET on CIFAR-10). Test problem (E) (RESNET-50 on IMAGENET) is not shown since a pre-trained model is used here.

- **Directional derivatives along CG search directions.** For CG, we evaluate the directional slope and curvature along each search direction $\boldsymbol{d}_p$ at the respective iterate $\boldsymbol{\theta}_p$ of the trajectory, i.e. $\partial_{\boldsymbol{d}_p} q(\boldsymbol{\theta}_p; \mathcal{B}_{m'})$ for the slope and $\partial^2_{\boldsymbol{d}_p} q(\boldsymbol{\theta}_p; \mathcal{B}_{m'})$ for the curvature. As the curvature $\nabla^2 q(\boldsymbol{\theta}; \mathcal{B}_{m'}) \equiv \boldsymbol{H}_{\mathcal{B}_{m'}}$ is independent of $\boldsymbol{\theta}$, this small distinction is irrelevant for the directional curvature. For the directional slope, however, it does make a difference.

**Results (biases).** The results for setting (i) are shown in Figure 8, for setting (ii) in Figures 2 and 9 and for (iii) in Figure 10. The biases in the directional slopes and curvatures can be observed across all scenarios. The biases in the slopes are even more pronounced along the CG search directions than along the eigenvectors of the curvature matrix. The opposite holds for the curvature biases that tend to be larger along the eigenvectors. This is consistent with our explanations from Section 3.2. Both the biases in the slope and curvature decrease with increasing mini-batch size. For setting (i), CG encounters a search direction with negative curvature (indicating that, as expected, the curvature matrix $\boldsymbol{H}_{\mathcal{B}}$ is indefinite) and thus terminates already after 4 iterations.

**Experimental procedure (overlaps).** Next, we compute the overlaps between the eigenspaces of the curvature matrices on different mini-batches $\mathcal{B}$ and $\tilde{\mathcal{B}}$. More specifically, we compute $\boldsymbol{\Omega}_{i,p} = (\boldsymbol{u}_i^\top \tilde{\boldsymbol{u}}_p)^2$, where $i, p \in \{1, \ldots, 100\}$, $\boldsymbol{u}_i$ is an eigenvector of $\boldsymbol{H}_{\mathcal{B}}$ and $\tilde{\boldsymbol{u}}_p$ is an eigenvector of $\boldsymbol{H}_{\tilde{\mathcal{B}}}$. The values $\boldsymbol{\Omega}_{i,p}$ are bounded between 0 ($\boldsymbol{u}_i$ and $\tilde{\boldsymbol{u}}_p$ are orthogonal) and 1 ($\boldsymbol{u}_i$ and $\tilde{\boldsymbol{u}}_p$ are identical, up to their sign). We apply a log-transform to $\boldsymbol{\Omega}_{i,p}$. In Figure 3 and Appendix B.4, values below $-8$ (i.e. $\boldsymbol{\Omega}_{i,p} \leqslant 10^{-8}$) are shown in black, values equal to 0 (i.e. $\boldsymbol{\Omega}_{i,p} = 10^0 = 1$) are shown in white.

**Results (overlaps).** The results are shown in Appendix B.4. They show that the eigenspaces of the curvature matrices are not perfectly aligned: Eigenvectors from $\mathcal{B}_m$ typically overlap with several eigenvectors from $\tilde{\mathcal{B}}$. The eigenspaces are more aligned at the larger batch size 2048 than at batch size 512. This seems reasonable since random vectors in high-dimensional spaces tend to be orthogonal to each other —less stochasticity (due to a larger batch size) thus leads to better alignment.

## B.5 SECTION 6.1 (DEBIASED CONJUGATE GRADIENTS): FIGURE 5

Here, we describe the experimental details for Figure 5 from Section 6.1. For the derivation and the mathematical details of the standard single-batch CG method and the debiased approach, see Appendix A.2. The experiment uses the fully trained ALL-CNN-C model from test problem (A) (see Appendix B.1) and the GGN curvature proxy $\boldsymbol{H}_{\mathcal{B}} \leftarrow \boldsymbol{G}_{\mathcal{B}} + \beta \boldsymbol{I}$ with $\beta = 0.0005$.

**Experimental procedure.** The experimental procedure consists of two steps: The computation of the CG trajectories and the evaluation of the four performance metrics.

1. **Computation of the CG trajectories.** For the single-batch CG approach, we use one mini-batch of size 1024, apply $K = 30$ CG iterations and store the trajectory $\boldsymbol{\theta}_\star = \boldsymbol{\theta}_0, \boldsymbol{\theta}_1, \ldots, \boldsymbol{\theta}_{30}$. For the debiased approach (details in Appendix A.2), we use two mini-batches of size 512 to construct the trajectory. As the debiased approach uses a total of 60 GGN-vector products (30 for the search directions and 30 for the update magnitudes) at half the computational cost (since the cost for the GGN-vector product scales linearly with the mini-batch size), the total cost of both approaches are comparable.

   For both approaches, the above procedure is repeated for 5 different mini-batches/mini-batch pairs, such that 10 trajectories are computed in total.

   We compute two additional trajectories: The standard CG trajectory for the full-batch quadratic $q(\cdot; \mathcal{D})$ and the trajectory for a debiased full-batch approach, where we use half the training data for the directions and the other half for the update magnitudes.

2. **Evaluation of performance metrics.** For each of the 12 trajectories, we evaluate four performance metrics: (i) the regularized loss $\mathcal{L}_{\text{reg}}$ (see Equation (1)) on the training set $\mathcal{D}$ and (ii) on the test set $\mathcal{D}_{\text{test}}$ as well as the accuracy (i.e. the relative number of correctly classified samples) on the (iii) training and (iv) test dataset.

**Results.** The mini-batch results are shown in Figure 5 and discussed in Section 6.1. The results for the two full-batch CG variants are shown in Figure 12. Surprisingly, the full-batch CG approach diverges after roughly 10 iterations. If we consider the training data as a (large) sample from the data

**Biases: Additional results for $H_{\mathcal{B}} \leftarrow \nabla^2 \mathcal{L}(\theta_\star; \mathcal{B}) + \beta I$ (batch size 512)**

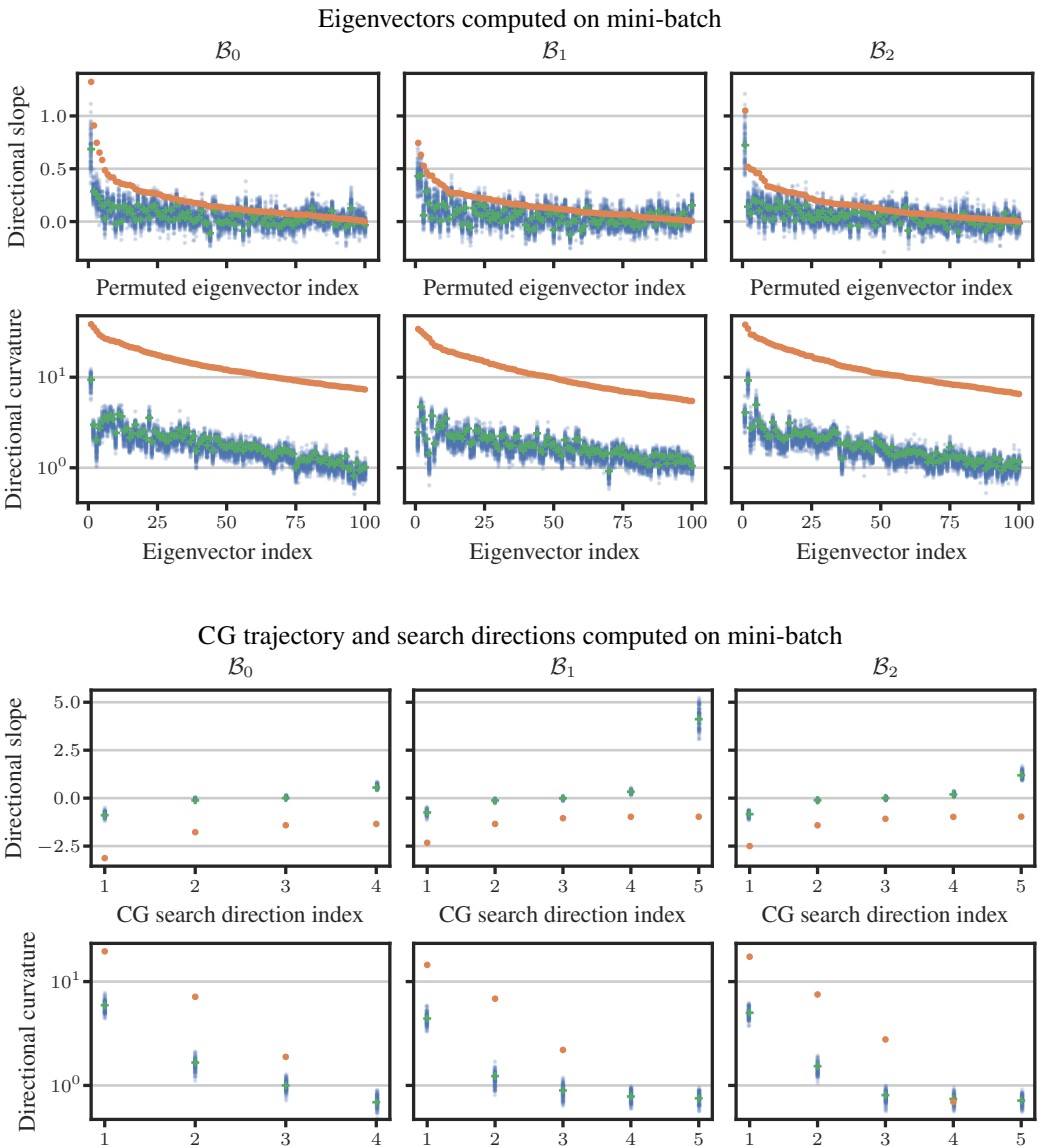

**Figure 8: Directional slopes and curvatures are biased.** The experimental setting is similar to Figure 2 but uses the Hessian of the empirical risk, i.e. $H_{\mathcal{B}} \leftarrow \nabla^2 \mathcal{L}(\theta_\star; \mathcal{B}) + \beta I$ at batch size 512. The upper plot *(Top)* shows the directional derivatives along the top 100 eigenvectors of $H_{\mathcal{B}}$, the lower plot *(Bottom)* shows the directional derivatives along the first 100 CG search directions. For the top panel of the upper plot, we switch the order and sign of the eigenvectors such that the orange dots are all above zero and in descending order. There are strong systematic biases in the directional slopes and curvatures. The curvature biases are more pronounced along the eigenvectors, whereas the biases in the slope are larger along the CG directions.

**Biases: Additional results for $H_{\mathcal{B}} \leftarrow G_{\mathcal{B}} + \beta I$ (batch size $512$)**

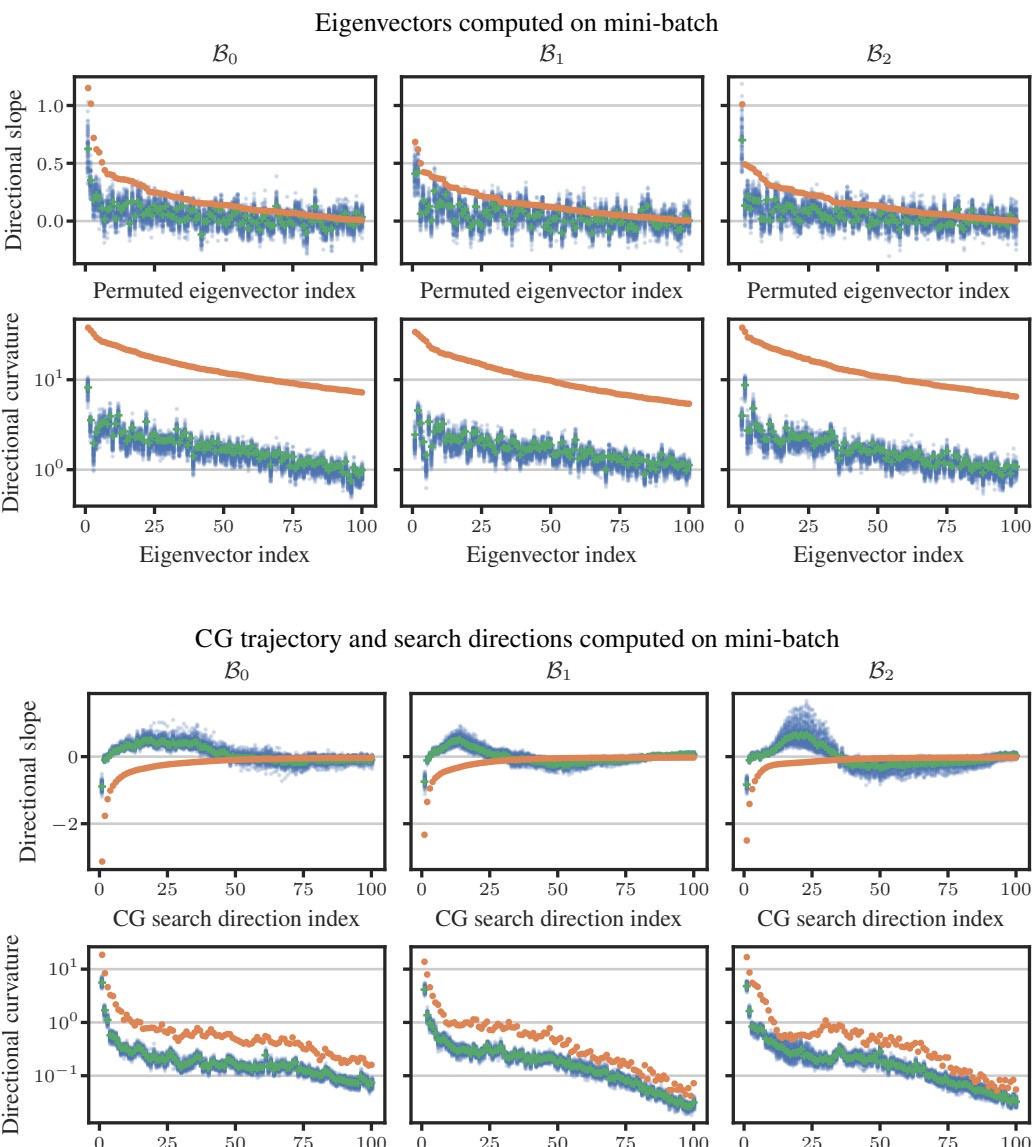

**Figure 9: Directional slopes and curvatures are biased.** The experimental setting is the same as in Figure 2: We use the GGN curvature proxy $H_{\mathcal{B}} \leftarrow G_{\mathcal{B}} + \beta I$ at batch size $512$. The upper plot *(Top)* shows the directional derivatives along the top $100$ eigenvectors of $H_{\mathcal{B}}$, the lower plot *(Bottom)* shows the directional derivatives along the first $100$ CG search directions. For the top panel of the upper plot, we switch the order and sign of the eigenvectors such that the orange dots are all above zero and in descending order. There are strong systematic biases in the directional slopes and curvatures. The curvature biases are more pronounced along the eigenvectors, whereas the biases in the slope are larger along the CG directions.

**Biases: Additional results for $H_\mathcal{B} \leftarrow G_\mathcal{B} + \beta I$ (batch size 2048)**

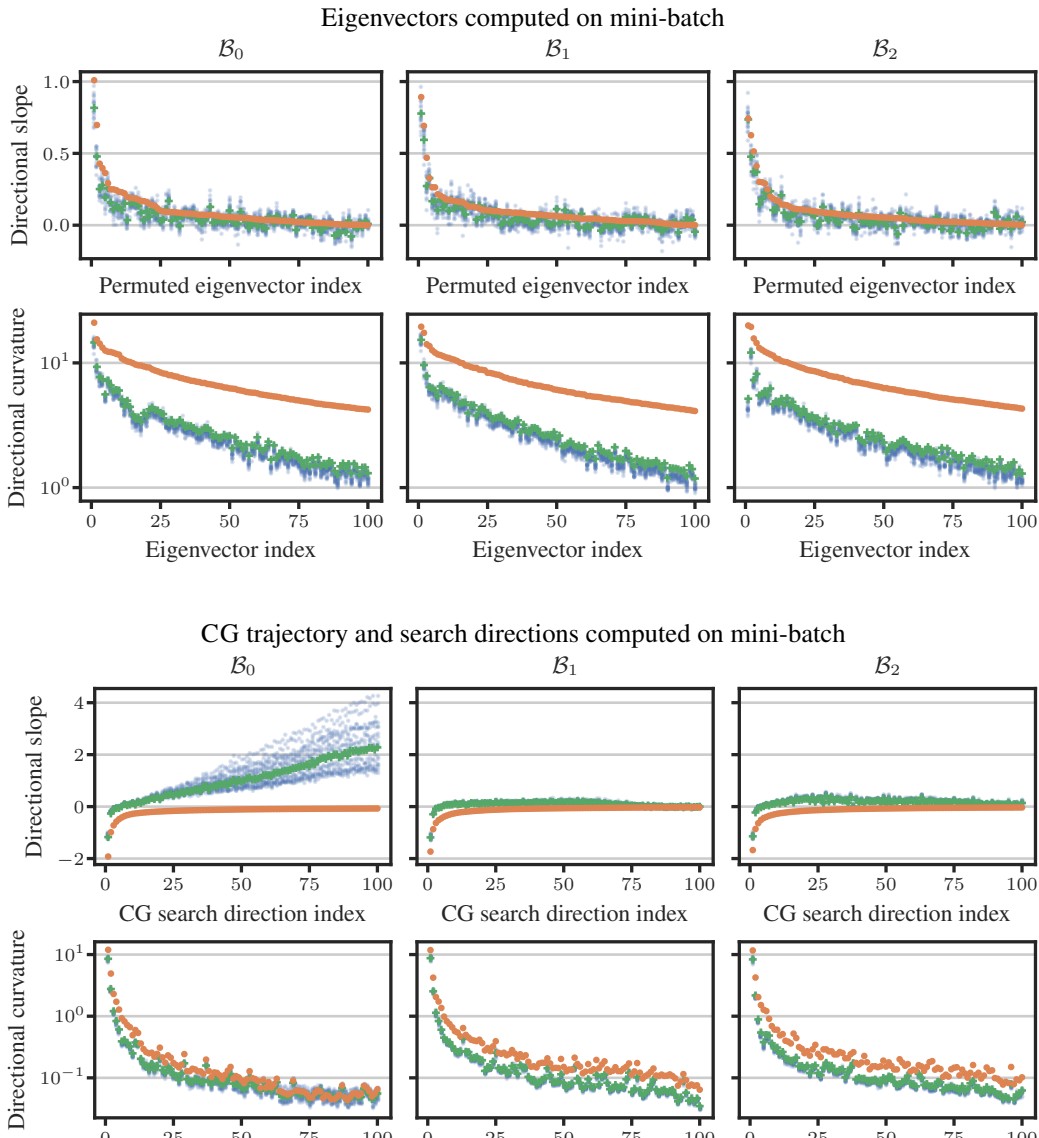

**Figure 10: Directional slopes and curvatures are biased.** The experimental setting is similar to Figure 2: We use the GGN curvature proxy $H_\mathcal{B} \leftarrow G_\mathcal{B} + \beta I$ but batch size 2048. The upper plot *(Top)* shows the directional derivatives along the top 100 eigenvectors of $H_\mathcal{B}$, the lower plot *(Bottom)* shows the directional derivatives along the first 100 CG search directions. For the top panel of the upper plot, we switch the order and sign of the eigenvectors such that the orange dots are all above zero and in descending order. There are strong systematic biases in the directional slopes and curvatures. The curvature biases are more pronounced along the eigenvectors, whereas the biases in the slope are larger along the CG directions.

**Eigenspace overlaps: Additional result for** $H_{\mathcal{B}} \leftarrow \nabla^2 \mathcal{L}(\boldsymbol{\theta}_\star; \mathcal{B}) + \beta \boldsymbol{I}$

**Batch size** 512

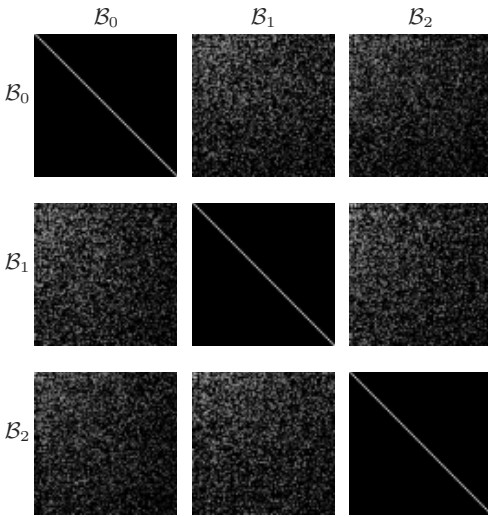

**Eigenspace overlaps: Additional results for** $H_{\mathcal{B}} \leftarrow G_{\mathcal{B}} + \beta \boldsymbol{I}$

**Batch size** 512      **Batch size** 248

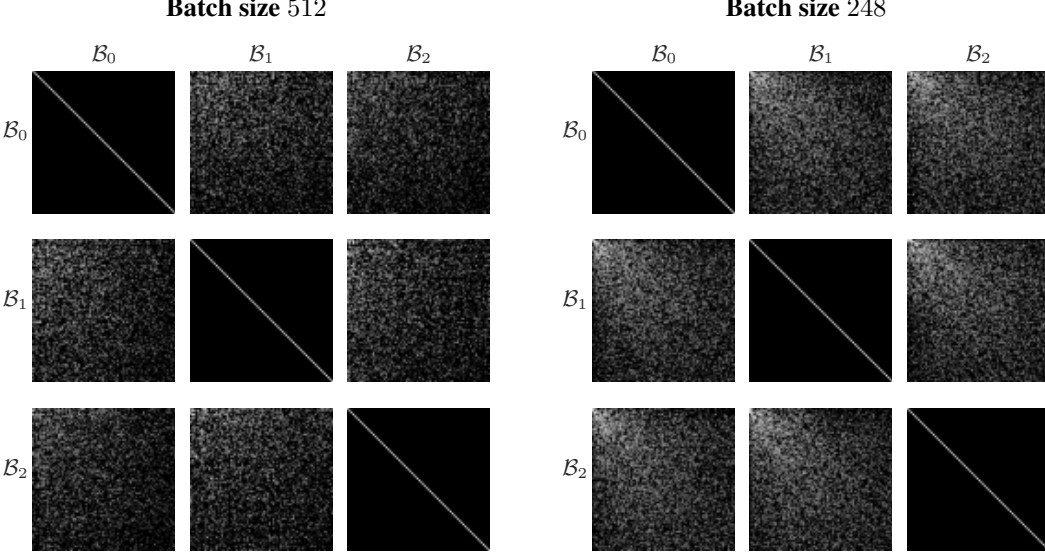

**Figure 11: In practice, eigenspaces are misaligned.** The experimental setting is similar to Figure 3. The top plot shows overlaps between three curvature matrices that use the Hessian of the empirical risk, i.e. $H_{\mathcal{B}} \leftarrow \nabla^2 \mathcal{L}(\boldsymbol{\theta}_\star; \mathcal{B}) + \beta \boldsymbol{I}$ at batch size 512. The bottom plots use the GGN curvature proxy $H_{\mathcal{B}} \leftarrow G_{\mathcal{B}} + \beta \boldsymbol{I}$ at batch size 512 *(Left)* and 2048 *(Right)*. The weights $\boldsymbol{\Omega}_{i,j}$ are shown as a $100 \times 100$ greyscale image (color ranges from black for $\boldsymbol{\Omega}_{i,j} \leqslant 10^{-8}$ to white for $\boldsymbol{\Omega}_{i,j} = 1$) for $m, m' \in \{0, 1, 2\}$. Clearly, the eigenspaces for different mini-batches are not perfectly aligned as eigenvectors from $\mathcal{B}_m$ overlap with several eigenvectors from $\mathcal{B}_{m'}$. At the larger batch size, the eigenspaces are more aligned.

distribution, the full-batch CG approach can be seen as an instance of the mini-batch approach with a very large mini-batch size. Thus, it might also exhibit the associated biases we describe in Section 3.2. Another explanation would be the approximation error in Equation (2): $\mathcal{L}_{\text{reg}}(\,\cdot\,;\mathcal{D}) \approx q(\,\cdot\,;\mathcal{D})$. If the trajectory moves far away from $\boldsymbol{\theta}_\star$, it leaves the region where the approximation is valid (whereas the debiased full-batch version might take smaller steps and thus remains stable). We leave it to future work to investigate this further.

**Additional results.** We also provide the results for two other mini-batch sizes $|\mathcal{B}| \in \{512, 2048\}$ (the debiased approach uses two mini-batches of size 256 and 1024, respectively). The results are shown in Figure 13.

**Runtime and memory consumption.** Our current implementation uses the naive approach that requires *two* matrix-vector products with the GGN for computing one debiased update magnitude (see Appendix A.2 for details). As this would skew the runtime comparison, we refrain from providing concrete numbers here. However, as we describe in detail in Appendix A.2, it is possible to implement the debiased approach in a more efficient way that only requires one additional matrix-vector product for each CG direction to compute the debiased update magnitude. Using the debiased approach at half the mini-batch size, this brings the computational costs down to roughly the same level as the standard approach (assuming that the runtime of a GGN-vector product scales linearly with the mini-batch size).

As explained in detail in Appendix A.2, the memory overhead for the efficiently implemented debiased CG approach is two additional vectors of size $P$. For the experimental setting used for Figure 5, the number of parameters is $P = 1{,}387{,}108$, which corresponds to 5.55 MB in single precision. The computational overhead incurred by the debiased approach is thus about 11.1 MB.

### B.6 SECTION 6.2 (DEBIASED LAPLACE APPROXIMATION): FIGURE 6

Figure 6 shows a comparison of the vanilla model, the mini-batch K-FAC LA, the debiased version, and the full-batch LA. Here, we describe the experiment in more detail and present additional results, in particular on OOD data. For the derivation and the mathematical details of these approaches, see Appendix A.3. The experiment uses the fully trained ALL-CNN-C model on CIFAR-10 data from test problem (B) (see Appendix B.1).

**Experimental procedure.** The experimental procedure consists of three steps: The computation of the K-FAC curvature approximations, the evaluation of the corresponding predictive class probabilities, and the computation of the performance metrics.

1. **K-FAC curvature approximations.** We consider a log-equidistant grid of 13 prior precisions $\beta$ between $10^{-4}$ and $10^0$ and add $\beta = 10$.

   For each of those 14 values, we compute the K-FAC curvature approximation via BACK-PACK (Dangel et al., 2020) (see Appendix B.2) using (i) a single batch of size 64, 256, and 1024, (ii) the two-batch debiased LA version (details in Appendix A.3.3) at batch sizes 32, 128, and 512, and (iii) the full-batch LA. For the latter, we accumulate mini-batch K-FAC approximations over the entire training set. As an additional baseline, we consider (iv) the vanilla model (without LA), which is independent of the prior precision $\beta$. For approaches (i) and (ii), we repeat the experiment for 5 different mini-batches/mini-batch pairs.

2. **Evaluation of predictive class probabilities.** The first step in the evaluation of the performance metrics is the computation of the predictive uncertainty. For a given test dataset of size $N_\diamond$, these can be represented as a matrix $\boldsymbol{P} \in \mathbb{R}^{N_\diamond \times C}$, where $\boldsymbol{P}_{n,c}$ is the probability that the $n$-th sample belongs to class $c$, i.e. rows of $\boldsymbol{P}$ sum to 1. In all experiments, we draw $S = 40$ MC samples $\{\boldsymbol{\theta}^{(s)}\}_{s=1}^{S}$ from the weight posterior following the procedure in Appendix A.3.2. The predictive class probabilities are then obtained via Equation (18). In the case of the vanilla model without LA, the sum in Equation (18) collapses to a single term corresponding to the MAP model.

   The evaluation procedure above can be applied to arbitrary test datasets. We consider the training data, test data and CIFAR-10-C (i.e. OOD datasets at severity 1 to 5).

3. **Performance metrics.** We consider the following performance metrics:
   - **Accuracy.** The predictive classes are obtained from $\boldsymbol{P}$ by extracting the class with the highest probability. The accuracy is the relative number of correctly classified samples.

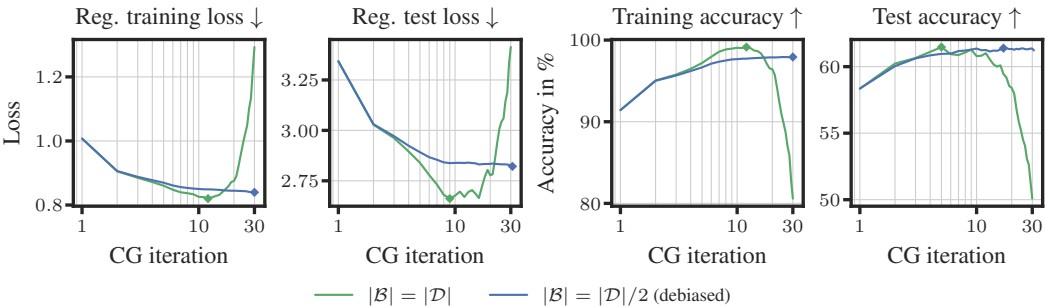

**Figure 12: Comparison of full-batch CG approaches.** We use the setting from Figure 5. The full-batch approach (shown as ——) applies standard CG to $q(\,\cdot\,;\mathcal{D})$ while the debiased approach (shown as ——) uses one half of the training data for the directions and the other half for the update magnitudes. The markers ◆ and ◆ are placed at peak performance. Surprisingly, the full-batch run diverges. The debiased CG run is stable.

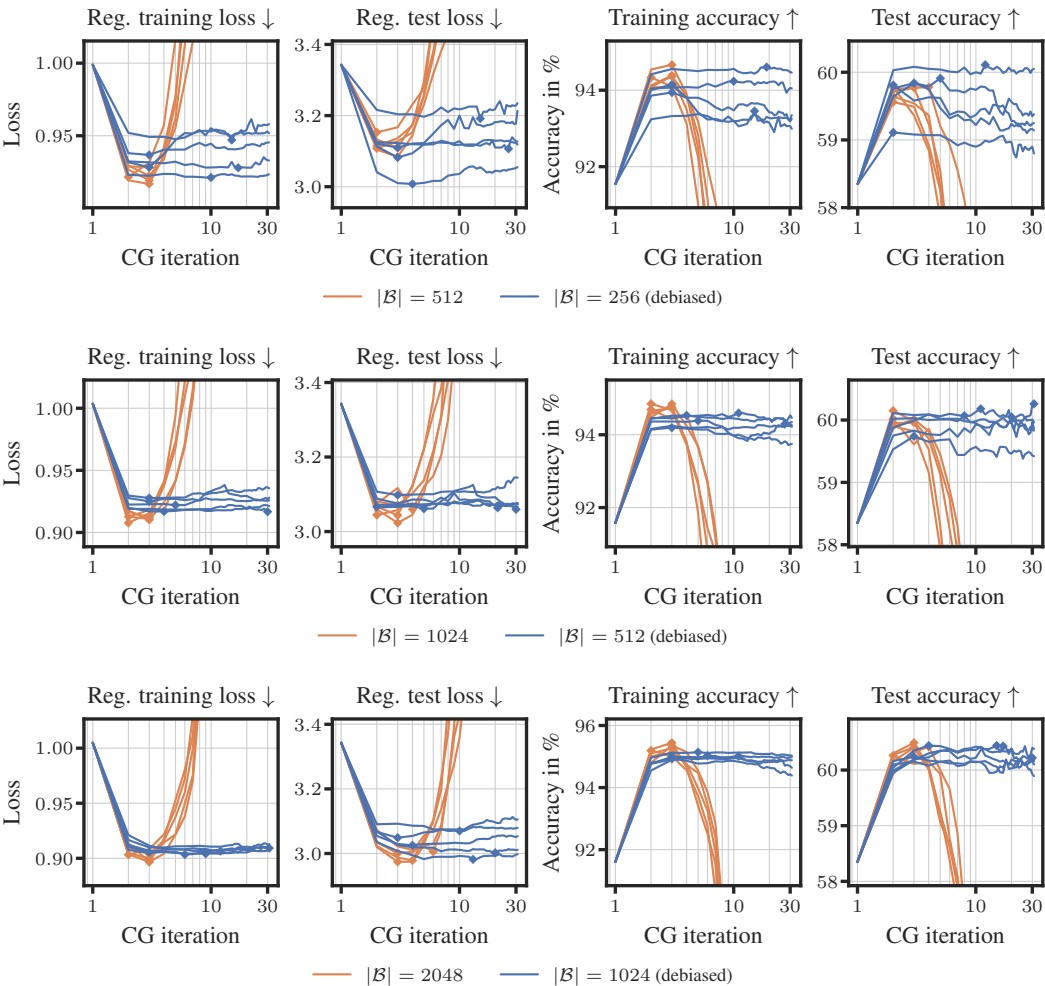

**Figure 13: Comparison of CG approaches at different batch sizes.** We use the same experimental setting as in Figure 5 but consider different mini-batch sizes $|\mathcal{B}| \in \{512, 1024, 2048\}$ (from *(Top)* to *(Bottom)*). The debiased approach uses two mini-batches of size 256, 512, and 1024, respectively. Across all mini-batch sizes, the single-batch CG runs diverge quickly while our debiased approach maintains stability.

- **Negative log-likelihood (NLL).** For each datum in the test set, we compute the negative log-probability of the true class. We then average over all samples in the test dataset. This coincides with the empirical risk from Equation (1), evaluated on the test dataset.

- **Expected calibration error (ECE).** The ECE is a measure of the calibration of the model's predictive probabilities. It groups the classification confidences (i.e. the maximum entry in each of $P$'s rows) into bins, and within these bins, compares the average confidence with the actual accuracy. We use `MulticlassCalibrationError` from `torchmetrics` (Detlefsen et al., 2022).

- **AUROC.** For the OOD datasets, we provide the Area Under the Receiver Operating Characteristic curve (AUROC). Our goal is to distinguish in-distribution (ID) and out-of-distribution (OOD) samples using the model's uncertainty. We use the entropy of the predictive distribution $p(\boldsymbol{y}_\diamond \mid \boldsymbol{x}_\diamond, \mathbb{D})$ as our uncertainty estimate $u(\boldsymbol{x}_\diamond) \in \mathbb{R}$ for input $\boldsymbol{x}_\diamond$. The ground-truth binary labels are the ID/OOD indicators of the test inputs. Several thresholds $\xi$ could be established to turn the scalar uncertainty estimates into binary predictions by using $\mathbf{1}_{\{u(\boldsymbol{x}_\diamond) > \xi\}}$. Instead of choosing a particular threshold and evaluating its accuracy, the AUROC metric directly evaluates $u(\cdot)$ by measuring the area under the plot of the true positive rate (TPR) against the false positive rate (FPR) for all possible thresholds.

**Additional experimental details.**

- **Uncertainty over weights but not biases.** As the prior acts only on the weights of the network but not its biases (see Appendix B.1), we only consider the uncertainty over the weights in the LA. This slightly reduces the size of the covariance matrix as it excludes the bias parameters.

- **Single vs. double precision.** Although the K-FAC factors are positive semi-definite by construction, their numerical eigenvalues can be negative due to numerical inaccuracies. To balance precision and computational cost, we use double precision for all computations until drawing the MC samples and single precision afterwards.

**Results.** The results are presented in Figures 6, 14 and 15. For the single-batch and debiased approach (where we use 5 mini-batches/mini-batch pairs each), we report the average performance as a dot and the min/max as a vertical line. The performance of the vanilla model is shown as a horizontal line as its performance does not depend on $\beta$, as explained above.

Across all datasets and performance metrics, the debiased LA mimics the behavior of the full-batch approach much better than the single-batch LA (although both approaches use a comparable amount of computational resources). In particular, for small prior precisions, where the covariance matrix relies almost exclusively on the K-FAC curvature information, the debiased LA maintains a good performance in contrast to the single-batch approach.

**Additional results.**

- **RESNET-50 on IMAGENET.** To showcase the scalability of our debiased LA approach, we repeat the experiment for a RESNET-50 model on the IMAGENET dataset (test problem (E) in Appendix B.1). The results are shown in Figure 16. Again, the debiased approach behaves similarly to the full-batch LA and maintains stability over the entire spectrum of prior precisions.

- **VIT LITTLE on IMAGENET.** Similar to the previous experiment, we also apply our LA debiasing approach to a VIT LITTLE model on the IMAGENET dataset (test problem (F) in Appendix B.1) to ensure that it is beneficial for other architectures as well. The results are shown in Figure 17. The debiased approach mimics the calibration behavior of the full-batch LA remarkably well and remains stable for lower prior precision values as well where the single-batch approach behaves erratically.

**Runtime analysis (computational overhead of debiasing).** We claim in Section 6.2 that, by using the debiased approach at half the mini-batch size that is used by the single-batch approach, we obtain a fair comparison (more details in Section 4.2.3). Here, we substantiate this claim by providing a detailed runtime comparison of the biased, debiased, and full-batch LA approaches on CIFAR-10.

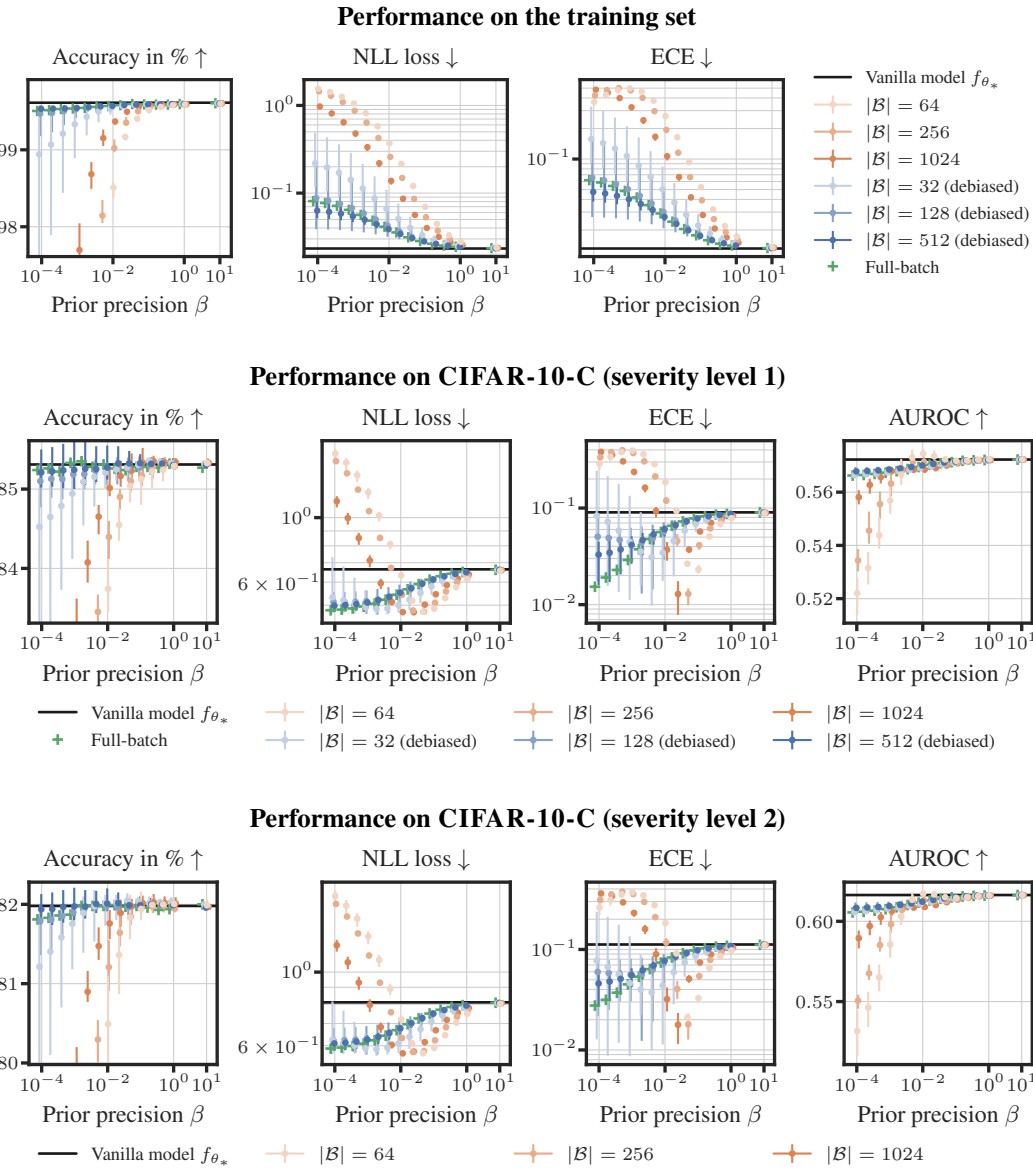

**Figure 14: Debiased LA mimics the full-batch LA very well.** The experimental setting is the same as in Figure 6, but we report results on additional datasets (training set and OOD datasets at severity level 1 and 2). For the OOD datasets, we report the AUROC metric in addition to the accuracy, NLL and ECE. In contrast to the single-batch approach, the debiased version mimics the behavior of the full-batch approach very well over the entire range of prior precisions and across datasets.

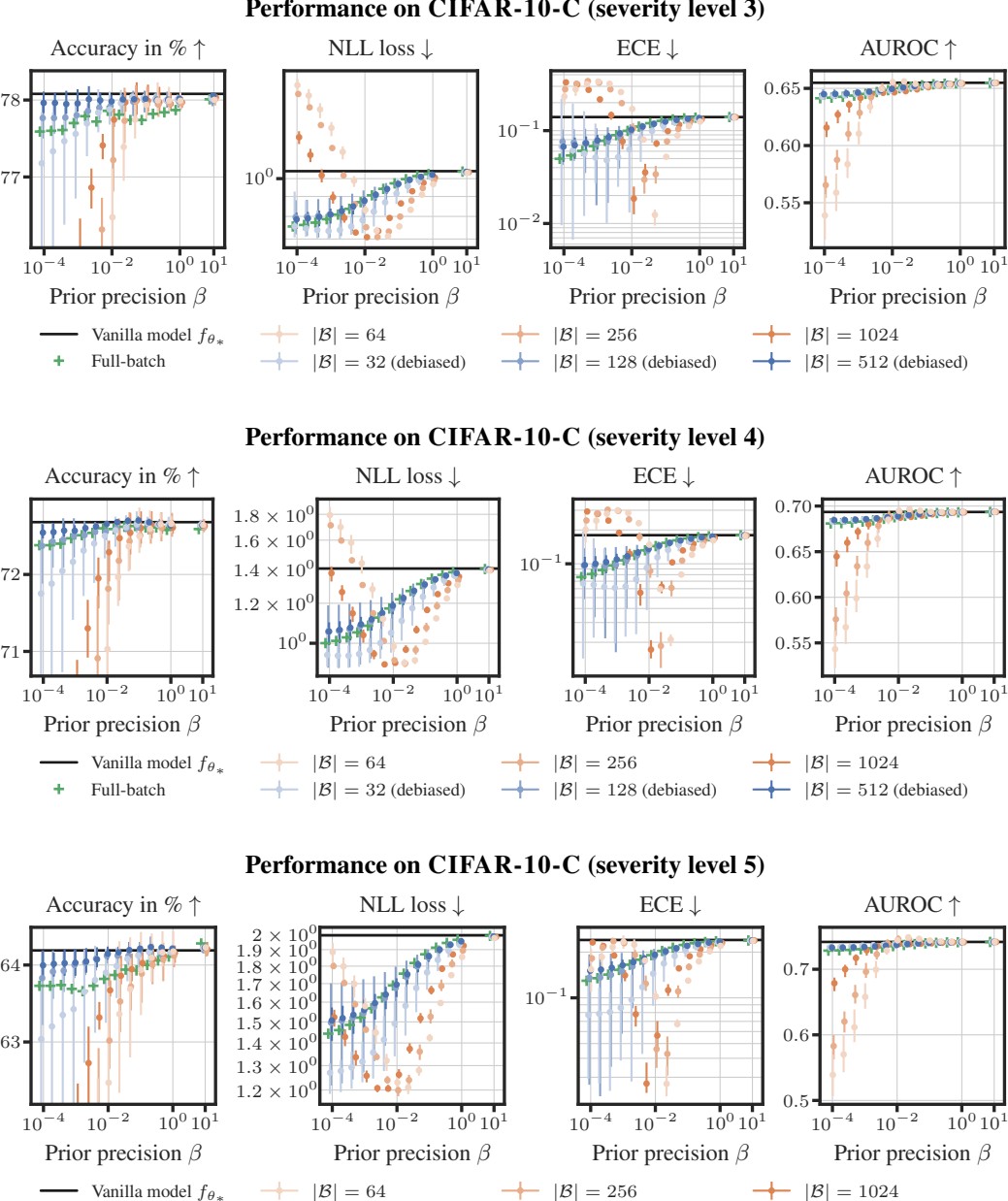

**Figure 15: Debiased LA mimics the full-batch LA very well.** The experimental setting is the same as in Figure 6, but we report results on additional datasets (OOD datasets at severity level 3, 4, and 5). We report the AUROC metric in addition to the accuracy, NLL, and ECE. In contrast to the single-batch approach, the debiased version mimics the behavior of the full-batch approach very well over the entire range of prior precisions and across datasets.

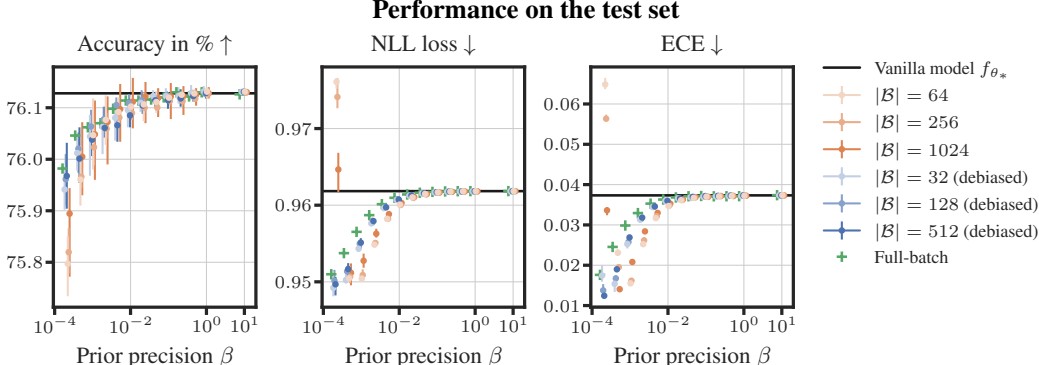

**Figure 16: Our approach scales to RESNET-50 on IMAGENET.** The experimental setting is the same as in Figure 6, but we use test problem (E) (see Appendix B.1) (RESNET-50 on IMAGENET). The results are consistent with the findings on CIFAR-100: The debiased approach behaves similarly to the full-batch LA and maintains stability even for small prior precisions.

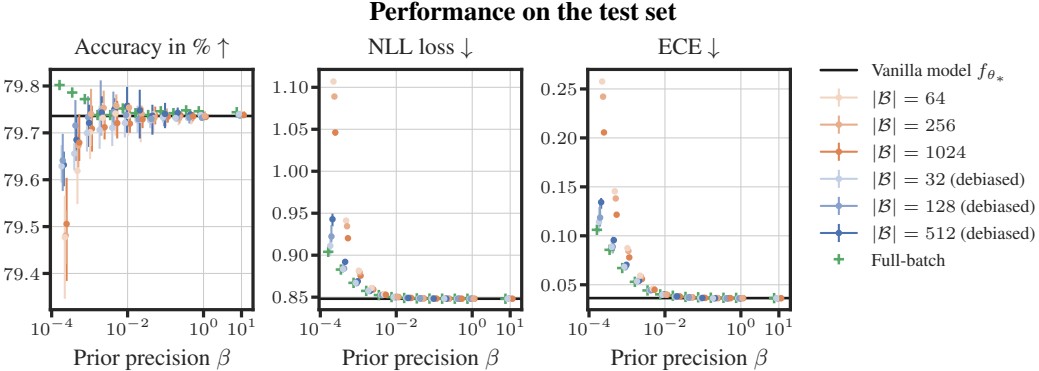

**Figure 17: Curvature biases are also present for the VIT LITTLE architecture on IMAGENET.** The experimental setting is the same as in Figure 6 and Figure 16, but we use test problem (F) (see Appendix B.1) (VIT LITTLE on IMAGENET). Again, the debiased approach behaves similarly to the full-batch LA and maintains stability even for small prior precisions (where the covariance is almost exclusively based on the K-FAC curvature information).

**Results (runtimes).** We report the runtimes of the biased (☐), debiased (☐), and full-batch (☐) K-FAC Laplace approximation schemes in Figure 18. We find that (i) both biased and debiased approaches are orders of magnitude cheaper than the full-batch LA, and (ii) the debiased approach introduces only negligible overhead and can, sometimes, even improve upon the runtime of the biased approach.

**Memory consumption.** The ALL-CNN-C model we use for the LA experiment in Section 6.2 has $P = 1,368,480$ trainable parameters (this amounts to 10.95 MB of memory in double precision). The K-FAC approximation requires storing 11,483,965 (91.87 MB) numbers to represent all Kronecker factors. The memory of a K-FAC approximation is thus roughly equivalent to that of 8.4 models. As the debiased approach is based on *two* such approximations in the case of a naive implementation, the overhead of debiasing is another 91.87 MB of memory. However, this can be significantly improved by building up the K-FAC approximations block by block: Having two corresponding blocks available (based on two different mini-batches) already suffices to compute the respective debiased block (see Appendix A.3.3). This way, we only have two blocks (represented by their Kronecker factors) in memory at the same time (instead of two entire K-FAC approximations) which greatly reduces the memory overhead of debiasing, down to the largest block. For the ALL-CNN-C model, this is two Kronecker factors of sizes $192 \times 192$ and $1728 \times 1728$ amounting to 3,022,848 numbers in total, i.e. the equivalent of 2.20 models or 24.18 MB in double precision. The statement that the debiased approach roughly doubles the memory consumption is thus a worst-case scenario.

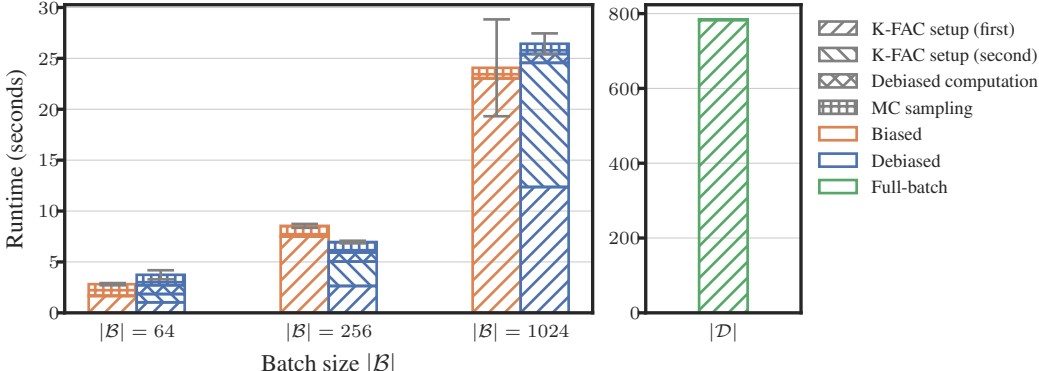

**Figure 18: The computational overhead of debiasing is negligible for Laplace approximations.** We follow the experimental setup of Section 6.2 and provide a detailed runtime comparison of the biased (□), debiased (□), and full-batch (□) K-FAC Laplace approximation schemes on CIFAR-10. The total runtime includes the construction of K-FAC's Kronecker factors (on one or two mini-batches for the biased and debiased approach, respectively), the debiasing computations (see Appendix A.3.3), and the Monte Carlo sampling of $S = 40$ weights. All reported runtimes are averages over 5 runs. The error bars for the total runtime cover one standard deviation. The results show that the debiased approach introduces a negligible overhead. Compared to the full-batch variant, both mini-batch approaches are orders of magnitude faster. Note that the evaluation (i.e., the computation of the predictive uncertainty) is not included in the comparison, as the time requirements are the same for all approaches; however, this step requires the most computational resources.

## B.7 ADDITIONAL EXPERIMENT: BIASES FOR A WIDERESNET 40-4 MODEL ARCHITECTURE

**Experimental setting and results.** Here, we extend our analysis from Section 3.1 (details in Appendix B.4) to test problem (C) (see Appendix B.1): A WIDE RESNET model on CIFAR-100. We use the GGN curvature proxy $H_\mathcal{B} \leftarrow G_\mathcal{B} + \beta I$ at batch size 256 and compute the directional slopes and curvatures along the top 100 eigenvectors of $H_\mathcal{B}$. The results are shown in Figure 19. Along the first few top-curvature directions, the curvature biases are even larger for this test problem than for the ALL-CNN-C model on CIFAR-100, but then they decay more quickly.

## B.8 ADDITIONAL EXPERIMENT: K-FAC AND THE DEPENDENCE OF THE BIASES ON MINI-BATCH SIZE

**Experimental setting.** Here, we extend our analysis of test problem (A) (see Appendix B.1) from Section 3.1 to K-FAC and investigate the impact of the mini-batch size on the curvature biases.

**Results & discussion.** Figure 20 shows the directional curvatures of the K-FAC approximation for four different mini-batch sizes. When we compute the eigenvectors and directional curvatures on the same mini-batch, we observe a systematic curvature bias that decreases with increasing mini-batch size. There are two phenomena at play: With increasing mini-batch size, (i) the green crosses move upwards and (ii) the orange dots move downwards. Our intuition for this is as follows: With increasing mini-batch size, the eigenvectors become more meaningful such that, on other data, they also exhibit large curvature—this explains (i). Similarly, with increasing mini-batch size, it gets harder to find directions of *extreme* curvature as these directions have to exhibit large curvature on *all* data points within the mini-batch—this explains (ii).

## B.9 ADDITIONAL EXPERIMENT: DEVELOPMENT OF THE BIASES OVER THE COURSE OF TRAINING

**Experimental setting and results.** Here, we investigate how the biases in the slope and curvature develop over the course of training for test problem (A) (see Appendix B.1). We use the same procedure as in Appendix B.4 but evaluate the biases at 10 different checkpoints during training (spread log-equidistantly between the first and last epoch). At each checkpoint, we draw 5 mini-

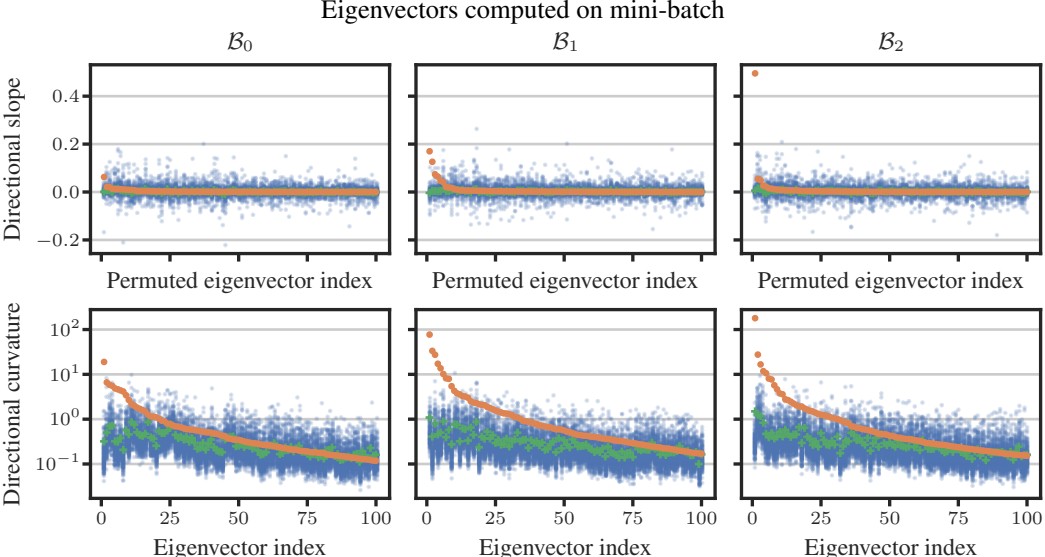

**Figure 19: Directional slopes and curvatures are biased.** The experimental setting is similar to Figure 2, but we use test problem (C) (WIDE RESNET on CIFAR-100) with the GGN curvature proxy $H_{\mathcal{B}} \leftarrow G_{\mathcal{B}} + \beta I$ at batch size 256. For the top panel, we switch the order and sign of the eigenvectors such that the orange dots are all above zero and in descending order. There is a strong, systematic bias, particularly in the curvature: Computing the eigenvectors and directional curvatures on the same data results in over-estimation that is very pronounced along the first few eigenvectors but then decays more quickly than in test problem (A).

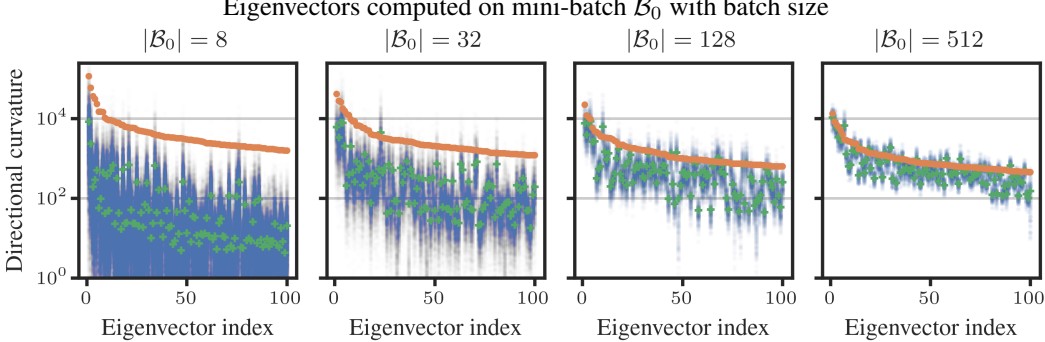

**Figure 20: Directional curvatures with K-FAC.** We use the CIFAR-100 dataset with the fully trained ALL-CNN-C model. For each mini-batch size $\in \{8, 32, 128, 512\}$, we draw one mini-batch and compute the top 100 eigenvectors $u_1, \ldots, u_{100}$ of the K-FAC matrix $K_{\mathcal{B}_0}$. We show the directional curvatures $u_p^\top K_{\mathcal{B}_m} u_p$, $p \in \{1, \ldots, 100\}$ on the same mini-batch ($m = 0$) as ●, all other mini-batches in the training set ($m \in \{1, 2, \ldots\}$) as ● and their average as ✚. Similar to Figure 2, we observe a systematic bias in the curvature. The bias decreases with increasing mini-batch size.

batches, compute the top 100 eigenvectors of the corresponding GGN-based quadratics, and finally evaluate the *relative* errors in the directional slopes and curvatures (where the ground truth is the full-batch quadratic's slope/curvature). For each mini-batch, this distribution of 100 relative curvature and slope biases is represented as a dot (at the mean relative bias) and a vertical line ranging from the 25% to the 75% percentile in Figure 21. While the biases in the slope remain relatively stable over the course of training, the biases in the curvature increase over more than 3 orders of magnitude up to a relative error of order 10 in epoch 349 (which is consistent with the results in Figure 2). This suggests that the eigenspaces of different curvature matrices become more and more misaligned as training progresses steadily increasing the need for effective debiasing strategies.

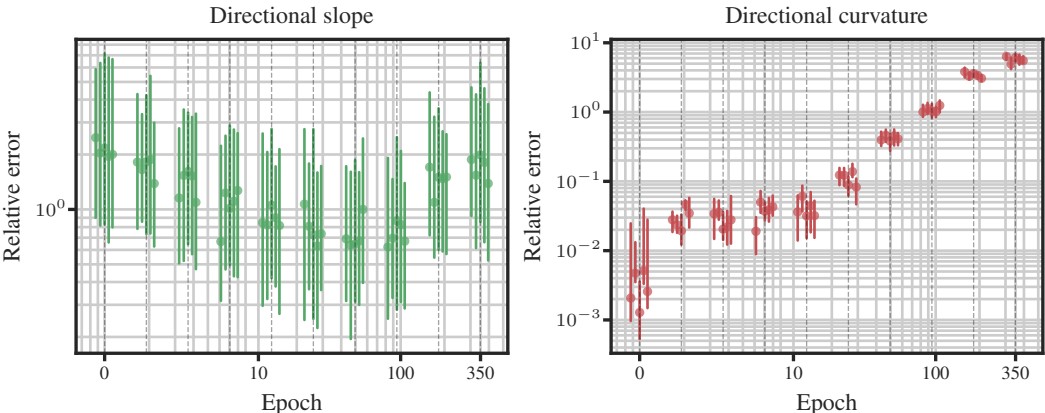

**Figure 21: Curvature biases increase over the course of training.** We evaluate the relative slope and curvature biases at different checkpoints during training for the ALL-CNN-C model on CIFAR-100. 5 mini-batches are drawn per checkpoint. For each mini-batch, the relative biases are represented as a dot ● (at the mean relative bias) and a vertical line — ranging from the 25% to the 75% percentile. While the biases in the slope remain relatively stable over the course of training, the biases in the curvature increase (over more than 3 orders of magnitude).

### B.10 ADDITIONAL EXPERIMENT: THE CURVATURE BIAS INCREASES WITH $P$

In high-dimensional spaces, it becomes increasingly unlikely that random vectors are aligned. While the eigenspaces of the curvature matrices are not completely random, they are subject to noise. It is thus conceivable that their overlap decreases as the number of parameters $P$ increases. If this hypothesis is true, the curvature biases should become more pronounced in large models.

**Experimental procedure & results.** To test this hypothesis, we use test problem (D) that implements a simple convolutional neural network with variable width and depth (for details, see Appendix B.1). For each fully trained model, we evaluate the *relative* error between $\partial^2_{\boldsymbol{u}_p} q(\boldsymbol{\theta}_\star; \mathcal{B})$ and $\partial^2_{\boldsymbol{u}_p} q(\boldsymbol{\theta}_\star; \mathcal{D})$ for the $\boldsymbol{G}_\mathcal{B}$'s top 100 eigenvectors at batch size 128. This procedure is repeated for 5 different mini-batches for each of the 9 models, resulting in a total of 45 error distributions (see Figure 22). The results confirm our hypothesis: The relative errors tend to increase with the number of parameters $P$. In the massively overparameterized regime, the biases might thus become even more relevant and effective debiasing strategies are needed.

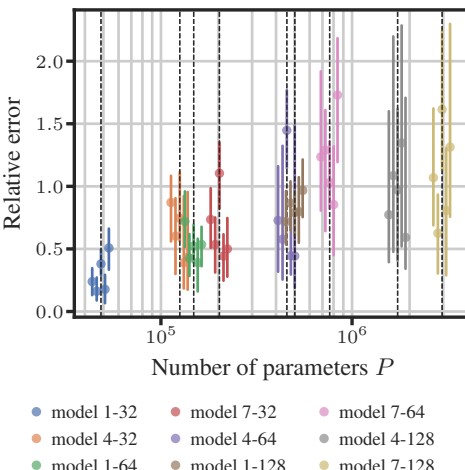

| | |
|---|---|
| ● model 1-32 | ● model 7-32 ● model 7-64 |
| ● model 4-32 | ● model 4-64 ● model 4-128 |
| ● model 1-64 | ● model 1-128 ● model 7-128 |

**Figure 22: Increasing curvature biases with $P$.** We train 9 convolutional neural networks with different widths and depths for 100 epochs on the CIFAR-10 dataset using ADAM with standard hyperparameters. For the fully trained model, we evaluate the relative error $|\partial^2_{\boldsymbol{u}_p} q(\boldsymbol{\theta}_\star; \mathcal{B}) - \partial^2_{\boldsymbol{u}_p} q(\boldsymbol{\theta}_\star; \mathcal{D})| \cdot |\partial^2_{\boldsymbol{u}_p} q(\boldsymbol{\theta}_\star; \mathcal{D})|^{-1}$ for $\boldsymbol{G}_\mathcal{B}$'s top 100 eigenvectors at batch size $|\mathcal{B}| = 128$. 5 different mini-batches are used per model resulting in a total of 45 error distributions (each consisting of 100 numbers). These distributions are represented by their median (as a dot ●) and the 25% and 75% percentiles (as a line segment —) $P$. The 5 distributions for each model are slightly spread along the $x$-axis for better visibility. The experiment confirms our hypothesis: The relative errors tend to increase with the number of parameters $P$.

