# OpenReview forum: "Debiasing Mini-Batch Quadratics for Applications in Deep Learning"
_ICLR.cc/2025/Conference — ICLR 2025 Poster_

### Official Review · Reviewer_9wSH · 2024-11-01

**Soundness:** 3
**Presentation:** 3
**Contribution:** 3
**Rating:** 8
**Confidence:** 3

**Summary:**

Often one would like to obtain a second-order Taylor approximation to a deep learning objective based on a minibatch.  This paper shows that the use of a minibatch results in a strong overestimation of the top Hessian eigenvalues.  The basic problem is that the directions of highest curvature for a minibatch tend to slightly 'overfit' (my words) that minibatch, and do not generalize to other minibatches or to the full-batch objective.  This paper shows that this bias can be corrected with a scheme that uses two separate minibatches - one to compute the top eigenvectors, and another to compute the directional curvature along these eigenvectors.  The paper discusses applications in both optimization (conjugate gradient method for computing a newton step), and uncertainty quantification (laplace's method)

**Strengths:**

The paper offers a simple solution to a problem of interest to multiple areas of deep learning.

**Weaknesses:**

In the application to optimization, the paper does not actually carry out an end-to-end optimization run with a second-order method using the proposed conjugate gradient scheme; instead, the paper just uses the proposed scheme to compute a single newton step.

**Questions:**

Regarding the motivation in section 3.1 -- just because the gradient lives in the top subspaces does _not_ mean that all the optimization progress occurs in that space.  To the contrary, the evidence suggests that the vast majority of the motion along the top Hessian eigenvectors is oscillatory, and 'cancels out.'  Long-term loss reduction is due to the components of the gradient that lie along the low-curvature Hessian eigenvectors.  For example, see "Does SGD really happen in tiny subspaces?" by Minhak Song, Kwangjun Ahn, Chulhee Yun: https://arxiv.org/abs/2405.16002.

I don't think this is fatal for the paper, since you are basically only using your improved estimate of the the top Hessian eigenvalues in order to properly compute the conjugate gradient _step size_, which will depend on the top Hessian eigenvalues.  But if you actually tried to optimize only along those eigenvectors, it wouldn't work (as the linked paper shows).

---

> ### Author Response · Authors · 2024-11-21
>
> Dear Reviewer 9wSH,
>
> Thanks a lot for your thoughtful feedback! We are glad that you think our paper offers a
> solution *to a problem of interest to multiple areas of deep learning*. In the
> following, we would like to address your concerns and questions.
>
> > In the application to optimization, the paper does not actually carry out an
> > end-to-end optimization run with a second-order method using the proposed conjugate
> > gradient scheme; instead, the paper just uses the proposed scheme to compute a single
> > newton step.
>
> You are right that we only use the proposed scheme to compute a single Newton step for an
> already well-trained model. However, we believe that this is a practically
> relevant scenario. Imagine that you have trained a model with a first-order optimizer
> and want to quantify the model's uncertainty via the Laplace approximation. The Laplace
> approximation requires that the model has converged to a local minimum such that the
> gradient vanishes. In practice, however, this assumption is often not fulfilled.
> Applying a single Newton step might thus be beneficial, as it moves the model towards
> the minimum of the quadratic approximation such that the assumptions of the Laplace
> approximation are more likely to be met.
>
> Our work presents approaches for obtaining more meaningful curvature
> estimates. However, this does not directly translate into a new optimizer. For example,
> curvature-based optimizers need to account for the fact that the loss is not exactly
> quadratic. This is often addressed by using curvature damping (resulting in a
> trust-region method) in combination with a line search. These components need to be
> carefully designed and tuned, which, we believe, would amount to a paper on its own.
>
>
> > Regarding the motivation in section 3.1 -- just because the gradient lives in the top
> > subspaces does not mean that all the optimization progress occurs in that space. To
> > the contrary, the evidence suggests that the vast majority of the motion along the top
> > Hessian eigenvectors is oscillatory, and 'cancels out.' Long-term loss reduction is
> > due to the components of the gradient that lie along the low-curvature Hessian
> > eigenvectors. For example, see "Does SGD really happen in tiny subspaces?" by Minhak
> > Song, Kwangjun Ahn, Chulhee Yun: https://arxiv.org/abs/2405.16002.
>
> > I don't think this is fatal for the paper, since you are basically only using your
> > improved estimate of the the top Hessian eigenvalues in order to properly compute the
> > conjugate gradient step size, which will depend on the top Hessian eigenvalues. But if
> > you actually tried to optimize only along those eigenvectors, it wouldn't work (as the
> > linked paper shows).
>
> Thank you for the reference - we were not aware of this paper. Our debiasing approach,
> in principle, benefits the *entire* parameter space, not just its top-$k$ subspace: It
> effectively corrects the curvature overestimation in the top-$k$ subspace *and* the
> underestimation in the low-curvature subspace. Updates within the low-curvature subspace
> could therefore also benefit from our debiasing strategies.
>
> Thanks again for your positive review! We hope that you will argue in favor of our paper
> during the reviewer discussion phase. If you have any further questions, please let us
> know.

---

### Official Review · Reviewer_ddR4 · 2024-11-01

**Soundness:** 2
**Presentation:** 2
**Contribution:** 2
**Rating:** 3
**Confidence:** 3

**Summary:**

This paper demonstrates a systematic bias in stochastic 2nd order approximation of empirical risk for neural networks. The study start from an observation showing that a particular curvature structure on minibatch Hessian is biased towards underestimation. Then, authors proposes a debiasing method.

**Strengths:**

- A computationally-cheap estimate of 2nd order information is an important topic for optimization
- Ample experimental observations
- Designing computational-cheap experiments to study 2nd order derivative of training loss of neural networks

**Weaknesses:**

I believe the notion of unbiased estimate is not well defined and studied. Define the Hessian associate with full-batch training loss as  $$H(x) = \frac{1}{n} \sum_{i=1}^n \nabla^2 f_i(x)$$. Similarly, we define the unbiased estimate of this matrix as $$H'(x) = \frac{1}{m} \sum_{k=1}^m \nabla^2 f_{i_k}(x)$$ where $i_k$ are uniformly drawn from $\{1,\dots, n\}$.
For each fixed vector $d$, we have $$E d^\top H' d^\top = d^\top H d$$. Thus, the **directional curvature**, defined in the paper, is unbiased. But, why this paper observe a bias in the estimate? The main issue is that they consider random directions $d$ depending on minibatches. I do not know why we need to choose random $d$s?

I believe the quantity of interest is not well defined and motivated, here. It is easy to provide an unbiased estimation of  $q$, directional curvature and directional slop. But, the authors want to estimating another quality that they do not exactly defined and it is not clear how its estimation connect with Newton's method or Laplace approximation.

If you want to estimate the maximum/minimum eigenvalue of the Hessian or even its condition number, the empirical Hessian provides an asymptotically unbiased estimate (see for example https://arxiv.org/pdf/1912.10754).

**Questions:**

- I recommend to replace $u_1$ and $u_2$ in Figure 1 by leading eigenvectors of full-batch Hessian to better grasp my comment in weaknesses. I expect to see that increasing batch size will lead to a better approximation of the curvature.
- What is exactly the quantity that you want to estimate and why it is important to estimate? Directional curvature on which random directions?
- What do you exactly mean by bias? To prove an estimate has bias, we need to take average. But, experiments in Figure 1 are not computing an average. How can I conclude from these experiments that the estimate is biased?

---

> ### Author Response · Authors · 2024-11-21
>
> Dear Reviewer ddR4,
>
> Thank you for your time and effort in reviewing our paper. We appreciate your feedback
> and points of criticisms. However, we believe there may have been some misunderstandings
> that affected the interpretation of our work. We hope to clarify these points in the
> following.
>
> > I believe the notion of unbiased estimate is not well defined and studied. Define the
> > Hessian associate with full-batch training loss as $H(x) = \frac{1}{n} \sum_{i=1}^n
> > \nabla^2 f_i(x)$. Similarly, we define the unbiased estimate of this matrix as $H'(x)
> > = \frac{1}{m} \sum_{k=1}^m \nabla^2 f_{i_k}(x)$ where $i_k$ are uniformly drawn from
> > $1, ..., n$. For each fixed vector $d$, we have $E d^\top H' d = d^\top H d$. Thus,
> > the directional curvature, defined in the paper, is unbiased.
>
> You are right that the Monte Carlo estimate $H'$ yields an unbiased estimate of $H$,
> i.e. $\mathbb{E}[H'] = H$. However, $\mathbb{E}[d^\top H' d]$ does not necessarily equal
> to $d^\top H d$ $-$ this only holds if $d$ is a vector of constants, i.e. *independent* of
> $H'$. For example, when $d$ is an eigenvector of $H'$ (and is, therefore, also a stochastic
> quantity), this assumption is violated and the directional curvature is thus not
> necessarily unbiased.
>
> In practice, we observe that this bias exists (details in Section 3.1). Let $(u_i(H'),
> \lambda_i(H'))$ denote the $i$-th eigenpair of $H'$. We observe that
> $\mathbb{E}[\lambda_i(H')] = \mathbb{E}[u_i(H')^\top H' \ u_i(H')] \neq
> \mathbb{E}[u_i(H')^\top H \ u_i(H')]$. In particular, we observe that the leading
> eigenvalues sytematically *overestimate* the true underlying directional curvature, e.g.
> $\mathbb{E}[\lambda_1(H')] > \mathbb{E}[u_1(H')^\top H \ u_1(H')]$ for the leading
> eigenpair.
>
> The bottom panel of Figure 2 illustrates this curvature bias. The top-$100$ eigenvalues
> $\lambda_i(H')$ are shown as orange dots; the green crosses represent the true
> directional curvatures $u_i(H')^\top H \ u_i(H')$. We use "bias" to refer to the
> systematic gap between them. The fact that we observe these biases over three different
> mini-batches suggests that this phenomenon is not due to chance but appears
> systematically. We provide further empirical evidence for this bias in Sections B.4 to
> B.10 and a theoretical explanation in Section 3.2.
>
> We hope it is clear now that your theoretical argument does not contradict our
> empirical evaluation or theoretical explanations. If you have follow-up questions about
> this, please let us know!
>
> > But, why this paper observe a bias in the estimate? The main issue is that they
> > consider random directions depending on minibatches. I do not know why we need to
> > choose random $d$ s?
>
> As you point out, both kinds of directions we consider in the paper - the top/bottom
> eigenvectors of $H'$ and the CG search directions - depend on the mini-batch. You
> rightfully ask why we consider these specific directions for our analyses. The reason is
> that these are the directions that applications (like Newton-type optimization via CG or
> the Laplace approximation) "see"/have access to. And the geometric information these
> methods collect *along* these directions (i.e. directional slopes/curvatures) affects
> their performance (details in Section 4.1).
>
> Consider the method of conjugate gradients (CG) as an example. The computational cost of
> each CG iteration increases with the mini-batch size. CG is thus typically applied to a
> *mini-batch* quadratic (that is, in your notation, based on $H'$, not $H$) to keep those
> costs feasible. In each step, CG computes a direction $d$ and an update magnitude; and
> the latter is determined by the ratio of the directional slope and curvature along $d$
> (details in Section 2.2). Now, if the directional slope and curvature are biased, i.e.
> they are *not representative* of the full-batch quadratic's slope and curvature, the
> update magnitude is biased as well. Depending on the severity of this bias, the
> algorithm will make misinformed updates and potentially become unstable. In order to
> build an efficient and robust optimization algorithm, it is thus crucial to know and
> understand these biases along the CG directions of the mini-batch quadratic.
>
> Similarly, the curvature biases along $H'$'s eigenvectors affect the uncertainty
> quantification via the Laplace approximation (details in Section 4.1).

---

> ### Author Response · Authors · 2024-11-21
>
> > I believe the quantity of interest is not well defined and motivated, here. It is easy
> > to provide an unbiased estimation of $q$, directional curvature and directional slop.
> > But, the authors want to estimating another quality that they do not exactly defined
> > and it is not clear how its estimation connect with Newton's method or Laplace
> > approximation.
>
> The quantities you mention ($q$, directional slope and directional curvature) are
> defined in Section 2.1: $q$ is defined in Equation (2) (the mini-batch counterpart is
> defined below) and the directional curvature and slope are defined in the unnumbered
> equation on the bottom of page 2. If you can point us to specific quantities that you
> believe are not well defined or motivated, we would be happy to provide further
> explanations (and make corresponding amendments to the text).
>
> > If you want to estimate the maximum/minimum eigenvalue of the Hessian or even its
> > condition number, the empirical Hessian provides an asymptotically unbiased estimate
> > (see for example https://arxiv.org/pdf/1912.10754).
>
> Thanks for the pointer. We want to stress, however, that what you describe is not the
> purpose of our paper. Our work provides insights and makes suggestions on how to (i)
> compute effective and robust steps in stochastic second-order optimizers and (ii)
> reliably quantify uncertainties via the Laplace approximation based on a *mini-batch* of
> data. For both applications, access to the largest and smallest eigenvalue of the
> empirical Hessian (or an approximation thereof) is insufficient.
>
> **Next, we address your questions.**
>
> > I recommend to replace $u_1$ and $u_2$ in Figure 1 by leading eigenvectors of
> > full-batch Hessian to better grasp my comment in weaknesses. I expect to see that
> > increasing batch size will lead to a better approximation of the curvature.
>
> Increasing the batch size will indeed improve the approximation of the curvature. But
> the associated computational costs increase linearly with mini-batch size. The purpose
> of Figure 1 is to hint that the curvature information along the eigenvectors of the
> *mini-batch* curvature matrix (this is what applications have access to) is *not*
> representative of the *full-batch* curvature. Our experiments demonstrate that our
> proposed debiasing strategies remove the problem *without* increasing computational
> costs.
>
> > What is exactly the quantity that you want to estimate and why it is important to
> > estimate? Directional curvature on which random directions?
>
> The quantity of interest depends on the application. For second-order optimization, the
> goal is to compute an estimate of the Newton step (details in Section 2.2). For the
> Laplace approximation, we aim to compute the covariance matrix of the approximate
> posterior distribution (see Equation (4) in Section 2.3). Ideally, we would compute
> these quantities based on the *full-batch* quadratic. However, as the associated
> computational costs increase with the mini-batch size, we typically rely on mini-batch
> approximations. The goal of our work is to compute these quantities (Newton step or
> covariance matrix) based on a *mini-batch quadratic*. As we show in the paper, doing so
> in a naive way leads to detrimental optimization steps and a deformed posterior
> covariance (see Section 4.1). Our approach to address this is to commit to the
> "imperfect" mini-batch directions along which the algorithms operate, but compute a
> representative estimate of the full-batch quadratic's shape along these directions based
> on a second mini-batch (details in Section 4).
>
> > What do you exactly mean by bias? To prove an estimate has bias, we need to take
> > average. But, experiments in Figure 1 are not computing an average. How can I conclude
> > from these experiments that the estimate is biased?
>
> By bias, we mean that for the set of directions along which the algorithms (CG or
> Laplace) operate, the directional slope and curvature of the mini-batch quadratic are
> *systematically* different from those of the full-batch quadratic.
>
> Figure 1 is not meant as a proof of this bias, but as a first hint that this bias
> exists. It shows a two-dimensional subspace in which the mini-batch quadratic looks
> quite different from the full-batch quadratic. As this observation is consistent
> across five different mini-batches, this indicates that there might be a systematic
> error. Sections 3.1 and B.4 to B.10 provide further empirical evidence for this bias.
>
> Again, thanks for your feedback. We hope that our clarifications help resolve the
> misunderstandings. If our responses have addressed your concerns, we kindly ask that you
> consider re-evaluating the paper in light of these clarifications. If you have any
> follow-up questions, please let us know.

---

> ### Comment · Reviewer_ddR4 · 2024-11-24
> **Following up**
>
> Thank you so much for your response. I have follow up questions.
>
> > Increasing the batch size will indeed improve the approximation of the curvature. But the associated computational costs increase linearly with mini-batch size.
>
> When we talked about biased or unbiased estimates, we hope for having a correct estimation in average over many random draws. We are not concerned about the computation. Many estimates become asymptotically unbiased, namely they are almost unbiased on large sample sizes. Have you check that the estimate is asymptotically unbiased?
>
> > You are right that the Monte Carlo estimate ...
>
> Directional curvature is not well defined for random directions $d'$ dependent on batchsize. Even, I do not know how to use your notations to express it. Curvature on random direction $d$? Is it a stochastic curvature averaged over random directions?
>
> I am not convinced that we need to replace the leading eigenvectors of Hessian with those from batchsizes. I can imagine that we can use the leading eigenvectors of full batchsize for only theoretical analysis and provide good bounds for quantity of interests such as directional curvature with $d$ is the leading eigenvector of the full-batch Hessian. I am afraid that the bias is mainly originated from a wrong perspective towards the problem.

---

> > ### Author Response · Authors · 2024-11-25
> >
> > Dear Reviewer ddR4,
> >
> > Thank you for your follow-up questions.
> >
> > > When we talked about biased or unbiased estimates, we hope for having a correct
> > > estimation in average over many random draws. We are not concerned about the
> > > computation.
> >
> > What one is concerned about or not is an aspect of scientific debate. We argue that
> > practitioners should indeed be concerned that the algorithms they use do not use the
> > "right" quantities. As the reactions of the other reviewers indicate, this is a widely
> > shared concern.
> >
> > To be hopefully more formally clear: We define bias as follows. First, we compute a
> > curvature matrix $H'$ on a mini-batch. We use this random variable to define an
> > estimator $d_i^T H' d_i$ for the true directional curvature $d_i^T H d_i$ (where $H$ is
> > computed on the full dataset, and is thus a deterministic quantity) given $H'$'s
> > eigenvector $d_i$. Figures 2, 8-10, 19, 20 clearly show that the expectation of the
> > estimator systematically does not equal the quantity it is supposed to estimate.
> >
> > The consistency of the mini-batch estimate is guaranteed, since the bias is measured
> > w.r.t. the full-batch directional curvature, so if one chooses the batch size to equal
> > the size of the dataset, the bias vanishes. However, this is practically infeasible, so
> > we _are_ concerned about the computation, as our paper proposes a practical mitigation
> > of mini-batch curvature biases.
> >
> > > Directional curvature is not well defined for random directions $d'$ dependent on
> > > batchsize. Even, I do not know how to use your notations to express it.
> >
> > We humbly disagree. The key insight is that the directional curvature is _also_ a random
> > variable w.r.t. the sampled mini-batch: If we compute the directional curvature along
> > $H'$'s eigenvectors $\{d_i\}$ on an additional mini-batch, we obtain a number $d_i^\top
> > H'' d_i$ (blue dot in Figure 2) that is different from $d_i^\top H' d_i$ (orange dot in
> > Figure 2). Since $H'$ is symmetric, the computation of its real eigenvectors $\{d_i\}$
> > is well-defined.
> >
> > > I am not convinced that we need to replace the leading eigenvectors of Hessian with
> > > those from batchsizes. I can imagine that we can use the leading eigenvectors of full
> > > batchsize for only theoretical analysis and provide good bounds for quantity of
> > > interests such as directional curvature with $d$ is the leading eigenvector of the
> > > full-batch Hessian. I am afraid that the bias is mainly originated from a wrong
> > > perspective towards the problem.
> >
> > We want to emphasize again that our paper is not a theoretical analysis of full-batch
> > eigenspaces. Instead, we propose a practical algorithm for making the eigenspectrum of
> > mini-batch curvature matrices more representative of the full-batch curvature matrix.
> > The systematic bias we uncover is backed by numerous experiments (Figures 1-4, 8-11) and
> > has a detrimental effect on both uncertainty quantification and second-order
> > optimization (Figures 5, 6, 12-17). We are _not_ concerned about providing bounds that
> > you mention.
> >
> > Our perspective towards the problem enables us to develop practical algorithms that
> > successfully mitigate the discrepancy between mini-batch and full-batch estimates, as
> > the figures cited above show.
> >
> > Thanks again for taking the time for this discussion. We hope our responses resolve your
> > remaining concerns and we can reach an agreement on our paper's contributions. If so,
> > please consider re-evaluating our work.

---

### Official Review · Reviewer_3MNZ · 2024-11-04

**Soundness:** 4
**Presentation:** 4
**Contribution:** 4
**Rating:** 8
**Confidence:** 3

**Summary:**

This paper addresses the systematic bias in mini-batch quadratic approximations used for second-order optimization and uncertainty quantification in deep learning. Mini-batch methods distort the quadratic shape, making uncertainty estimates unreliable. The authors provide a theoretical explanation, discuss the impact on optimization, and propose debiasing strategies to improve accuracy.

**Strengths:**

This is an interesting theoretical investigation collaborated with numerical justifications. It teaches me something that I would expect with a nice and convincing narrative. The overall clarity and language of this paper is good. The insights are well-delivered. The theoretical analyses are mostly backed up by numerical experiments.

**Weaknesses:**

I did not identify any obvious weaknesses.

I did not check the proofs.

**Questions:**

How well do the proposed debiasing strategies scale for large foundation models with millions or billions of parameters?

Can debiased mini-batch quadratics improve uncertainty quantification in foundation models, especially in tasks requiring precise uncertainty, like language modeling or image synthesis?

Can the theoretical insights and debiasing techniques be effectively adapted for transformer-based foundation models?

Could reduced bias in mini-batch quadratics enhance transfer learning by improving local loss landscape approximations during fine-tuning?

How might the debiasing methods apply to multi-modal foundation models, where different data modalities add complexity to curvature estimation?

---

> ### Author Response · Authors · 2024-11-21
>
> Dear Reviewer 3MNZ,
>
> Thanks so much for your positive review! We are glad that you found our paper
> *interesting* and appreciated the *overall clarity* and *well-delivered* insights. In
> the following, we will address your questions.
>
> **Disclaimer:** We have uploaded a new version of our manuscript that addresses the
> concerns you raised. We will refer to this updated version in our responses.
>
> > How well do the proposed debiasing strategies scale for large foundation models with
> > millions or billions of parameters?
>
> To address your question, we have conducted an additional experiment on the ViT Little
> architecture (Dosovitskiy et al., 2020, ICLR) on ImageNet data (see Figure 17 in the
> updated manuscript). The results show that (i) our proposed KFAC Laplace approach scales
> to larger models and datasets and (ii) that the curvature biases also exist for these
> types of models.
>
> We have also added a runtime analysis to the manuscript (see Appendix B.6, in particular
> Figure 18). It shows that the runtime overhead of the debiased Laplace approach is
> negligible compared to the single-batch version (given that the debiased version uses
> half the mini-batch size).
>
> > Can debiased mini-batch quadratics improve uncertainty quantification in foundation
> > models, especially in tasks requiring precise uncertainty, like language modeling or
> > image synthesis?
>
> This is an interesting and relevant application of our debiasing strategy. One could,
> e.g., provide curvature estimates for the Low-Rank Adaptation (LoRA) layers (Hu et al.,
> 2021, arXiv) of large language models (LLMs) that are computationally efficient to
> compute (compared to the scale of the entire model). As shown in Figure 17 in our
> updated manuscript, the curvature biases also exist in Transformer architectures.
> Therefore, the debiasing of the LoRA curvatures could significantly improve uncertainty
> quantification via the Laplace approximation in popular LLM architectures.
>
> In Appendix B.10, we show that the curvature bias tends to increase with the number of
> parameters in ConvNet architectures. For LLMs with orders of magnitude more parameters,
> we thus believe that the biases might be even more severe in these models and could
> likely be mitigated with our debiasing strategy.
>
> > Can the theoretical insights and debiasing techniques be effectively adapted for
> > transformer-based foundation models?
>
> Yes. Our theoretical insights and developed debiasing strategies do not assume any
> specific model architecture. The inequalities that formalize the biases (in particular
> Equation (7)) thus hold and the debiasing strategies can, in principle, be applied
> (assuming we can compute a curvature estimate like KFAC). As already mentioned above, we
> did apply our Laplace debiasing techniques to the ViT Little architecture on Imagenet
> data (see Figure 17).
>
> > Could reduced bias in mini-batch quadratics enhance transfer learning by improving
> > local loss landscape approximations during fine-tuning?
>
> Yes. Since our debiasing strategies yield more representative curvature estimates, we
> believe that they could improve the learning dynamics during fine-tuning.
>
> One could argue that our CG experiment (Figure 5) is a fine-tuning scenario on the same task and dataset. Here, the performance with the standard single-batch CG approach collapses, while the debiased version maintains stability.
>
> Sliwa et al. (2024, arXiv) propose using the Laplace approximation as a regularizer during
> fine-tuning in continual learning tasks, which prevents catastrophic forgetting
> and leads to robust models. We believe that eliminating the curvature biases for the
> Laplace approximation would benefit this setting, leading to further improvements in
> both transfer- and continual learning.
>
> > How might the debiasing methods apply to multi-modal foundation models, where
> > different data modalities add complexity to curvature estimation?
>
> From a practical perspective, our debiasing technique could be applied in the same way.
> The different data modalities indeed lead to a more complex curvature structure in loss
> surfaces (Wang et al., 2023, Springer), where it is all the more important to have
> unbiased estimates to not skew the landscape based on biased mini-batch-based estimates.
>
> Thanks again for your thoughtful questions! We hope that you will continue to support
> our paper during the reviewer discussion phase! If you have any follow-up questions,
> please let us know.

---

### Official Review · Reviewer_SWR8 · 2024-11-04

**Soundness:** 4
**Presentation:** 3
**Contribution:** 3
**Rating:** 8
**Confidence:** 3

**Summary:**

This paper investigates biases in mini-batch quadratic approximations, for second-order optimization in deep learning. Authors show that when computing quadratic on mini-batches rather than the full dataset, it introduces curvature distortions, leading to overly narrow approximations. The bias, assigned to “regression to the mean”, affects Newton steps in optimizers and Laplace approx for uncertainty quantification. They provide a “two-batch” strategy to decouple direction selection from magnitude estimation, to align mini batch approx more closely with full-batch behaviour with minimal computational overhead. They demonstrate the improve stability of their strategy on CG and K-FAC experiments.

**Strengths:**

Authors study a well-known but often overlooked issue in mini-batch approx within second-order optimisations and uncertainty quantification for deep learning: the bias in curvature estimates due to mini-batch sampling. While the issue is not new, the approach of a two-batch debasing strategy is an effective and novel (to my knowledge) way to address it and improve the accuracy of the approximates without suffering too much of a computational overhead. This strategy could generalise well across various settings in ML where the full-batch quadratics are intractable. The paper is supported by nice empirical experiments on CG and K-FAC demonstrating the effectiveness of their solution. The authors provide a theoretical intuition for the bias arising from that “regression to the mean” adding an interesting insight.  The experiments are systematically presented and show clear performance difference between single-batch and debasing method . While very technical, authors make an effort to make it as digestible as possible, with the inclusion of figures to visualise the bias and the impact of their debasing strategy.

**Weaknesses:**

The experimental scope is pretty limited, with only ablations on CNN architecture and CIFAR-10/100 datasets. Testing on a wider variety of architectures, such as transformers or larger datasets, would strengthen a lot the paper’s claims on general applicability. Another limitations lies in the lack of ablations to investigate the sensitivity of the two-batch approach to mini-batch size, as well as the composition of mini-batches used for debasing. Without these, it’s unclear how the two-batch batch method would perform under different resource constraints or mini-batch sampling strategies. Last, the paper discusses the computational trade-offs, but does not provide sufficient detail to me on actual runtime and memory costs for different debasing configuration, leaving with an incomplete understanding of the true method’s scalability.

**Questions:**

— How do you anticipate the method to scale or adapt with different architecture and larger dataset, which have different curvature profiles and optimisation challenges?
— Could you clarify the criteria for selecting mini-batches in the two-batch debasing process? Is there any benefit to use overlapping vs. non-overlapping mini-batches, or would a mini-batch with a specific loss profile provide better results?
— Can you provide more details on runtime/memory costs associated with the two-batch approach? While authors claim it roughly doubles, it would be helpful to understand how this really scales with batch size, and whether costs remain manageable as model size and dataset complexity grow

---

> ### Author Response · Authors · 2024-11-21
>
> Dear Reviewer SWR8,
>
> Thanks a lot for your positive feedback on our work! We are happy that you acknowledged
> the novelty of our approach that *could generalise well across various settings in ML*
> and appreciated the *nice empirical experiments* and our efforts to make this rather
> technical paper *as digestible as possible*. We address your concerns and questions in
> the following.
>
> **Disclaimer:** We have uploaded a new version of our manuscript that addresses
> the concerns you raised. We will refer to this updated version in our responses.
>
> > The experimental scope is pretty limited, with only ablations on CNN architecture and
> > CIFAR-10/100 datasets. Testing on a wider variety of architectures, such as
> > transformers or larger datasets, would strengthen a lot the paper's claims on general
> > applicability.
>
> > How do you anticipate the method to scale or adapt with different architecture and
> > larger dataset, which have different curvature profiles and optimisation challenges?
>
> You are right that the empirical results we present in the main text are limited to
> the ALL-CNN-C architecture on CIFAR-10 and CIFAR-100 data. However, we do consider four
> more test problems (see items (C), (D), (E) and (F) in Appendix B.1):
>
> - Figure 19 in our updated manuscript shows the bias in the directional slopes and
>   curvatures - similar to Figure 2 - for a fully-trained WideResNet 40-4 on CIFAR-100
>   data.
> - In Appendix B.10, we study the effect of $P$ (the number of parameters in the model)
>   on the curvature biases for a variety of ConvNet models.
> - Figures 16 and 17 present additional results with the debiased KFAC Laplace approach
>   for a ResNet-50 and the ViT Little (Dosovitskiy et al., 2020, ICLR) architecture on
>   Imagenet data.
>
> > Another limitations lies in the lack of ablations to investigate the sensitivity of
> > the two-batch approach to mini-batch size, as well as the composition of mini-batches
> > used for debasing. Without these, it’s unclear how the two-batch batch method would
> > perform under different resource constraints or mini-batch sampling strategies.
>
> > Could you clarify the criteria for selecting mini-batches in the two-batch debasing
> > process? Is there any benefit to use overlapping vs. non-overlapping mini-batches, or
> > would a mini-batch with a specific loss profile provide better results?
>
> Our experiments on the debiased Laplace approximation (Section 6.2 in the main paper and
> Figures 14-17 in the appendix) consider different mini-batch sizes. We observe that the
> two-batch approach consistently outperforms the similar-cost single-batch approach in
> terms of its ability to mimic the full-batch behavior. However, regarding the debiased
> CG experiments, we agree with you about the lack of ablation studies to
> investigate the sensitivity of the two-batch approach with respect to mini-batch size.
> Therefore, we have added results for two additional mini-batch sizes in Figure 13 of the
> updated manuscript. While the single-batch runs quickly diverge, the debiased approaches
> maintain stability across all mini-batch sizes.
>
> For now, we use the most common mini-batch sampling strategy, which is uniform random
> sampling, i.e. we shuffle the training data and then split it into mini-batches of equal
> size. Deviating from this strategy (e.g. by using samples with a specific loss profile)
> would skew the sample distribution away from the true generative process, which would
> introduce new biases.
>
> In the extreme case where the two mini-batches perfectly overlap, we are back at the
> single-batch strategies with the associated biases. In the other extreme where the two
> mini-batches are completely independent, we decouple the directions and magnitudes
> completely, effectively removing these biases. Loosening the independence constraint (by
> allowing for some overlap) would re-introduce biases. We would expect a gradual
> increase in the biases as the overlap between the two mini-batches increases.
>
> > Last, the paper discusses the computational trade-offs, but does not provide
> > sufficient detail to me on actual runtime and memory costs for different debasing
> > configuration, leaving with an incomplete understanding of the true method’s
> > scalability.
>
> > Can you provide more details on runtime/memory costs associated with the two-batch
> > approach? While authors claim it roughly doubles, it would be helpful to understand
> > how this really scales with batch size, and whether costs remain manageable as model
> > size and dataset complexity grow
>
> Thanks for pointing this out. To address your concern, we have added a runtime analysis
> to Appendix B.6 (see Figure 18 in the new version of the paper). It shows that the
> computational runtime overhead incurred by the debiased KFAC Laplace approach is
> negligible compared to the single-batch approach (given that we halve the mini-batch
> size for the debiased version).

---

> ### Author Response · Authors · 2024-11-21
>
> In terms of memory overhead, the debiased KFAC Laplace approach requires an additional
> KFAC approximation on a second mini-batch. One such approximation requires roughly
> $92$ MB using the ALL-CNN-C architecture. With a naive implementation, the memory overhead
> caused by the debiased approach is thus another $92$ MB. However, this memory overhead
> can be significantly reduced by constructing the debiased blocks one by one (more details
> on the bottom of page 37), bringing the memory overhead down to $24$ MB on ALL-CNN-C.
>
> For the debiased CG algorithm, we explain in Section A.2 that an efficient computation
> only causes a memory overhead of two vectors of the size of the parameter space. In the experimental setting used in Figure 5, this amounts to about $11$ MB.
>
> Thanks again for your thoughtful feedback and interesting questions. We hope that you
> will support our paper during the reviewer discussion phase. If you have any further
> questions or concerns, please let us know.

---

### Author Response · Authors · 2024-12-04

Dear Reviewers,

Thank you so much for your constructive feedback and thoughtful questions!

We are delighted that three of the four Reviewers (3MNZ, 9wSH and SWR8) clearly
recommend accepting our paper with a score of 8. We hope that our thorough rebuttal for
Reviewer ddR4 helped resolve the misunderstandings and addressed all concerns
satisfactorily.

In response to your suggestions, we have made the following updates to our paper (the
references we provide below refer to this updated version):
- We extended our experiment on debiased CG by two additional mini-batch sizes to
  investigate its sensitivity with respect to mini-batch size (see Figure 13). While the
  single-batch runs quickly diverge, the debiased approaches maintain stability across
  all mini-batch sizes.
- We have also added a runtime analysis to the manuscript (see Figure 18). It shows that
  the runtime overhead of the debiased Laplace approach is negligible compared to the
  single-batch version.
- We have conducted an additional experiment on the ViT Little architecture on ImageNet
  data (see Figure 17). The results show that (i) the curvature biases also exist for
  these types of models, (ii) our proposed KFAC Laplace approach scales to larger models
  and datasets, and (iii) that the debiased approach effectively mitigates the biases.

These updates, we believe, have further strengthened the paper, addressing key points
raised during the review process. We appreciate the constructive and positive
evaluations and feel encouraged by the positive feedback!

---

### Meta-Review · Area_Chair_zeqF · 2024-12-23

**Metareview:**

This paper investigates systematic biases in mini-batch quadratic approximations used for second-order optimization and uncertainty quantification in deep learning. The authors demonstrate that using mini-batches to estimate curvature leads to distortions in the quadratic shape, particularly overestimating top eigenvalues. They provide theoretical explanations for this phenomenon and propose practical debiasing strategies using a two-batch approach. The paper shows how these biases affect applications like conjugate gradient optimization and Laplace approximation for uncertainty quantification, and demonstrates that their debiasing methods can significantly improve performance in these areas.

The paper's strengths lie in its identification of an important issue in deep learning optimization and uncertainty quantification, providing both theoretical insights and empirical evidence for the observed biases. The proposed debiasing strategies are practical and computationally efficient, showing clear improvements in concrete applications. The paper is well-presented with clear explanations and visualizations.

Initially, the experimental scope was somewhat limited, focusing mainly on CNN architectures and CIFAR datasets. However, the authors addressed this concern in their rebuttal by expanding their experiments to include larger models and datasets.

The paper addresses a significant and previously underexplored issue in deep learning, with potential impact across multiple areas including optimization and uncertainty quantification. The proposed debiasing strategies are novel, practical, and show clear improvements over existing methods. The majority of reviewers strongly support acceptance, and the authors have adequately addressed reviewer concerns and expanded their experiments in the rebuttal.

**Additional Comments On Reviewer Discussion:**

During the rebuttal period, reviewers raised concerns about the limited experimental scope, sensitivity to mini-batch size, and details on runtime and memory costs. The authors responded by adding experiments on the ViT Little architecture with ImageNet data, demonstrating that their approach scales to larger models and that the curvature biases exist in transformer architectures as well. They also provided results for additional mini-batch sizes in their CG experiments, showing that their debiased approach maintains stability across different batch sizes. A detailed runtime analysis was added, showing that the overhead of their debiased Laplace approach is negligible compared to the single-batch version.

One reviewer questioned the theoretical foundations, particularly the definition and measurement of bias. The authors clarified their approach and provided additional explanations to address these concerns.

The authors satisfactorily addressed the major concerns raised by reviewers. The additional experiments and analyses provided in the rebuttal significantly strengthen the paper. While some limitations remain, the paper's novel contributions and potential impact outweigh these shortcomings. The strong support from three out of four reviewers, combined with the authors' thorough responses, reinforces the decision to accept this paper.

---

### Decision · Program_Chairs · 2025-01-22

Accept (Poster)